# Learning-Augmented Hierarchical Clustering

**Vladimir Braverman** [1 2 3]   **Jon C. Ergun** [4]   **Chen Wang** [2 5]   **Samson Zhou** [5]

## Abstract

Hierarchical clustering (HC) is an important data analysis technique in which the goal is to recursively partition a dataset into a tree-like structure while grouping together similar data points at each level of granularity. Unfortunately, for many of the proposed HC objectives, there exist strong barriers to approximation algorithms with the hardness of approximation. We consider the problem of hierarchical clustering given auxiliary information from natural oracles in the learning-augmented framework. Our main results are algorithms that given learning-augmented oracles, compute efficient approximate HC trees for the celebrated Dasgupta's and Moseley-Wang objectives that overcome known hardness barriers.

## 1. Introduction

Hierarchical clustering (HC) is a popular data analysis technique that recursively partitions a dataset throughout a tree-like structure, so that similar data points are grouped together at different levels of granularity. Specifically, the input is a set of $n$ data points and a measure of similarity or dissimilarity between the points, which induces a weighted graph whose vertices represent the data points and whose edge weights represent the pairwise measure between the vertices. The output is a binary dendrogram, which is a rooted tree whose leaves represent the individual data points and whose internal nodes each represent a cluster of the data points in its subtree, thus providing a hierarchical representation of relationships within the dataset.

Hierarchical clustering has several advantages over flat clusterings such as $k$-means or $k$-median, where the dataset is partitioned into a fixed number of clusters. For example,

Alphabetical order of author names. [1]Johns Hopkins University [2]Rice University [3]Google Research [4]Carnegie Mellon University [5]Texas A&M University. Correspondence to: Vladimir Braverman <vova@cs.jhu.edu>, Jon C. Ergun <ergunjon0@gmail.com>, Chen Wang <chen.wang.research@gmail.com>, Samson Zhou <samsonzhou@gmail.com>.

*Proceedings of the $42^{nd}$ International Conference on Machine Learning*, Vancouver, Canada. PMLR 267, 2025. Copyright 2025 by the author(s).

the "correct" number of clusters in flat clusterings is often a difficult question that is the focus of a sequence of works dating back to the 1950s (Thorndike, 1953). In hierarchical clustering, there is no fixed number of clusters that needs to be determined in advance. Another advantage of hierarchical clustering is that the dendrogram simultaneously captures structure at all levels of granularity, whereas flat clustering does not identify further structure inside each of the clusters. Hence, hierarchical clustering arises in various applications where data exhibits hierarchical structure, such as biology and phylogenetics (Sneath et al., 1973; Sotiriou et al., 2003), image and text analysis (Steinbach et al., 2000), and community detection (Leskovec et al., 2020).

Despite a wealth of heuristics for both agglomerative bottom-up (Ward Jr, 1963) and divisive top-down approaches (Guénoche et al., 1991), formal mathematical understanding of hierarchical clustering often stagnated due to the absence of well-posed objectives until a relatively recent work by Dasgupta (Dasgupta, 2016). Subsequently, additional objectives (Moseley & Wang, 2017; Cohen-Addad et al., 2019) were proposed to quantify the performance of dendrograms with $n$ leaves, so that high-revenue similarity trees and low-cost dissimilarity trees correspond to desirable hierarchical partitions of the dataset.

For a number of these objectives, various algorithms have been proven to achieve specific approximation guarantees. For example, a divisive clustering algorithm based on the sparsest cut subroutine was shown to give an $O(\sqrt{\log n})$ approximation (Charikar & Chatziafratis, 2017b; Cohen-Addad et al., 2019; Deng et al., 2025) for Dasgupta's objective (Dasgupta, 2016), while the long-used agglomerative heuristic was shown to give a 2-approximation for the dissimilarity objective proposed by (Cohen-Addad et al., 2019) and a $\frac{1}{3}$-approximation for the similarity objective proposed by (Moseley & Wang, 2017), i.e., the Moseley-Wang (MW) objective. These objectives were further explored in the contexts of better approximation factors (Chatziafratis et al., 2020b; Alon et al., 2020), sublinear computation models (Rajagopalan et al., 2021; Assadi et al., 2022; Agarwal et al., 2022), and graphs with special properties (Charikar et al., 2019b; Manghiuc & Sun, 2021).

Unfortunately, (Charikar et al., 2019a) showed that average-linkage cannot do better than $\frac{3}{2}$-approximation for the for-

mer objective or better than $\frac{1}{3}$-approximation for the latter. More general hardness of approximation results were given, showing the impossibility of achieving roughly 1.003-approximation (Chatziafratis et al., 2020a) for dissimilarity under the Unique Games Conjecture and the impossibility of achieving $(1 - C)$-approximation for the Moseley-Wang objective (Chatziafratis et al., 2020b) under the Small Set Expansion hypothesis, for a fixed constant $C > 0$. Moreover, for the Dasgupta objective, (Charikar & Chatziafratis, 2017a; Roy & Pokutta, 2017) showed that under the Small Set Expansion hypothesis, there is no constant approximation in polynomial time for *any constant*. Thus, we seek new practical approaches that enable better approximation guarantees without assumptions about the underlying dataset or weight function.

**Learning-augmented algorithms.** We draw inspiration from the recent advances in the predictive capabilities of machine learning models. On one hand, datasets often have additional auxiliary information that can be used to improve algorithmic performance if accurate. For example, in many applications, the input dataset can retain insightful patterns exhibited by similar datasets generated from previous instances. On the other hand, machine learning models lack provable guarantees and can result in wildly inaccurate predictions when generalizing to unfamiliar inputs (Szegedy et al., 2014). Nevertheless, *learning-augmented algorithms* (Mitzenmacher & Vassilvitskii, 2020) that overcome worst-case computational limits have been designed for a number of applications, such as warm-starts for faster algorithms (Dinitz et al., 2021; Chen et al., 2022c; Davies et al., 2023), data structures optimized for specific query distributions (Kraska et al., 2018; Mitzenmacher, 2018; Lin et al., 2022; Fu et al., 2025), online algorithms with some "forecast" of the future (Purohit et al., 2018; Gollapudi & Panigrahi, 2019; Lattanzi et al., 2020; Wang et al., 2020; Wei & Zhang, 2020; Bamas et al., 2020; Im et al., 2021; Lykouris & Vassilvitskii, 2021; Aamand et al., 2022; Anand et al., 2022; Azar et al., 2022; Grigorescu et al., 2022; Khodak et al., 2022; Gupta et al., 2022; Jiang et al., 2022; Antoniadis et al., 2023; Shin et al., 2023; Benomar & Perchet, 2023), input-sensitive sketches for more space-efficient streaming algorithms (Hsu et al., 2019; Indyk et al., 2019; Jiang et al., 2020; Chen et al., 2022b;a; Li et al., 2023), and classical NP hard problems (Braverman et al., 2024; Cohen-Addad et al., 2024; Braverman et al., 2025).

For clustering problems, (Ergun et al., 2022; Nguyen et al., 2023; C. S. et al., 2024) introduced *flat* clustering algorithms that use polynomial runtime and achieve approximation guarantees beyond NP hardness limits. Though their techniques are specific to $k$-means and $k$-median clustering, their work nevertheless serves as an important conceptual message that demonstrates machine learning oracles can be used to improve upon traditional techniques for cluster analysis. Furthermore, for graph-base problems, recent results by (Cohen-Addad et al., 2024; Braverman et al., 2024; Dong et al., 2025) have shown that natural learning-augmented oracles could help overcome NP-hardness constraints as well. We thus ask whether machine learning models can be used to provably improve (graph-based) *hierarchical* clustering.

## 1.1. Our Contributions

In this paper, we consider the problem of hierarchical clustering given a possibly erroneous oracle that uses auxiliary information, e.g., through clusterings of similar datasets, to provide local information about the relationship between queried data points. In particular, we consider a *splitting oracle* that, on an input query of a triplet $(u, v, w)$ of vertices, outputs the vertex that is first separated away from the other two with respect to an optimal or near-optimal hierarchical clustering tree. In other words, if the oracle is consistent with some ground-truth tree, it will output the vertex that is *not* in the same subtree as the other two vertices, under their least common ancestor. We remark that such oracle advice is natural due to the plethora of machine learning models that are trained on related instances of graphs, where the triplet relationships are already labeled.

Using triplet split-away information is common in the literature, and there have been results explored in similar settings, e.g., tree reconstruction with *accurate* triplet relationships given (Aho et al., 1981), triplets are given as constraints (Chatziafratis et al., 2018), noisy triplet information with fresh randomness (Emamjomeh-Zadeh & Kempe, 2018), and algorithms with quartet information (Jiang et al., 2000; Snir & Yuster, 2011; Alon et al., 2014). Furthermore, the reconstruction of phylogenetic CSPs, which provides an oracle with a similar form to ours, is an extensively studied line of work (see, e.g., (Chatziafratis & Makarychev, 2023)). From the machine learning perspective, it is possible to learn such oracles in the PAC learning framework (see Appendix J for details).

In line with existing literature on learning-augmented algorithms, we investigate a stochastic and independently responding splitting oracle, where randomness is introduced only *once*. Specifically, this implies that the oracle correctly responds with a probability of $p$ for some constant $p > 1/2$, independently across vertex triplets. Additionally, repeated queries of the same triplet consistently yield the same (possibly erroneous) responses, which rules out basic boosting strategies such as repeatedly making the same query to the oracle. We remark this splitting oracle mirrors numerous machine learning models that are trainable with data yet exhibit inherent noise.

We now present our main results for learning-augmented hierarchical clustering. We first show that for the Dasgupta objective, we can use such a splitting oracle to achieve a

constant factor approximation in polynomial time.

**Theorem 1.** *There exists an algorithm that, given a weighted undirected graph $G = (V, E, w)$ and a splitting oracle $\mathcal{O}$, with high probability, in polynomial time and $O(n^3)$ queries computes a hierarchical clustering tree $\mathcal{T}$ such that $\mathsf{cost}_G(\mathcal{T}) \leq O(1) \cdot \mathsf{OPT}^{\mathrm{Das}}(G)$, where $\mathsf{OPT}^{\mathrm{Das}}(G)$ is the cost of the optimal hierarchical clustering tree $\mathcal{T}^*$, i.e., $\mathsf{OPT}^{\mathrm{Das}}(G) = \mathsf{cost}_G(\mathcal{T}^*)$.*

By comparison, (Charikar & Chatziafratis, 2017a; Roy & Pokutta, 2017) showed that there is no polynomial-time algorithm that could achieve any constant approximation to Dasgupta's objective, under the Small Set Expansion hypothesis. Hence, Theorem 1 illustrates that the power of a splitting oracle can be used to break complexity hardness limitations. On the other hand, we remark that the runtime of the algorithm of Theorem 1, although polynomial, is perhaps embarrassingly large. We thus give an algorithm that uses the splitting oracle and $\tilde{O}(n^3)$ time to achieve approximation guarantees beyond the current state-of-the-art oblivious algorithms.

**Theorem 2.** *There exists an algorithm that, given a weighted undirected graph $G = (V, E, w)$ and a splitting oracle $\mathcal{O}$, with high probability, in $O(n^3 \log n)$ time and $O(n^3)$ queries computes a hierarchical clustering tree $\mathcal{T}$ such that $\mathsf{cost}_G(\mathcal{T}) \leq O(\sqrt{\log \log n}) \cdot \mathsf{OPT}^{\mathrm{Das}}(G)$, where $\mathsf{OPT}^{\mathrm{Das}}(G)$ is the cost of the optimal hierarchical clustering tree $\mathcal{T}^*$, i.e., $\mathsf{OPT}^{\mathrm{Das}}(G) = \mathsf{cost}_G(\mathcal{T}^*)$.*

By comparison, the best-known polynomial-time oblivious algorithm achieves $O(\sqrt{\log n})$-approximation (Charikar & Chatziafratis, 2017b; Cohen-Addad et al., 2019). Thus our algorithm that achieves $O(\sqrt{\log \log n})$ approximation gives very competitive practical bounds – the improvement of our algorithm is approximately 2.3 times better for $n = 10^{10}$.

Turning our attention to the Moseley-Wang objective, we similarly show that a splitting oracle can also be used to achieve any constant factor approximation in polynomial time.

**Theorem 3.** *There exists an algorithm that, given a weighted undirected graph $G = (V, E, w)$ and a splitting oracle $\mathcal{O}$, with high probability, in $O(n^2 \cdot \mathrm{polylog}\, n)$ time and $O(n^2)$ queries computes a hierarchical clustering tree $\mathcal{T}$ such that $\mathsf{rev}_G(\mathcal{T}) \geq (1 - o(1)) \cdot \mathsf{OPT}^{\mathrm{MW}}(G)$, where $\mathsf{OPT}^{\mathrm{MW}}(G)$ is the revenue of the optimal hierarchical clustering tree $\mathcal{T}^*$, i.e., $\mathsf{OPT}^{\mathrm{MW}}(G) = \mathsf{rev}_G(\mathcal{T}^*)$.*

We note that (Chatziafratis et al., 2020b) showed the APX-hardness of the $(1 - C)$ approximation for Moseley-Wang objective under the Small Set Expansion hypothesis, for a fixed constant $C \in (0, 1)$. As such, Theorem 3 again also shows the power of splitting oracles to overcome impossibility barriers. Since a $n$-vertex graph could have input

size as large as $\Theta(n^2)$, the time complexity in Theorem 3 is near-linear in the worst case.

Finally, we observe that our algorithms possess favorable properties that are extremely amenable to sublinear algorithms. As such, we can obtain the following results in the streaming and parallel computation (PRAM) settings.

**Theorem 4.** *In the single-pass graph streaming and the PRAM settings, there exists:*

- *a single-pass (dynamic) streaming algorithm that, given a weighted undirected graph $G = (V, E, w)$ in a $\mathrm{poly}(n)$-length dynamic stream and an offline splitting oracle $\mathcal{O}$, with high probability, uses $O(n \cdot \log^3 n)$ bits of space and polynomial time computes a hierarchical clustering tree $\mathcal{T}$ such that $\mathsf{cost}_G(\mathcal{T}) \leq O(1) \cdot \mathsf{OPT}^{\mathrm{Das}}(G)$ (Theorem 21).*

- *a PRAM algorithm that, given a weighted undirected graph $G = (V, E, w)$ and a splitting oracle $\mathcal{O}$, with high probability, in $O(n^2 \cdot \mathrm{polylog}\, n)$ work and $\log^3 n$ depth computes a hierarchical clustering tree $\mathcal{T}$ such that $\mathsf{rev}_G(\mathcal{T}) \geq (1 - o(1)) \cdot \mathsf{OPT}^{\mathrm{MW}}(G)$ (Theorem 22).*

Our results in the sublinear settings similarly outperform the state-of-the-art in the HC algorithms without oracle advice. For instance, in the single-pass graph streaming setting, (Assadi et al., 2022; Agarwal et al., 2022) designed semi-streaming algorithms with $O(1)$ approximation but exponential time. By comparison, our algorithm only uses polynomial time, leveraging the advantage of the splitting oracle.

## 2. Preliminaries

We present the definition of the hierarchical clustering problem, our splitting oracle model, and the HC objectives in this section.

### 2.1. The hierarchical clustering problem

We consider the hierarchical clustering problem with a splitting oracle $\mathcal{O}$. The hierarchical clustering problem is defined as follows. We are given an $n$-vertex weighted undirected input graph $G = (V, E, w)$, and our goal is to produce a binary tree $\mathcal{T}$ whose root node corresponds to the vertex set $V$ and the leaves represent the singleton vertices. The vertices set contained in the internal nodes form a Laminar set family: suppose node $x$ has children $(y, z)$, it represents a split $S_x \to (S_y, S_z)$, where $S_x = S_y \cup S_z$. In this work, we assume without loss of generality $n \geq 200 \log n$ – the bound holds for any $n \geq 2500$, and if $n$ is a constant we can simply use a brute-force algorithm.

Hierarchical clustering trees only define a data structure, and there are many ways to construct "valid" HC trees.

What eventually matters is to construct "good" HC trees – a notion that does not have a universal way to define. Popular approaches include heuristics, which work well subjectively but lack formal guarantees, and objective functions, which provide rigorous frameworks to study the *optimal* trees and the *approximation algorithms*. In recent years, the latter approach has attracted considerable attention with popular objective functions by (Dasgupta, 2016; Moseley & Wang, 2017; Cohen-Addad et al., 2019).

**Notation.** For each internal node $x$ in $\mathcal{T}$, we use $\texttt{leaves}_{\mathcal{T}}[x]$ to denote the leaves in the induced subtree of $x$. Each internal node of an HC tree can be described by *lowest common ancestor* (LCA) of vertices. For two vertices $(u, v)$ on the leaves of $\mathcal{T}$, we use $\texttt{LCA}_{\mathcal{T}}(u, v)$ to denote the node that is the lowest common ancestor of $u$ and $v$. We can further generalize this notion to a set of vertices, i.e., for a set $X \subseteq V$, the node $\texttt{LCA}_{\mathcal{T}}(X)$ refers to the lowest common ancestor of *all* vertices in $X$. For a set $X$, we call the induced subtree $\mathcal{T}_X$ a *maximal subtree* of $\mathcal{T}$ if $\texttt{leaves}_{\mathcal{T}}[\texttt{LCA}_{\mathcal{T}_X}(X)] = X$, i.e., the lowest common ancestor of $X$ in $\mathcal{T}_X$ induces all leaves of $X$ in $\mathcal{T}$.

Let $r$ be the root of a hierarchical clustering tree $\mathcal{T}$, and for any internal node $x$, we let $\texttt{dist}_{\mathcal{T}}(r, x)$ be the number of edges on the shortest path between $r$ and $x$. We say node $x$ on level $\texttt{level}_{\mathcal{T}}(x)$ is a *higher level* node than node $y$ with level $\texttt{level}_{\mathcal{T}}(y)$ in $\mathcal{T}$ if $\texttt{dist}_{\mathcal{T}}(r, x) > \texttt{dist}_{\mathcal{T}}(r, y)$. For two internal nodes $x$ and $y$ in $\mathcal{T}$, we use $x = \texttt{pa}(y)$ to denote the relationship of $x$ being the *parent node* of $y$. Note that if $x = \texttt{pa}(y)$, it automatically implies that $\texttt{level}_{\mathcal{T}}(y) = \texttt{level}_{\mathcal{T}}(x) + 1$.

**The split-away vertex.** Note that in any hierarchical clustering tree, if we look at a triplet of vertices $(u, v, w)$, there must exist a vertex that *split away* from the two others in the optimal tree $\mathcal{T}^*$, i.e., two vertices with a LCA that is same as the LCA of *all* three vertices in $\mathcal{T}^*$. Formally, for a triplet of vertices $(u, v, w)$, we define "$w$ splits away from $(u, v)$" as follows.

**Definition 5** (Split-away vertex). Let $G = (V, E, w)$ be a $n$-vertex graph, and let $\mathcal{T}^*$ be the optimal HC tree for $G$. Given a triplet of vertices $(u, v, w)$, we say $w$ *splits away from* $(u, v)$ (in $\mathcal{T}^*$) if $\texttt{LCA}_{\mathcal{T}^*}(w, u)$ (resp. $\texttt{LCA}_{\mathcal{T}^*}(w, v)$) is *equal to* $\texttt{LCA}_{\mathcal{T}^*}(\{u, v, w\})$.

### 2.2. The splitting oracle model

We study the hierarchical clustering problem with a natural oracle advice model. In particular, we assume an oracle $\mathcal{O} : V \times V \times V \to V$ that takes a triplet of vertices $(u, v, w)$, probabilistically correctly returns the vertex that "split away" from the other two vertices in the *optimal tree*[1]. The formal

[1]We provide a discussion for splitting oracles with an *approximately* optimal HC tree in Appendix I.2.

definition is given as follows.

**Definition 6** (The splitting oracle for hierarchical clustering). Let $G = (V, E)$ be a $n$-vertex graph, and let $\mathcal{T}^*$ be the optimal hierarchical clustering tree of $G$. The oracle $\mathcal{O} : V \times V \times V \to V$ is a function that upon being queried with a triplet of vertices $(u, v, w)$, responds as follows

- with probability $p$, the correct answer on which vertex splits away from the two others in $\mathcal{T}^*$.

- with probability $(1 - p)$, an arbitrary (adversarial) answer on which vertex splits away from the two other vertices.

The randomness is taken independently over all the queries and is *fixed* across different queries on the same triplet. We assume each query to the oracle takes $O(1)$ time.

Assuming the correct probability of an oracle is some constant $p > 1/2$ is very common in the literature, especially for graph problem (Braverman et al., 2024; Cohen-Addad et al., 2024; Dong et al., 2025). For the convenience of presentation, we assume $p = \frac{9}{10}$ in this paper, and we provide a discussion about general success probabilities in Appendix I.1. Observe that by the fixed randomness for each triplet, there are at most $\binom{n}{3}$ many answers that $\mathcal{O}$ can have. This setting rules out trivial algorithms that simply get the correct by querying multiple times and boosting the success probability.

Several hierarchical clustering objectives are proved to be hard to approximate in polynomial time under very plausible complexity assumptions. As such, our goal is to explore whether we can obtain better approximation guarantees with the splitting oracle.

### 2.3. Objective functions for hierarchical clustering

We introduce the objective functions for hierarchical clustering we are going to discuss in this paper. These include the Dasgupta *minimization* (cost) objective (Dasgupta, 2016) and Moseley-Wang *maximization* (revenue) objective (Moseley & Wang, 2017). We start with the minimization objective as prescribed by (Dasgupta, 2016).

**Problem 1** (HC under Dasgupta's cost function). Given an $n$-vertex weighted graph $G = (V, E, w)$ with vertices corresponding to data points and edges measuring their similarity, create a rooted tree $\mathcal{T}$ whose leaf nodes are $V$. The goal is to *minimize* the cost of this tree $\mathcal{T}$ defined as

$$\texttt{cost}_G(\mathcal{T}) := \sum_{e=(u,v)\in E} w(e) \cdot |\texttt{leaves}_{\mathcal{T}}[\texttt{LCA}_{\mathcal{T}}(u, v)]|, \tag{1}$$

where $|\texttt{leaves}_{\mathcal{T}}[\texttt{LCA}_{\mathcal{T}}(u, v)]|$ is the number of leaf-nodes in the sub-tree of $\mathcal{T}$ rooted at the lowest common

ancestor of $u$ and $v$. We use $\mathsf{OPT}^{\text{Das}}(G)$ to denote the cost of an optimal HC tree under Dasgupta's cost for the graph $G$.

Roughly speaking, Dasgupta's objective accumulates the cost on an edge $(u, v)$ by the number of leaves *inside* the subtree where $u$ and $v$ are first split. In contrast, the Moseley-Wang objective focuses on the dual of Dastupta's objective: it gathers the revenue on an edge $(u, v)$ by the number of leaves *outside* the subtree where $u$ and $v$ are first split. Formally, the Moseley-Wang objective can be given as follows.

**Problem 2** (HC under Moseley-Wang revenue function). Given an $n$-vertex weighted graph $G = (V, E, w)$ with vertices corresponding to data points and edges measuring their similarity, create a rooted tree $\mathcal{T}$ whose leaf nodes are $V$. The goal is to *maximize* the revenue $\mathsf{rev}_G(\mathcal{T})$ of a tree $\mathcal{T}$ defined as

$$\sum_{e=(u,v)\in E} w(e) \cdot (n - |\texttt{leaves}_{\mathcal{T}}[\texttt{LCA}_{\mathcal{T}}(u, v)]|)$$
$$= \sum_{e=(u,v)\in E} w(e) \cdot |\texttt{non-leaves}_{\mathcal{T}}[\texttt{LCA}_{\mathcal{T}}(u, v)]|,$$
$$(2)$$

where $|\texttt{leaves}_{\mathcal{T}}[\texttt{LCA}_{\mathcal{T}}(u, v)]|$ is the number of leaf-nodes in the sub-tree of $\mathcal{T}$ rooted at the lowest common ancestor of $u$ and $v$, and $|\texttt{non-leaves}_{\mathcal{T}}[\texttt{LCA}_{\mathcal{T}}(u, v)]|$ is the number of nodes that are *not* among $\texttt{leaves}_{\mathcal{T}}[\texttt{LCA}_{\mathcal{T}}(u, v)]$. We use $\mathsf{OPT}^{\text{MW}}(G)$ to denote the revenue of an optimal HC tree under Dasgupta's cost for the graph $G$.

Observe that both objectives are *composeable* w.r.t. edges, i.e., it is possible to divide the total objective to objectives induced by each (or each set of) edge(s). For any HC tree $\mathcal{T}$ and any set of edge $E_1 \subseteq E$, we use $\mathsf{rev}_G(\mathcal{T}, E_1)$ and $\mathsf{cost}_G(\mathcal{T}, E_1)$ to denote the revenue and the cost induced by the edges in $E_1$.

By a straightforward calculation, one can show that for any HC tree $\mathcal{T}$, there is $\mathsf{rev}_G(\mathcal{T}) = \sum_{e=(u,v)\in E} w(e) \cdot n - \mathsf{cost}_G(\mathcal{T})$. Since $\sum_{e=(u,v)\in E} w(e) \cdot n$ is a deterministic function of the graph $G$ itself, the optimal HC tree $\mathcal{T}^*$ under the two objectives are the same. However, the two objectively admits vastly different *approximation* algorithms. In particular, for the minimization objective, (Dasgupta, 2016) and the following work (Roy & Pokutta, 2016; Charikar & Chatziafratis, 2017b) showed that we can achieve an $O(\sqrt{\log n})$ approximation in polynomial time, and there is no $O(1)$ approximation in polynomial time assuming Small Set Expansion (SSE) hypothesis. On the other hand, for the revenue maximization objective, (Moseley & Wang, 2017) proved that the average-linkage heuristic can achieve a $1/3$ approximation in polynomial time. Therefore, we would

naturally expect different results for hierarchical clustering with the splitting oracle with the two objectives.

# 3. The Definitions and Results for Partial Hierarchical Clustering Trees

A technical backbone of our algorithms in this paper is the *partial hierarchical clustering tree*. Roughly speaking, these structures replicate the organizational framework of the optimal HC tree, exhibiting only minor "ambiguity" within small subsets of vertices. In this section, we formally define the strong and weak partial trees and give efficient construction algorithms for them. We remark that our constructions of the partial HC trees are entirely based on the vertex set $V$ and the oracle $\mathcal{O}$ of the input graph, irrespective of specific objective functions. This inherent independence renders our partial HC trees highly versatile and potentially of significant interest in their own right.

## 3.1. Partial hierarchical clustering trees

We start by showing the definition of partial hierarchical clustering trees, which are very similar to the normal HC trees: the internal nodes represent subsets of vertices, and the leaves are individual vertices. However, in partial HC trees, we allow a collection of vertices that are not *too large* to have *unknown* local clustering, and we simply represent the whole set of vertices as a leaf node in the tree. The formal definition is as follows.

**Definition 7** (Partial hierarchical clustering trees). A partial hierarchical clustering tree $\mathcal{I}$ is a binary tree such that

1. The root represents the vertex set $V$.

2. For a node $x$ with children $(y, z)$, it represents a split $S_x \to (S_y, S_z)$, where $S_x = S_y \cup S_z$.

3. The leaves of $\mathcal{I}$ corresponds to
   - either a singleton vertex in $V$.
   - or a set of vertices $S \subseteq V$ such that $S \leq 50000 \log n$. In this case, we call the leave a *super-vertex*.

Compared to the full hierarchical clustering tree, the partial HC tree allows the leaves to be 'contracted' vertices with size at most $O(\log n)$. We now define a partial tree that is *strongly consistent* with a hierarchical clustering tree $\mathcal{T}$.

**Definition 8** (Partial tree *strongly* consistent with $\mathcal{T}$). Let $\mathcal{I}$ be a partial hierarchical clustering tree and let $\mathcal{T}$ be a (standard) hierarchical clustering tree. We say $\mathcal{I}$ is *strongly consistent* with $\mathcal{T}$ if

1. (*Strong contraction property*) Each super-vertex induces a maximal subtree in $\mathcal{T}$.

2. (*Subtree preservation property*) For any pair of leaves $(x, y)$ in $\mathcal{I}$, let $X$ and $Y$ be the set of leaves corresponding to $x$ and $y$ in $\mathcal{T}$ (recall that the leaves of $\mathcal{I}$ can be super-vertices). The subtree induced by $\text{LCA}_{\mathcal{I}}(x, y)$ contains the *exactly* the same set of vertices as induced by $\text{LCA}_{\mathcal{T}}(X \cup Y)$.

In other words, a partial tree $\mathcal{I}$ is strongly consistent with $\mathcal{T}$ if there exists a way to locally arrange tree structures for every super-vertex to *exactly recover* $\mathcal{T}$. An illustration of the strongly consistent partial tree (w.r.t. $\mathcal{T}$) can be found in Figure 1.

The strong partial tree is a very helpful data structure for HC. Nevertheless, finding such a strong partial tree could be challenging. As such, we also define partial trees that are *weakly* consistent with the tree $\mathcal{T}$ as follows.

**Definition 9** (Partial tree *weakly* consistent with $\mathcal{T}$)**.** Let $\mathcal{I}$ be a partial hierarchical clustering tree and let $\mathcal{T}$ be a (standard) hierarchical clustering tree. We say $\mathcal{I}$ is *weakly consistent* with $\mathcal{T}$ if

1. (*Weak contraction property*) Each super-vertex corresponds to a collection of maximal subtrees in $\mathcal{T}$, i.e., $\cup_i V_i$ such that each $V_i$ satisfies

$$\texttt{leaves}_{\mathcal{T}}[\text{LCA}_{\mathcal{T}}(V_i)] = V_i.$$

Furthermore, the collection $\cup_i V_i$ is with out-degree at most 2 in $\mathcal{T}$ such that

   (a) At most one edge is connected to a node that is the parent of the LCA of $\cup_i V_i$.

   (b) At most one edge is connected to a node that is a sibling of a maximal subtree of $\mathcal{T}$ induced by (a subset of) $\cup_i V_i$.

2. (*Subtree preservation property*) For any pair of leaves $(x, y)$ in $\mathcal{I}$, let $X$ and $Y$ be the set of leaves corresponding to $x$ and $y$ in $\mathcal{T}$ (recall that the leaves of $\mathcal{I}$ can be super-vertices). The subtree induced by $\text{LCA}_{\mathcal{I}}(x, y)$ contains the *exactly* the same set of vertices as induced by $\text{LCA}_{\mathcal{T}}(X \cup Y)$.

The difference between the weak and strong consistency is that in weakly consistent partial trees, the "contraction" of vertices can happen in any consecutive region of the original tree $\mathcal{T}$. An illustration of the weakly consistent partial tree (w.r.t. $\mathcal{T}$) can be found in Figure 2.

Our goal is to use oracle $\mathcal{O}$, and vertex set $V$ to construct a partial HC tree $\mathcal{I}^*$ that is consistent with the optimal HC tree $\mathcal{T}^*$.

## 3.2. Main results of partial HC trees

We now give our main results for the strong and weak partial trees, respectively. In particular, our results include

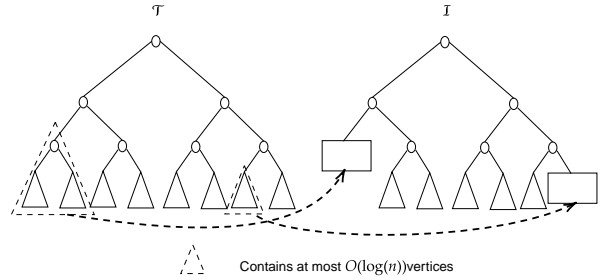

*Figure 1.* An illustration of the strongly consistent partial HC trees as defined in Definition 8. The boxes indicate super-vertices whose clustering is *unknown* in the partial HC tree.

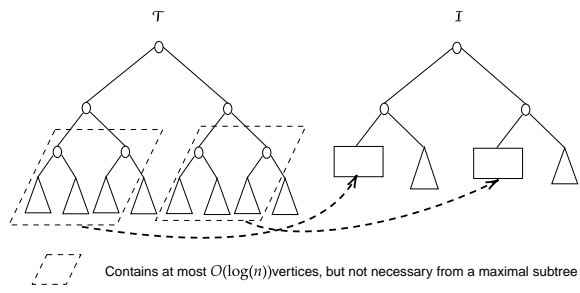

*Figure 2.* An illustration of the weakly consistent partial HC trees as defined in Definition 9. The boxes indicate super-vertices whose clustering is *unknown* in the partial HC tree.

- An algorithm that, with high probability, constructs a partial tree strongly consistent with the optimal tree $\mathcal{T}^*$ in $O(n^3 \log n)$ time and $O(n^3)$ queries to the splitting oracle. Furthermore, the algorithm uses only $\tilde{O}(n)$ space; and

- An algorithm that, with high probability, constructs a partial tree weakly consistent with the optimal tree $\mathcal{T}^*$ in $\tilde{O}(n^2)$ time and queries to the splitting oracle. Furthermore, the algorithm can be implemented in the PRAM model with $\tilde{O}(n^2)$ work and polylog $n$ depth.

**Result for strongly consistent partial HC trees.** Our main theorem to construct strongly consistent partial HC trees is as follows.

**Theorem 10.** *There exists an algorithm that given a splitting oracle $\mathcal{O}$ of a weighted undirected graph $G = (V, E, w)$, with high probability, in $O(n^3 \log n)$ time and $O(n^3)$ queries computes a partial hierarchical clustering tree $\mathcal{I}$ that is strongly consistent with the optimal hierarchical clustering tree $\mathcal{T}^*$. Furthermore, the algorithm has the following properties.*

   *i). The runtime of the algorithm is deterministic, and*

*the high probability randomness is over the correctness guarantee.*

*ii). The algorithm can be implemented in $O(n \log n)$ space.*

While the algorithm outlined in Theorem 10 necessitates $O(n^3)$ queries to the oracle $\mathcal{O}$, rendering it less efficient, the overall running time of $\tilde{O}(n^3)$ is tolerable, especially considering the hardness of HC.

**Result for weakly consistent partial HC trees.** We now show our algorithmic results for the weakly consistent partial HC trees, which enjoy much better efficiency in both the running time and the number of oracle queries.

**Theorem 11.** *There exists an algorithm that given a splitting oracle $\mathcal{O}$ of a weighted undirected graph $G = (V, E, w)$, with high probability, in $O(n^2 \cdot \text{polylog } n)$ time and $O(n^2)$ queries computes a partial hierarchical clustering tree $\mathcal{I}$ that is weakly consistent with the optimal hierarchical clustering tree $\mathcal{T}^*$.*

We note that since the number of longest dependent calls for our weak partial tree is at most $O(\log^3 n)$, Theorem 11 implies a PRAM algorithm with $O(n^2 \cdot \text{polylog } n)$ work and $O(\log^3 n)$ depth. The formal statement is as follows.

**Corollary 12.** *There exists a PRAM algorithm that given a graph $G = (V, E)$ and a splitting oracle $\mathcal{O}$, with high probability, in $O(n^2 \cdot \text{polylog } n)$ work and $O(\log^3 n)$ depth computes a partial hierarchical clustering tree $\mathcal{I}$ that is weakly consistent with the optimal HC tree $\mathcal{T}^*$.*

We suspect that by modifying some subroutines of the algorithm in Theorem 11, we could possibly bring the number of time and queries to $\tilde{O}(n)$. The study of a *sublinear-time* algorithm is an interesting future problem.

**A (very) high-level technical summary of the partial trees.** As we will discuss in Appendix B, simply following the oracle advice does *not* lead to any valid outputs. The construction of the partial trees requires a careful aggregation of the split-away information to recover subtrees with sizes $\Omega(\log n)$. One observation here is that if we are lucky to select a vertex $u$ from the *smaller* side of the first binary partition, we could determine the correct "side" for each vertex $v \in V$ in the first partition of the optimal tree with high probability. We could then recursively recover the partition in the optimal tree $\mathcal{T}^*$. However, getting a vertex from the smaller side of the first partition is not easy, and we would need to test multiple vertices and use the *size* of the resulting set as the indicator for correctness. This includes careful design and analysis of different algorithmic subroutines, which is the main technical challenge. A detailed technical overview could be find in Appendix B.

We present the HC algorithms using the results for partial HC trees for the rest of the main paper. We defer the detailed analysis of the partial trees to Appendices F to H.

## 4. Polynomial Time Algorithms for Dasgupta's Hierarchical Clustering Objective

We introduce our polynomial time algorithms for Dasgupta's HC objective in this section. These results include an $O(1)$-approximation algorithm in polynomial time (albeit some large constant on the exponent) and an $O(\sqrt{\log \log n})$-approximation algorithm in $\tilde{O}(n^3)$ time. Our algorithms crucially rely on the strongly consistent partial tree in Theorem 10.

### 4.1. A Polynomial-time Algorithm for $O(1)$-approximation on Dasgupta's HC Objective

We now introduce our algorithm that finds an HC tree with $O(1)$-approximation to Dasgupta's objective in polynomial time. One can find the formal statement of the result in Theorem 1. The algorithm of Theorem 1 is as Algorithm 1.

---

**Algorithm 1** A polynomial-time algorithm for the Dasgupta's HC objective

---

**Input:** Input graph $G = (V, E, w)$; Splitting oracle $\mathcal{O}$
**Output:** A hierarchical clustering tree $\mathcal{T}$
Run the strong partial tree approximation algorithm in Theorem 10 to obtain partial tree $\mathcal{I}$
**for** *each super-vertex in $\mathcal{I}$* **do**
    On input vertex set $S$, *exhaustively search* the sparsest cut $(A, B)$ on the induced subgraph $G[S]$
    Partition the vertices as $S \rightarrow (A, B)$, and recurse on $G[A]$ and $G[B]$
**end**

---

We now prove the efficiency and the approximation guarantees for Dasgupta's objective. The following lemma provides the efficiency for Algorithm 1.

**Lemma 4.1.** *Algorithm 1 runs (deterministically) in $O(n^{50002})$ time and uses $O(n^3)$ queries.*

*Proof.* By Theorem 10, the first step of the algorithm that computes the strong partial tree takes $\tilde{O}(n^3)$ time and $O(n^3)$ queries. Note that we only take queries in this step.

For the second step, when the input size is $s$, an exhaustive search on the sparsest cut takes $O(2^s)$ time. As such, let $X$ be the set of induced vertices for a single super-vertex of $\mathcal{I}$, since we have $|X| \leq 50000 \log n$, it only takes $O(n^{50000})$ time to find the sparsest cut. Similarly, we can show that in each recursive call, the runtime is at most $O(n^{50000})$. By Fact A.1, there are at most $O(\log n)$ nodes in a binary tree with $O(\log n)$ leaves. As such, the recursive sparsest cut for a single super-vertex in $\mathcal{I}$ takes $O(n^{50000} \cdot \log n)$ time. There

are at most $O(n)$ super-vertices in the tree; therefore, the total runtime of the second step takes $O(n^{50000} \cdot \log n \cdot n) = O(n^{50002})$ time.

Combining the efficiency of the two steps gives us the desired efficiency bound. □

The main lemma for the approximation guarantees of Algorithm 1 is as follows.

**Lemma 4.2.** *Conditioning on the high probability guarantees of Theorem 10, Algorithm 1 outputs an HC tree $\mathcal{T}$ that achieves $O(1)$-approximation to the Dasgupta's objective.*

*Proof.* For any partial tree $\mathcal{I}$, we first partition the edges into $E_{\text{cross}}$ and $E_{\text{same}}$ based on whether the edge $(u, v) \in E$ crosses different partial trees, i.e.,

1. $(u, v) \in E_{\text{cross}}$ iff $u \in X$ and $v \in Y$ for some super-vertices $X \neq Y$ in $\mathcal{I}$.

2. $(u, v) \in E_{\text{same}}$ iff $u, v \in X$ for some super-vertex $X$ in $\mathcal{I}$.

Since $E = E_{\text{cross}} \cup E_{\text{same}}$, by using Observation 1, we can show that $\mathsf{OPT}^{\text{Das}} = \mathsf{cost}_G(\mathcal{T}^*) = \mathsf{cost}_G(\mathcal{T}^*, E_1) + \mathsf{cost}_G(\mathcal{T}^*, E_2)$. We now analyze the costs w.r.t. to $E_1$ and $E_2$, respectively.

1. For $E_{\text{cross}}$, we argue that $\mathsf{cost}_G(\mathcal{T}, E_{\text{cross}}) = \mathsf{cost}_G(\mathcal{T}^*, E_{\text{cross}})$. To see this, note that if $u$ and $v$ are of different super-vertices, by the definition of partial HC trees that are *strongly consistent* with the optimal tree $\mathcal{T}^*$, there is

   $$\texttt{leaves}_{\mathcal{T}}[\texttt{LCA}_{\mathcal{T}}(u, v)] = \texttt{leaves}_{\mathcal{T}^*}[\texttt{LCA}_{\mathcal{T}^*}(u, v)].$$

   As such, we have $\mathsf{cost}_G(\mathcal{T}, E_{\text{cross}}) = \mathsf{cost}_G(\mathcal{T}^*, E_{\text{cross}})$ by the definition of the cost function.

2. For $E_{\text{same}}$, we argue that $\mathsf{cost}_G(\mathcal{T}, E_{\text{same}}) \leq O(1) \cdot \mathsf{cost}_G(\mathcal{T}^*, E_{\text{same}})$. Formally, for each super-vertex $X$, we can use Proposition 17 on $G[X]$ to argue that the $\mathsf{cost}_G(\mathcal{T}, E_{\text{same}}[X]) \leq O(1) \cdot \mathsf{cost}_G(\mathcal{T}^*, E_{\text{same}}[X])$, where $E_{\text{same}}[X]$ stands for the set of edges in $E_{\text{same}}$ with *both* endpoints in $X$. Therefore, we can apply this calculation to every super-vertex to get the desired approximation factor.

We now use Observation 1 again on $E_{\text{cross}}$ and $E_{\text{same}}$ to bound that

$$\begin{aligned} \mathsf{cost}_G(\mathcal{T}) &= \mathsf{cost}_G(\mathcal{T}, E_{\text{cross}}) + \mathsf{cost}_G(\mathcal{T}, E_{\text{same}}) \\ &\leq \mathsf{cost}_G(\mathcal{T}^*, E_{\text{cross}}) + O(1) \cdot \mathsf{cost}_G(\mathcal{T}^*, E_{\text{same}}) \\ &\leq O(1) \cdot (\mathsf{cost}_G(\mathcal{T}^*, E_{\text{cross}}) + \mathsf{cost}_G(\mathcal{T}^*, E_{\text{same}})) \\ &= O(1) \cdot \mathsf{cost}_G(\mathcal{T}^*) = O(1) \cdot \mathsf{OPT}^{\text{Das}}, \end{aligned}$$

as desired. □

Combining Theorem 10, Lemma 4.2, and Lemma 4.1 leads to the proof of Theorem 1 (see Appendix C for a more formal version).

## 4.2. An $\tilde{O}(n^3)$ Time Algorithm for $O(\sqrt{\log \log n})$-approximation on Dasgupta's HC Objective

One drawback of the algorithm we have in Section 4.1 is that the efficiency is "theoretical only" – after all, a runtime of $O(n^{50002})$ is nowhere near being practical. Observe that the subroutine that leads to the very large exponent is the exhaustive search of the *optimal* sparsest cut. Therefore, we can hope to use some more efficient approximation for sparsest cuts while not sacrificing too much on the approximation guarantee. This intuition leads us to Algorithm 2.

---

**Algorithm 2** An $\tilde{O}(n^3)$ time algorithm for the Dasgupta's HC objective

---

**Input:** Input graph $G = (V, E, w)$; Splitting oracle $\mathcal{O}$
**Output:** A hierarchical clustering tree $\mathcal{T}$
Run the strong partial tree approximation algorithm in Theorem 10 to obtain partial tree $\mathcal{I}$
**for** *each super-vertex in $\mathcal{I}$* **do**

> On input vertex set $S$, find an $O(\sqrt{\log |S|})$-approximation of the sparsest cut $(A, B)$ on the induced subgraph $G[S]$ using the algorithm of Proposition 18
> Partition the vertices as $S \to (A, B)$, and recurse on $G[A]$ and $G[B]$

**end**

---

The main lemmas for the efficiency and approximation guarantees for Algorithm 2 are as follows.

**Lemma 4.3.** *With high probability, Algorithm 2 runs in $O(n^3 \log n)$ time and uses $O(n^3)$ queries.*

**Lemma 4.4.** *Conditioning on the high probability guarantees of Theorem 10, Algorithm 2 outputs an HC tree $\mathcal{T}$ that achieves $O(\sqrt{\log \log n})$-approximation to the Dasgupta's objective.*

The proof of Lemma 4.3 and Lemma 4.4 could be found in Appendix C.

## 5. Near-linear Time Algorithms for Moseley-Wang Hierarchical Clustering Objective

We introduce our algorithm for the Moseley-Wang HC objective in this section. The algorithm is based on the weakly consistent trees as in Theorem 11, which could be implemented in near-linear time. The algorithm is described as Algorithm 3.

---

**Algorithm 3** A near-linear time algorithm for the Moseley-Wang HC objective

---

**Input:** Input graph $G = (V, E, w)$; Splitting oracle $\mathcal{O}$
**Output:** A hierarchical clustering tree $\mathcal{T}$
Run the weak partial tree approximation algorithm in Theorem 11 to obtain partial tree $\mathcal{I}$
    **for** *each super-vertex in $\mathcal{I}$* **do**
      | partition the leaves *arbitrarily* to obtain an HC tree $\mathcal{T}$
**end**

---

Since each super-vertex contains $O(\log n)$ vertices, we can always partition the super-vertices in polylog $n$ time[2]. There are at most $O(n/\log n)$ such super-vertices, and the total runtime overhead is at most $O(n \cdot \text{polylog}\, n)$.

Our analysis for Algorithm 3 is more involved compared to the results in Section 4 – it requires careful handling of the contributions of the 'less significant edges' to the Moseley-Wang objective. In particular, we could prove the following structural result. Let $\mathcal{I}$ be any partial HC tree that is weakly consistent with the optimal tree $\mathcal{T}^*$, we show that the set of edges $(u, v)$ such that

a). has at most $O(\log^2 n)$ non-leaves in $\mathcal{T}^*$; and

b). let $X$ and $Y$ be corresponding super-vertices that contain $u$ and $v$ in $\mathcal{I}$; there is leaves$_{\mathcal{T}^*}[\text{LCA}_{\mathcal{T}^*}(X)] \cap$ leaves$_{\mathcal{T}^*}[\text{LCA}_{\mathcal{T}^*}(Y)] \neq \emptyset$.

can contribute to at most an $o(1)$ fraction of the optimal cost. To the best of our knowledge, the structural result was not known before.

## 6. Learning-augmented Sublinear Algorithms for Hierarchical Clustering

In this section, we explore *sublinear* algorithms for hierarchical clustering with the splitting oracle. Among these is a *semi-streaming* algorithm, capable of computing an $O(1)$ approximation of Dasgupta's HC objective within *polynomial time*. Additionally, we introduce a PRAM algorithm that achieves a $(1 - o(1))$ approximation of the Moseley-Wang objective, utilizing $\tilde{O}(n^2)$ work and polylog $n$ depth. From a technical standpoint, these algorithms represent straightforward extensions of the results outlined in Theorems 1 to 3. Despite their simplicity, these algorithms demonstrate the advantages of the splitting oracle in modern sublinear computation models. Specifically, we compare our sublinear algorithms with previous results as follows:

1. In the streaming setting, (Assadi et al., 2022; Agarwal et al., 2022) designed single-pass streaming al-

gorithms that achieve $\tilde{O}(n)$ memory usage and $O(1)$-approximation to Dasgupta's objective, albeit in *exponential time*. By improving the time efficiency to polynomial time, our streaming result echoes a similar narrative in the offline setting, demonstrating significantly more efficient constructions with the splitting oracle.

2. For the parallel setting, (Agarwal et al., 2024) (cf. (Agarwal et al., 2022)) provided parallel algorithms (in the PRAM and the similar MPC settings, see Appendix A.3 for details of these models) for Dasgupta's objective with $\tilde{O}(n^2)$ work and polylog $n$ depth that achieve polylog $n$ approximation. Since the objectives are different, their result is not directly comparable to ours; however, the conceptual message here is still that the splitting oracle is able to significantly improve the approximation guarantee and the efficiency.

Due to space limits, we only give the algorithms as in Algorithm 4 and Algorithm 5, and defer their proofs to Appendix E.

---

**Algorithm 4** A polynomial-time single-pass semi-streaming algorithm for the Dasgupta's HC objective

---

**Input:** Input graph $G = (V, E, w)$; Splitting oracle $\mathcal{O}$
**Output:** A hierarchical clustering tree $\mathcal{T}$
**Before** the start of the stream, run the strong partial tree approximation algorithm in Theorem 10 to obtain partial tree $\mathcal{I}$
**for** *each edge $(u, v)$ with during the insertion/deletion stream* **do**
    | If $u, v \in X$ for any super-vertex $X$ in $\mathcal{I}$, update $(u, v)$ with the same insertion/deletion update
    | Otherwise, ignore the edge
**end**
**for** *each super-vertex in $\mathcal{I}$ after the stream, run recursive sparsest cut as follows* **do**
    | On input vertex set $S$, *exhaustively search* the sparsest cut $(A, B)$ on the induced subgraph $G[S]$.
    | Partition the vertices as $S \to (A, B)$, and recurse on $G[A]$ and $G[B]$.
**end**

---

**Algorithm 5** A near-linear work, poly-logarithmic depth PRAM algorithm for the Moseley-Wang HC objective

---

**Input:** Input graph $G = (V, E, w)$; Splitting oracle $\mathcal{O}$
**Output:** A hierarchical clustering tree $\mathcal{T}$
Run the PRAM weak partial tree approximation algorithm in Corollary 12 to obtain partial tree $\mathcal{I}$
    **for** *each super-vertex in $\mathcal{I}$* **do**
      | Partition the leaves *arbitrarily* to obtain an HC tree $\mathcal{T}$
**end**

---

[2]Arbitrary balanced partitions requires $O(\log n \cdot \log \log n)$ time

## Acknowledgements

We thank anonymous ICML reviewers for the helpful comments and suggestions. Vladimir Braverman is supported in part by the Naval Research (ONR) grant N00014-23-1-2737 and NSF CNS-2333887 award. Samson Zhou is supported in part by NSF CCF-2335411. The work was conducted in part while Samson Zhou was visiting the Simons Institute for the Theory of Computing as part of the Sublinear Algorithms program.

## Impact Statement

This paper presents work that advances the Machine Learning and Algorithm Design fields. Due to its theoretical nature, we do not see any *immediate* societal impacts, although there are many potential societal consequences of our work for downstream applications.

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

# A. Additional Technical Preliminaries

## A.1. Concentration inequalities

We now present the standard concentration inequalities used in our proofs. We start from the following standard variant of Chernoff-Hoeffding bound.

**Proposition 13** (Chernoff-Hoeffding bound). *Let $X_1, \ldots, X_n$ be $n$ independent random variables with support in $[0, 1]$. Define $X := \sum_{i=1}^{n} X_i$. Then, for every $\delta \in (0, 1]$, there is*

$$\Pr\left(|X - \mathbb{E}[X]| > \delta \cdot \mathbb{E}[X]\right) \leq 2 \cdot \exp\left(-\frac{\delta^2 \, \mathbb{E}[X]}{3}\right).$$

*Furthermore, for every $\delta > 0$, there is*

$$\Pr\left(|X - \mathbb{E}[X]| > \delta \cdot \mathbb{E}[X]\right) \leq 2 \cdot \exp\left(-\frac{\delta^2 \, \mathbb{E}[X]}{2 + \delta}\right).$$

## A.2. Standard results for trees

**Fact A.1.** *Any binary tree $\mathcal{T}$ with $n$ leaves contains at most $n$ internal nodes.*

*Proof.* Consider the collection of the internal nodes that are the parents of the leaves. Since the tree is binary, there are *at most* $n/2$ such nodes. Contract all these internal nodes with the leaves, and we can obtain a new tree $\mathcal{T}'$ such that the number of leaves is $n/2$. As such, we can again count an upper bound for the number of the parents for the leaves in $\mathcal{T}'$ as $n/4$. We can continue this until we only have the root, and the number of internal nodes is at most

$$\sum_{i=1}^{n} \frac{n}{2^i} \leq \sum_{i=\infty}^{n} \frac{n}{2^i} = n,$$

as desired. □

## A.3. The PRAM and the Massively Parallel Computation (MPC) Models

We briefly introduce the parallel models we investigate for the Moseley-Wang objective. In particular, we investigated the classical PRAM model and the Massively Parallel Computation model in our work.

**The PRAM model.** The Parallel Random Access Machine (PRAM) model is a widely used theoretical framework in parallel computing. It provides a simplified abstraction of a parallel computer system, where multiple processors work simultaneously to solve a computational problem. In the PRAM model, each processor has direct access to a common memory space (RAM), and communication between processors and the RAM is instantaneous ("parallel" RAM). Processors can read from and write to any memory location in parallel, hence the term "Random Access".

In the PRAM model, there are usually two objectives for algorithm designers to optimize: the total *work*, defined as the total number of elementary operations, and the *depth*, defined as the length of the longest dependent call in the algorithm. In the theoretical abstract version of PRAM, we do *not* care about the number of processors we use in the algorithm.

**The MPC model.** The Massively Parallel Computation (MPC) model is a theoretical framework used to analyze algorithms designed for modern parallel computing architectures, e.g., the MapReduce framework. In this model, communications are conducted in *synchronized rounds*, and a machine can communicate with any other machine in a round. Furthermore, each machine can do unlimited local computation between the rounds. Unlike traditional CONGEST models, the communication here is limited only by the *memory size* – a size $s$ machine cannot send or receive more than $s$ bits of information.

For graph problems, suppose each machine has size $s$, we typically use $\tilde{O}(n^2/s)$ machines, where $O(n^2)$ is the worst-case input size (alternatively, one can also target the more instance-optimal $\tilde{O}(m/s)$ machines).

Our goal in the MPC model is to minimize two objectives: $i)$. the memory size $s$ of each machine; and $ii)$. the number of parallel rounds. In particular, if our algorithm works with $s = O(n^\delta)$ memory for any $\delta \in (0, 1)$, we call the algorithm *fully scalable* in the MPC model. Typically, the best MPC algorithms would ask for fully scalable memory and polylog $n$ rounds.

**A reduction between the PRAM and the MPC algorithms.** The PRAM and the MPC models share a great deal of similarities. And indeed, the following reduction is known.

**Proposition 14.** *Suppose there exists a PRAM algorithm that computes a function $f$ with $w(n)$ work and $d(n)$ depth, where $n$ is the input size of $f$. Then, there exists a fully scalable MPC algorithm that computes $f$ with $O(w(n))$ total memory and $O(d)$ rounds. The memory per machine can be made $O(n^\delta)$ for any $\delta \in (0,1)$, and the number of machines is $O(\frac{W(n)}{n^\delta} \cdot \text{polylog } n)$.*

### A.4. Existing Techniques for Dasgupta's Objective

We discuss some known techniques for Dasgupta's minimization HC objective in this section. We will use these techniques in our hierarchical clustering algorithms for Dasgupta's objective.

OPTIMAL HIERARCHICAL CLUSTERING TREES

We first give an observation that characterizes the "composability" of HC costs with respect to the edges under Dasgupta's objective.

**Observation 1** ((Dasgupta, 2016)). *Let $G$ be any graph, and let $E_1$ and $E_2$ be two disjoint subsets of edges in $G$. For any HC tree $\mathcal{T}$, let $\text{cost}_G(\mathcal{T}, E_1)$ and $\text{cost}_G(\mathcal{T}, E_2)$ be the HC costs induced by edges in $E_1$ and $E_2$, respectively. Then,*

$$\text{cost}_G(\mathcal{T}) = \text{cost}_G(\mathcal{T}, E_1) + \text{cost}_G(\mathcal{T}, E_2).$$

Observation 1 shows that to bound the total cost of the HC tree under Dasgupta's objective, it suffices to bound the edges split by the internal nodes.

APPROXIMATE HC TREES WITH RECURSIVE BALANCED MIN-CUTS AND SPARSEST CUTS

Dasgupta's work proved that finding the optimal trees for the hierarchical clustering function is NP-hard (Dasgupta, 2016). Consequently, significant attention has been directed towards developing *approximation algorithms* for efficient hierarchical clustering on the graph. A well-known approach involves obtaining an approximation of the optimal hierarchical clustering by iteratively employing *sparsest cuts* on the graph, e.g., (Dasgupta, 2016; Deng et al., 2025). Formally, we can define the sparsest cuts and the HC trees created by recursively applying the sparsest cuts on the induced subgraphs as in Definition 15 and Definition 16.

**Definition 15** (**Sparsest Cuts**). For any parameter $\beta$ such that $0 < \beta < 1$, we say that a cut $(A^*, B^*)$ is a *sparsest cut* if its *sparsity (edge expansion)* is minimized, i.e.

$$\frac{w(A^*, B^*)}{\min\{|A^*|, |B^*|\}} \leq \frac{w(A, B)}{\min\{|A|, |B|\}}$$

for any cut $(A, B)$ of $G$.

**Definition 16** (**Recursive Sparsest Cut Procedure**). We say an HC tree $\mathcal{T}$ is obtained by the *recursive sparsest procedure* on $G$ if for each non-leaf node $z$ of $\mathcal{T}$, the **cut**$(\mathcal{T}[z])$ is obtained by a (possibly approximate) sparsest cut $(A, B)$ on the subgraph induced by $\mathcal{T}[z]$. We call an HC tree obtained by recursively applying (approximate) sparsest cuts on induced subgraphs as a recursive sparsest cut HC tree.

Previous work (see, e.g. (Charikar & Chatziafratis, 2017b; Assadi et al., 2022)) proved that if one applies the procedure in Definition 16, we can get an $O(1)$ approximation of the optimal HC tree.

**Proposition 17** ((Charikar & Chatziafratis, 2017b; Assadi et al., 2022)). *For any graph $G = (V, E, w)$, let $\mathcal{T}_{sparse}$ be an HC tree obtained by the recursive sparsest cut procedure in Definition 16, there is*

$$\text{cost}_G(\mathcal{T}_{sparse}) \leq O(1) \cdot \text{OPT}(G).$$

Note that finding the exact sparsest cut for Proposition 17 is NP-hard. The first way to circumvent this issue is to use approximation algorithms, especially for the best-known $O(\sqrt{\log n})$ approximation for sparsest cut in polynomial time (Arora et al., 2004). The formal guarantee is as follows.

**Proposition 18** ((Arora et al., 2004))**.** *There exists a randomized algorithm that given a graph $G = (V, E, w)$, with high probability, in $\tilde{O}(|V|^2)$ time finds a partition $A \cup B = V$ such that*

$$\frac{w(A, B)}{\min\{|A|, |B|\}} \leq O(\sqrt{\log |V|}) \cdot \frac{w(A^*, B^*)}{\min\{|A^*|, |B^*|\}},$$

*where $(A^*, B^*)$ is the sparsest cut of $G$.*

We cannot immediately massage Proposition 18 with Proposition 17 since Proposition 17 does *not* state what will happen for *approximate* sparsest cuts. Fortunately, by the results in (Charikar & Chatziafratis, 2017b; Assadi et al., 2022), we can indeed obtain an $O(\alpha)$-approximation algorithm for $\mathsf{OPT}(G)$ by recursively applying the $\alpha$-approximate sparsest cut.

**Proposition 19** ((Charikar & Chatziafratis, 2017b; Assadi et al., 2022))**.** *Let $\mathcal{T}$ be a recursive sparsest cut tree obtained by recursively applying $\alpha$-approximation sparsest cuts on the induced subgraphs (as in Definition 16). Then, we have*

$$\mathsf{cost}_G(\mathcal{T}) \leq O(\alpha) \cdot \mathsf{OPT}(G).$$

There is another way to deal with the NP-hardness issue of the sparsest cut. If we can reduce the *input size*, we can possibly obtain the *exact optimal* cuts on induced subgraphs of size $O(\log n)$. This strategy allows us to leverage our strong partial tree whose 'unknown' clustering is only restricted to the induced subgraphs with $O(\log n)$ size.

## B. Technical Overview

In this section, we give a high-level overview of our techniques. We also provide intuition on our algorithmic design choices, including a number of potential pitfalls, as well as a number of natural other approaches and why they do not work.

### B.1. Why not simply follow the oracle (or other related strategies)?

At first glance, one might wonder whether the splitting oracle trivializes the problem. A natural question is whether it is possible to simply follow the oracle to recover the optimal tree $\mathcal{T}^*$. Since the oracle only returns the relative information among a *triplet* of vertices $(u, v, w)$, it is not immediately clear how to translate the answers from the oracle to a partition of vertices. After taking a closer look at the problem, we could observe issues with a handful of straightforward approaches.

The first natural approach is to pretend the oracle is always correct and construct a tree from the "splitting-away" information between the triplets. Unfortunately, due to the error probability and adversarial answers, there may *not* exist an underlying tree consistent with the answers to the queries. As such, it is unclear how the algorithm could produce a definitive answer.

The second approach we could try is to frame the problem as a phylogenetic reconstruction problem, e.g., take all the "splitting away" for triplets as constraints, and try to construct an HC tree that satisfies as many constraints as possible. However, such an approach has two issues: $i$). by a recent result of (Chatziafratis & Makarychev, 2023), the phylogenetic reconstruction problem is itself UG-hard; and $ii$). the HC tree we constructed may prioritize a small number of *wrong* answers from the oracle that happen to induce very large additive error.

A more involved idea is to "aggregate" the oracle answers to construct the HC tree's partitions. To this end, an algorithm to determine the partition of a vertex $v$ is to fix $u$ in the smaller subtree and look into the number of vertices $t \in V$ that split away from $(u, v)$. More concretely, consider the split of the tree on the root $V \to (S_1, S_2)$, and suppose we know a vertex $u$ that is on the smaller subtree of the root partition (this is a big "suppose" as we will see later). Then for any vertex $v$ that is in the same subtree of $u$, we can get many vertices $t \in V$ from $\mathcal{O}$ with the answer "$t$ split away from $(u, v)$". On the other hand, for a vertex $v$ that is in the opposite subtree of $u$, only a few vertices $t$ from $\mathcal{O}$ would answer "$t \in V$ split away from $(u, v)$". The gap is large enough to apply concentration inequalities and find a separation between the cases. As such, we can recursively apply the above procedure and produce the optimal tree $\mathcal{T}^*$.

Unfortunately, the above idea only works for the idealized case where we indeed know a vertex $u$ from the smaller part of the root partition. For the general case, the algorithm requires a surprising amount of new ideas and technical work. In particular, note that the aforementioned algorithm faces two major challenges: (1). as the partition goes deep down the HC tree, the sizes of the subtrees become too small for high-probability guarantees; and (2). it is not clear how to find a "good" vertex $u$ that induces the root cut. To elaborate on challenge (1), note that when the subtree induces $o(\log n)$ leaves, it is generally not possible for us to guarantee correctness for the subsequent partitions. As such, we must handle some form

of "ambiguity" when dealing with subtrees induced on vertex sets with smaller sizes. Our approach to this challenge is to forgo the guarantees inside each leaf with $o(\log n)$ vertices and work with the respective objective functions to show that the additive error is tolerable. In particular, we use the notion of *partial* hierarchical clustering trees that approximately capture the structure of the optimal HC tree $\mathcal{T}^*$ until the size of the induced vertex set becomes too small. In particular, we require specific structural properties of the costs of HC trees under Dasgupta's and the Moseley-Wang objectives. We provide more details about partial HC trees and how to use them to overcome challenge (1) in Appendix B.2.

Challenge (2) is even trickier and requires more care. Observe that in the example of root cut $V \to (S_1, S_2)$, if $u$ is in the *bigger* side of the partition, the argument may *not* work. However, since we only have access to the *triplet split* information, retrieving whether a vertex $u$ is on the smaller side of a particular tree split seems to be too much to ask. In particular, consider an example that the optimal tree $\mathcal{T}^*$ first makes two splits of small subtrees of size $n^{0.99}$, as illustrated in Figure 3. Here, if we use $u_2$ as the "baseline" vertex to perform the split, we can still get a valid partition. However, the structure of the obtained tree is very different from $\mathcal{T}^*$, and the additive error could be huge. Furthermore, since the actual tree $\mathcal{T}^*$ is hidden from us, it is not immediately clear how could we distinguish a partition obtained by using $u_1$ vs. $u_2$. The problem becomes even more intriguing when we want to obtain near-linear time efficiency. We will discuss the intuition and techniques to handle challenge (2) in Appendix B.3.

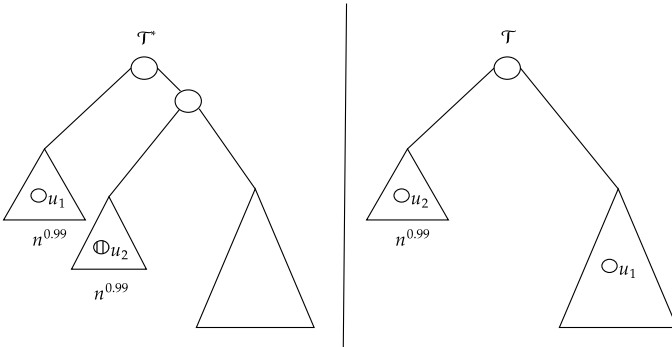

*Figure 3.* An illustration of the hard example that the straightforward majority voting does not work. Left: the optimal HC tree $\mathcal{T}^*$; Right: the outcome of the tree if we use $u_2$ as the baseline to perform the partition we discussed.

## B.2. Partial hierarchical clustering trees and HC

We now delve into more details about the definition of partial trees and how we use them to obtain low additive errors in Dasgupta's and the Moseley-Wang objectives.

**The definition of partial hierarchical clustering trees.** As discussed, a central element of our techniques is the notion of partial hierarchical clustering (HC) trees. The generic definition of the partial HC tree is similar to the normal HC tree, with the root representing the entire vertex set and the internal nodes representing the subsets of vertices. However, on the *leaf* level, we allow the leaves of partial HC trees to contain *multiple vertices*, up to $O(\log n)$ many vertices, and contract them into a single leaf, which we call "super-vertices". The partial HC tree is then allowed to be oblivious of the clustering *inside* each leaf node.

A partial HC tree is only useful if it can somehow capture the optimal tree $\mathcal{T}^*$. To this end, we introduce the *strongly consistent* and *weakly consistent* partial HC trees. Roughly speaking, a partial HC tree $\mathcal{I}$ is said to be strongly consistent with the optimal tree $\mathcal{T}^*$ if it follows *every partition* of the optimal tree in a top-down manner until the induced size of vertices is of size less than $O(\log n)$, in which case we simply collapse the leaf into a super-vertex. Note that in the strongly consistent partial HC trees, every super-vertex induces a maximal tree in $\mathcal{T}^*$ – here, a maximal tree means a tree where its induced vertices are exactly the leaves of their lowest common ancestor. By comparison, the *weakly consistent* partial HC tree allows vertices that do not necessarily form maximal trees to collapse into a single super-vertex. For instance, if there are $O(\log n)$ vertices in *multiple* subtrees in $\mathcal{T}^*$, and suppose the LCAs of these subtrees are close to the root and

form a consecutive segment in $\mathcal{T}^*$, the weakly consistent partial HC tree can still collapse *all* of the vertices into a single super-vertex. We provide illustrations of the weakly and strongly consistent partial trees in Figure 1 and Figure 2, and one can refer their formal definition to Definitions 8 and 9 in Section 3.

We will eventually show that, given the splitting oracle, we can efficiently construct both the strongly and the weakly consistent partial trees *regardless* of the input graph and the objective functions. However, for now, we first discuss *why* partial HC trees are useful for our HC objectives.

**Using partial HC trees for hierarchical clustering.** Intuitively, if we can obtain a partial HC tree that is consistent with the optimal tree $\mathcal{T}^*$, we can build an actual HC tree by "fixing" the clustering of the super-vertices locally to obtain a good approximation. Indeed, we note that if a partial HC tree is *strongly* consistent with $\mathcal{T}^*$, we can straightforwardly obtain approximation algorithms for both Dasgupta's and the Moseley-Wang objectives. For Dasgupta's objective, we can run the optimal or approximate *recursive sparsest cuts* for the subgraphs induced by the super-vertices. Note that since the subgraphs are only of size $O(\log n)$, we can even afford to find the exact optimal recursive sparsest cuts in *polynomial time*. The case for the Moseley-Wang objective is even easier: since the number of leaves outside each super-vertex is at least $n - O(\log n)$, any arbitrary partition of the super-vertices can still give us a $(1 - o(1))$ approximation.

Unfortunately, as we will see shortly, the strongly consistent partial tree can only be implemented in $\tilde{O}(n^3)$ time – a tolerable yet far-from-optimal efficiency. As such, for the Moseley-Wang objective, we further investigate the algorithm that only uses the *weakly consistent* partial HC trees, which we can build in near-linear time. In this case, for two vertices that are in the same super-vertex, we can replicate the argument for the strongly consistent partial HC trees again to get a $(1 - o(1))$ approximation. However, challenges arise when the two vertices $(u, v)$ are in different super-vertices of the partial HC tree. Here, the number of induced leaves can differ by an $O(\log n)$ additive factor, but the number of non-leaves induced by $(u, v)$ can be very small in $\mathcal{T}^*$ (say, $o(\log n)$). As such, the $O(\log n)$ difference on the *size* of non-leaves might lead to *infinity multiplicative gap* in the revenue, which makes controlling the overall approximation factor hard. To tackle this issue, we prove some new structural results for HC trees under the Moseley-Wang objective: we show for all the edges $(u, v)$ that only induce a very small number of non-leaves *and* are "far away" from each other in the optimal tree, the contribution of such edges to the optimal objective can only be an $o(1)$ fraction. As such, we can simply ignore the approximation guarantees on these edges and obtain an $(1 - o(1))$ approximation of the Moseley-Wang objective.

## B.3. The construction of the partial HC trees

We now come back to the efficient construction of the partial HC trees that are strongly and weakly consistent with the optimal HC tree. For simplicity, we slightly abuse the notation to let $V$ always denote the vertex set in the high-level discussion, even if we are talking about a subset of vertices[3].

### B.3.1. STRONGLY CONSISTENT PARTIAL HC TREES

We first discuss the case for the strongly consistent HC tree, which only requires top-down splits. As we have discussed before, for any fixed partition, since we do *not* have any information about which side a vertex $u$ is on, it is generally very hard for us to know which obtained partition is actually consistent with the split in $\mathcal{T}^*$. To address this challenge, introduce the *small-tree splitting order*, which, roughly speaking, is a thought process that recursively draws the smaller side of the subtree in $\mathcal{T}^*$. For instance, in the tree $\mathcal{T}^*$ prescribed in Figure 3, the first $n^{0.99}$ vertices form the first small tree $V_1^{\text{small}}$, the second $n^{0.99}$ vertices form the second small tree $V_2^{\text{small}}$, and so on.

We shall show that if $u$ is among the first few small trees in the small-tree splitting order, we can recover the set of vertices as the *sibling* of the small tree[4]. For example, if we select $u_2$ in Figure 3, we can recover the subtree on the right but not the subtree that contains $u_1$. Our strategy is as follows: for a fixed vertex $u$ and a vertex $v$ whose split is to be determined, in addition to testing how many vertices $t \in V$ such that $t$ splits away from $(u, v)$, we also test the number of vertices $t \in V$ such that $v$ *splits away from* $(u, t)$. To see why this additional test helps, let us again look at the example in Figure 3. With the additional subroutine, if we use $u_2$ as the fixed vertex, the vertices in $V_1^{\text{small}}$ will split away from many $(u_2, t)$ pairs. On the other hand, for every vertex $v$ on the sibling subtree of $V_2^{\text{small}}$, it splits away from $(u, t)$ only if $t \in V_2^{\text{small}}$, which creates a clear signal. By careful handling of cases, we could argue that the algorithm works for general cases as long as $u$ belongs to an "early enough" small tree.

---

[3]In the formal analysis of Appendices G and H, we use $\widetilde{V}$ as the set of vertices of the current recursion level.

[4]Since "sibling" is a generic word, we call this set "counterpart" in our formal description in Appendices G and H to avoid confusion.

The above strategy provides a new way to identify a "good" $u$: it suffices to only look at the *size* of the set of vertices we recover. In particular, if $u$ is among the *root* cut, it surely induces the largest size on the set of the recovered vertices. As such, a simple exhaustive search can find such a vertex $u$ and the corresponding set $T$. Since there are $n$ vertices to be tested, and each test requires $O(n^2)$ time, the total time for each partition is at most $O(n^3)$. We can then recursively run this procedure, which will lead to a partial tree that is *strongly consistent* in $O(n^4)$ time. Furthermore, using a simple sampling trick, we could reduce the time for each test to $O(n \log n)$ time and queries, which brings the total number of time and queries to $\tilde{O}(n^3)$. Furthermore, since we only need to maintain counters for each vertex, the entire algorithm can be implemented in $O(n \log n)$ space.

### B.3.2. WEAKLY CONSISTENT PARTIAL HC TREES

The exhaustive search subroutine in the above idea inevitably leads to $\tilde{\Theta}(n^3)$ time and queries on the splitting oracle. This gives us a new, and perhaps more intriguing challenge: if we only want to get partial trees that are *weakly consistent* with the optimal tree, can we improve the efficiency? Note that if we target a near-linear running time, we cannot always hope to get a $u$ from the smaller side of the *root* partition. For a concrete example, let us look at the tree $\mathcal{T}^*$ in Figure 3 again. Here, before we "hit" a vertex in the first small tree of size $n^{0.99}$, we will *not* be able to produce a root cut. However, by the size of the first small tree, we will need to test at least $n^{0.01}$ vertices if we sample vertices uniformly at random. The overhead can be further enlarged: suppose the root cut splits the vertices into $n-1$ vertices and a single vertex, and suppose this process continues for $n^{0.99}$ levels; then, it is entirely unclear how to avoid the $n^{0.99}$ overhead.

**The vertical split idea.** The above hard instance inspires us to resort to "vertical" splits of the tree – that is, instead of finding a vertex $u$ on the split of the root, we use a vertex $u$ that is "sufficiently early" in the small-tree split order. To elaborate, we can efficiently find a vertex $u$ that is among the *union* of smaller subtrees that collectively induced at least $n/\text{polylog}\, n$ leaves. In this way, we can still recover a *maximal subtree* whose induced leaves $T$ is of size at least $(n - n/\text{polylog}\, n)$ – a size reduction that is significant enough for the entire algorithm to converge in polylog $n$ iterations. Finally, our algorithm will guarantee that $V$ is a composable set from a *single* maximal tree, which implies that $T$ and $V \setminus T$ are composable sets, which allow us to recurse on both sides.

**The use of "horizon sets".** There is yet another subtle issue in the above idea: we have to ensure both parts of the split always maintain the *weak consistency* property. Specifically, it is crucial to maintain super-vertices with an out-degree of at most 2, where each is linked to at most one parent node in $\mathcal{T}^*$ and one sibling node in $\mathcal{T}^*$. Within our vertical split concept, as $T$ invariably forms a single composable set, achieving this is straightforward. However, in the residual part of the split algorithm—here, $V \setminus T$—sustaining weak consistency becomes notably more complex. There can be two cases for such a guarantee to hold: either $a$). $V \setminus T$ itself is a single composable set, which happens when $V$ is a split on the *root* vertex, or $b$). $V \setminus T$ has some "orphaned" vertices – the sibling vertices of the subtree induced by $T$ in $V$ (see Definition 28 for the formal definition).

Our approach to handle both the $a$) and $b$) cases is to use a semi-invariant *horizon set* $V_{\mathrm{H}} \supseteq V$. The idea here is that instead of finding $T$ on $V$, we find it on $V_{\mathrm{H}}$, which is roughly defined as the set of vertices for us to find the partition $T$ on *before* a split on the root node. In particular, suppose the set of orphaned vertices is of relatively small size in $V$. We can always find a vertex $u \in V$ such that $u$ is split *earlier* than the orphaned vertex set in the small-tree split order of $V_{\mathrm{H}}$. Therefore, we can make sure that the set $T$ to be found in the new iteration will include the orphaned vertices in $V$. In this way, we always keep at most *one* edge connecting to the sibling of the orphaned vertices in the *current iteration*. An illustration of the role of the horizon set can be shown as Figure 9 in Appendix H.

**The candidate vertex and root test.** The above analysis assumed the size of orphaned vertices is relatively small in $V$. However, what if the size of orphaned vertices becomes large in $V$? In this scenario, one of two sub-cases must happen: either we run into a root split, or we have a large set of vertices which is not on the root of $V_{\mathrm{H}}$, but occupies a large fraction among the remaining $V$. (Note that in case of a root split, the entire set of $V \setminus T$ is an orphaned set.) The challenge here is that we should update $V_{\mathrm{H}}$ in the former case while keep using the same $V_{\mathrm{H}}$ in the latter case, which requires us to distinguish the cases.

To this end, we employ a new idea to select a "candidate vertex" that splits away from the orphaned set to address this challenge. Concretely, since the set of orphaned vertices is sufficiently large, when doing random sampling, we can get a vertex $u'$ from the orphaned set. Then, we use $V_{\mathrm{H}}$ to test whether there exist vertices that split away from $(u', t)$ for

sufficiently many $t \in V_{\mathrm{H}}$. The idea here is that if $u'$ is in the orphaned set, and there still exists a vertex $u$ that splits earlier than $u'$ in (the small-tree split order of) $V_{\mathrm{H}}$, then $u$ should split away from many $(u', t)$ pairs. On the other hand, if $u'$ is on the smaller side of the root cut of $V_{\mathrm{H}}$, there is only a small number of $t$ that any $u \in V$ can split away from. As such, we can make progress by either identifying the "right" candidate vertex that splits earlier than $u'$, or by switching the horizon $V_{\mathrm{H}}$ and recurse on the root cut.

**Merging of two weakly consistent partial trees.** In the case of strongly consistent partial trees, the merging of subtrees is very straightforward: since the splits always follow the top-down order of internal nodes, we can easily merge the two subtrees with a common parent node. In the case of weakly consistent partial trees, the story is much more complicated. For the merge to be correct, we have to correctly identify the "orphaned" subtree in the previous recursion; however, since we only have access to the splitting oracle, and the actual optimal tree structure is hidden from us, it is not immediately clear how could we identify the "correct" internal node to merge the trees.

Fortunately, we could utilize the "good vertex" from the previous recursion to identify the "orphaned" set of vertices $V^{\mathrm{orphan}}$. In particular, let $u$ be the "good vertex" we used to split the tree; since $u$ also belongs to $V^{\mathrm{orphan}}$, for each vertex $v$, we can test how many times $v$ *splits away* from $(u, t)$ for $t \in T$, where $T$ is the single maximal tree to be merged in the level of recursion. If $v \in V^{\mathrm{orphan}}$, such a vertex should not split away from $(u, t)$; otherwise, if $v \notin V^{\mathrm{orphan}}$, $v$ should split away from $(u, t)$. Since the size of $T$ is large enough, we could identify $V^{\mathrm{orphan}}$ correctly with high probability, and perform the merge correctly. An illustration for this idea to identify $V^{\mathrm{orphan}}$ is in Figure 10 in Appendix H.

**The complexity of the algorithm for weakly consistent trees.** Similar to the case for the strongly consistent trees, the subroutine that gives the sibling of a small tree for a fixed vertex $u$ takes $O(n^2)$ time. However, in the new algorithm, we only need to sample and test $O(\log n)$ vertices $u$ for each iteration. Similarly, the root test and the tree merging subroutines both use $O(\log n)$ vertices and $\tilde{O}(n^2)$ time. Furthermore, since we can roughly reduce the instance size by a $(1 - 1/\mathrm{polylog}\, n)$ factor every iteration, the entire process converges in polylog $n$ iterations. As such, we could argue that the total running time is $\tilde{O}(n^2)$, and the longest chain of dependent calls is polylog $n$. This would imply a near-linear time offline algorithm and a parallel algorithm with near-linear work and poly-logarithmic depth.

# C. Missing Proofs of Section 4 (Dasgupta's Objective)

We give the proofs we skipped in Section 4 in this section.

## C.1. Missing proofs of Section 4.1

**Formal proof of Theorem 1.** With Algorithm 1, we can obtain the poly-time efficiency from Lemma 4.1. Furthermore, since the strong partial tree algorithm of Theorem 10 succeeds with high probability, the approximation guarantee of Lemma 4.2 holds with high probability as well. This concludes the proof.

## C.2. Missing proofs of Section 4.2

*Proof of Lemma 4.3.* The first step of the algorithm that computes the strong partial tree takes $O(n^3 \log n)$ time and $O(n^3)$ queries by Theorem 10. We need to argue that the second step takes at most $O(n^3 \log n)$ time as well. Note that by Proposition 18, the algorithm, with high probability, runs in $\tilde{O}(s^2)$ time and finds an $O(\sqrt{\log s})$-approximation of the sparsest cut $(A, B)$, where $s$ is the input size. In our case, for a single super-vertex of $\mathcal{I}$ with $|X| \le 50000 \log n$, the running time is therefore $O(\log^2 n)$ only. By Fact A.1, there are at most $O(\log n)$ nodes in a binary tree with $O(\log n)$ leaves, which implies $O(\log^3 n)$ running time for a single super-vertex. Finally, with at most $O(n)$ super-vertices in the tree, the second step only takes $O(n \cdot \log^3 n) = O(n^3 \log n)$ time, as desired. $\qquad\square$

*Proof of Lemma 4.4.* Similar to the proof of Lemma 4.2, for any partial tree $\mathcal{I}$, we first partition the edges in to $E_{\mathrm{cross}}$ and $E_{\mathrm{same}}$ based on whether the edge $(u, v) \in E$ crosses different partial trees, i.e.,

1. $(u, v) \in E_{\mathrm{cross}}$ iff $u \in X$ and $v \in Y$ for some super-vertices $X \ne Y$ in $\mathcal{I}$.

2. $(u, v) \in E_{\mathrm{same}}$ iff $u, v \in X$ for some super-vertex $X$ in $\mathcal{I}$.

Again, by using Observation 1, we can show that $\mathsf{OPT}^{\mathrm{Das}} = \mathsf{cost}_G(\mathcal{T}^*) = \mathsf{cost}_G(\mathcal{T}^*, E_1) + \mathsf{cost}_G(\mathcal{T}^*, E_2)$. The costs w.r.t. to $E_1$ and $E_2$ are therefore as follows.

1. For $E_{\mathrm{cross}}$, we have that $\mathsf{cost}_G(\mathcal{T}, E_{\mathrm{cross}}) = \mathsf{cost}_G(\mathcal{T}^*, E_{\mathrm{cross}})$ by using the same argument of Lemma 4.2.

2. For $E_{\mathrm{same}}$, we argue that $\mathsf{cost}_G(\mathcal{T}, E_{\mathrm{same}}) \leq O(\sqrt{\log \log n}) \cdot \mathsf{cost}_G(\mathcal{T}^*, E_{\mathrm{same}})$. Formally, since algorithm in Proposition 18 finds an $O(\sqrt{\log s})$-approximation for each super-vertex $X$, we can use Proposition 17 on $G[X]$ to argue that the $\mathsf{cost}_G(\mathcal{T}, E_{\mathrm{same}}[X]) \leq O(\sqrt{\log \log n}) \cdot \mathsf{cost}_G(\mathcal{T}^*, E_{\mathrm{same}}[X])$ since $s = O(\log n)$. Therefore, we can apply the same argument to every super-vertex to get the desired approximation factor.

We now use Observation 1 again on $E_{\mathrm{cross}}$ and $E_{\mathrm{same}}$ to bound that

$$
\begin{aligned}
\mathsf{cost}_G(\mathcal{T}) &= \mathsf{cost}_G(\mathcal{T}, E_{\mathrm{cross}}) + \mathsf{cost}_G(\mathcal{T}, E_{\mathrm{same}}) \\
&\leq \mathsf{cost}_G(\mathcal{T}^*, E_{\mathrm{cross}}) + O(\sqrt{\log \log n}) \cdot \mathsf{cost}_G(\mathcal{T}^*, E_{\mathrm{same}}) \\
&\leq O(\sqrt{\log \log n}) \cdot (\mathsf{cost}_G(\mathcal{T}^*, E_{\mathrm{cross}}) + \mathsf{cost}_G(\mathcal{T}^*, E_{\mathrm{same}})) \\
&= O(\sqrt{\log \log n}) \cdot \mathsf{cost}_G(\mathcal{T}^*) = O(\sqrt{\log \log n}) \cdot \mathsf{OPT}^{\mathrm{Das}},
\end{aligned}
$$

as desired. $\qquad \square$

## D. Missing Details of Section 5 (Moseley-Wang Objective)

We now prove the approximation guarantee of Theorem 3. To this end, we will show the following technical lemma that lower bounds the number of non-leaves between $\mathcal{T}$ and $\mathcal{T}^*$.

**Lemma D.1.** *Let $\mathcal{T}$ be a hierarchical clustering tree obtained by Algorithm 3, and let $u, v \in V$ be any two vertices. Then, conditioning on the high probability event that $\mathcal{I}$ is a partial tree that is weakly consistent with $\mathcal{T}^*$, there is*

- *If $u$ and $v$ are in the same super-vertex $X$ of $\mathcal{I}$, then, there is*

$$
|\texttt{non-leaves}_{\mathcal{T}}[\texttt{LCA}_{\mathcal{T}}(u, v)]| \geq n - 50000 \log n.
$$

- *If $u$ and $v$ are in different super-vertices $X$ and $Y$ of $\mathcal{I}$, then, there is*

$$
|\texttt{non-leaves}_{\mathcal{T}}[\texttt{LCA}_{\mathcal{T}}(u, v)]| \geq |\texttt{non-leaves}_{\mathcal{T}^*}[\texttt{LCA}_{\mathcal{T}^*}(u, v)]| - 50000 \log n.
$$

*Furthermore, if*

$$
\texttt{leaves}_{\mathcal{T}^*}[\texttt{LCA}_{\mathcal{T}^*}(X)] \cap \texttt{leaves}_{\mathcal{T}^*}[\texttt{LCA}_{\mathcal{T}^*}(Y)] = \emptyset,
$$

*then, we additionally have*

$$
|\texttt{non-leaves}_{\mathcal{T}}[\texttt{LCA}_{\mathcal{T}}(u, v)]| = |\texttt{non-leaves}_{\mathcal{T}^*}[\texttt{LCA}_{\mathcal{T}^*}(u, v)]|.
$$

*Proof.* We prove the two cases separately as follows.

- **$u$ and $v$ are in the same super-vertex $X$ of $\mathcal{I}$.** By our construction, every leaf that is outside the subtree induced by $\texttt{leaves}_{\mathcal{T}}[\texttt{LCA}_{\mathcal{T}}(X)]$ counts as a non-leave of $(u, v)$. As such, since $|X| \leq 50000 \log n$, it is straightforward to get that

$$
|\texttt{non-leaves}_{\mathcal{T}}[\texttt{LCA}_{\mathcal{T}}(u, v)]| \geq n - 50000 \log n.
$$

- **$u$ and $v$ are in different super-vertices $X, Y$ of $\mathcal{I}$.** Suppose w.log. that $u \in X$ and $v \in Y$. By the *subtree preserving property*, we have $\texttt{leaves}_{\mathcal{T}}[\texttt{LCA}_{\mathcal{T}}(u, v)] = \texttt{leaves}_{\mathcal{T}^*}[\texttt{LCA}_{\mathcal{T}^*}(X \cup Y)]$. We now discuss further two sub-cases:

  a). If $\texttt{leaves}_{\mathcal{T}^*}[\texttt{LCA}_{\mathcal{T}^*}(X)]$ and $\texttt{leaves}_{\mathcal{T}^*}[\texttt{LCA}_{\mathcal{T}^*}(Y)]$ are disjoint. In this case, we have exactly $\texttt{leaves}_{\mathcal{T}^*}[\texttt{LCA}_{\mathcal{T}^*}(X \cup Y)] = \texttt{leaves}_{\mathcal{T}^*}[\texttt{LCA}_{\mathcal{T}^*}(u, v)]$, which implies $\texttt{leaves}_{\mathcal{T}}[\texttt{LCA}_{\mathcal{T}}(u, v)] = \texttt{leaves}_{\mathcal{T}^*}[\texttt{LCA}_{\mathcal{T}^*}(u, v)]$. Therefore, we have

$$
|\texttt{non-leaves}_{\mathcal{T}}[\texttt{LCA}_{\mathcal{T}}(u, v)]| = |\texttt{non-leaves}_{\mathcal{T}^*}[\texttt{LCA}_{\mathcal{T}^*}(u, v)]|.
$$

  This proves the "furthermore" part of the second case.

b). If $\texttt{leaves}_{\mathcal{T}^*}[\texttt{LCA}_{\mathcal{T}^*}(X)]$ and $\texttt{leaves}_{\mathcal{T}^*}[\texttt{LCA}_{\mathcal{T}^*}(Y)]$ have intersections. In this case, note that by the *weak contraction property*, we must have *inclusion* relationship between $\texttt{leaves}_{\mathcal{T}^*}[\texttt{LCA}_{\mathcal{T}^*}(X)]$ and $\texttt{leaves}_{\mathcal{T}^*}[\texttt{LCA}_{\mathcal{T}^*}(Y)]$. Suppose without loss of generality, that $\texttt{leaves}_{\mathcal{T}^*}[\texttt{LCA}_{\mathcal{T}^*}(X)] \supseteq \texttt{leaves}_{\mathcal{T}^*}[\texttt{LCA}_{\mathcal{T}^*}(Y)]$, the differences between $\texttt{leaves}_{\mathcal{T}^*}[\texttt{LCA}_{\mathcal{T}^*}(X \cup Y)]$ and $\texttt{leaves}_{\mathcal{T}^*}[\texttt{LCA}_{\mathcal{T}^*}(u, v)]$ is at most the set of $X$. Since $|X| \leq 50000 \log n$, we have

$$|\texttt{non-leaves}_{\mathcal{T}}[\texttt{LCA}_{\mathcal{T}}(u, v)]| \geq |\texttt{non-leaves}_{\mathcal{T}^*}[\texttt{LCA}_{\mathcal{T}^*}(u, v)]| - 50000 \log n,$$

as desired.

This concludes the proof of Lemma D.1. $\qquad\qquad\square$

To prove the approximation guarantee of Theorem 3, we will need a handful of structural observations for the optimal HC trees in the Moseley-Wang objective. We first observe that for any optimal HC tree $\mathcal{T}^*$ necessarily has a "monotone edge weight" property between an internal node and its descendants.

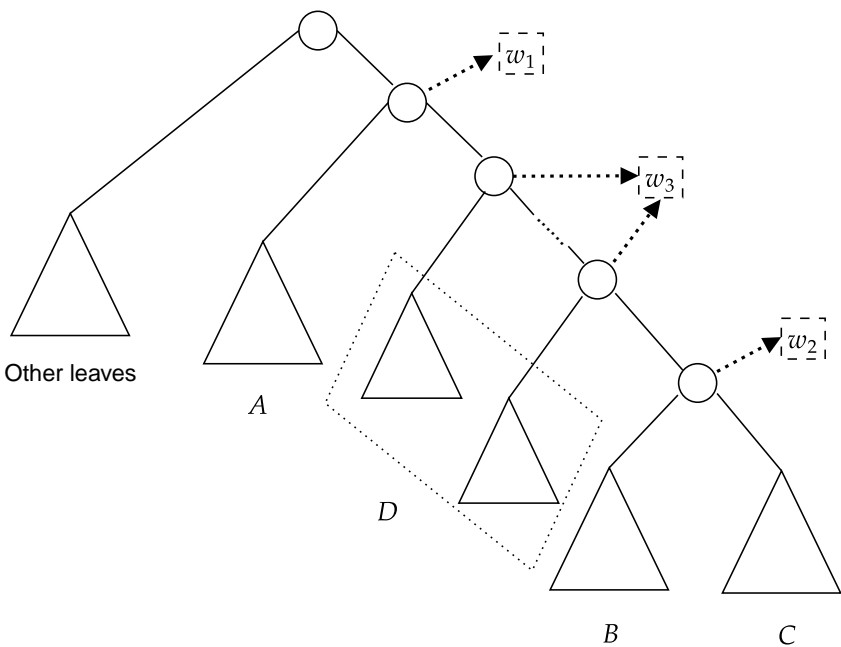

*Figure 4.* An illustration of the edge weights and leaves described in Claim D.2.

**Claim D.2.** *Let $z_1$ and $z_2$ be any two internal nodes of the optimal HC tree $\mathcal{T}^*$ under the Moseley-Wang objective, and let $z_2$ be a descendant node of $z_1$. We use $w_1$ and $w_2$ to denote the total weights of the edges induced by $z_1$ and $z_2$, respectively. Furthermore, let $w_3$ be the total weights of the edges induced on the internal nodes between $z_1$ and $z_2$.*

*Suppose the partition induced by $z_1$ is $S \to (A, S \setminus A)$, and let $B$ and $C$ be the set of vertices induced by $z_2$ such that $B \cup C \subseteq S \setminus A$. Furthermore, let $D = (S \setminus A) \setminus (B \cup C)$, and suppose $\max\{|B|, |C|\} \geq |A|$. Then, there is*

$$\frac{w_1 - (\frac{|A|}{|D|} + 1) \cdot w_3}{|A| + |D|} \leq \frac{w_2}{\max\{|B|, |C|\}}.$$

*An illustration of the edges and leaves used in the statement can be found in Figure 4.*

*Proof.* The claim is similar in spirit to the "switching lemma" under Dasgupta's objective as proved in (Høgemo et al., 2021). Suppose w.log. that $|B| \geq |C|$. For the purpose of this proof (and also that of Claim D.3), we let $\mathsf{rev}_G(\mathcal{T}, E_1)$ be the revenue

induced by a subset of edges $E_1$ with $\mathcal{T}$ being the HC tree of $G$. We further observe some useful relationships between the weights $w_1$, $w_2$, $w_3$ and the weights $w(A, B)$, $w(A, C)$, $w(A, D)$, $w(B, C)$, $w(B, D)$, and $w(C, D)$ as follows.

$$w_1 = w(A, D) + w(A, B) + w(A, C) \qquad w_2 = w(B, C) \qquad w_3 \geq w(B, D) + w(C, D).$$

We first prove a self-contained structural claim that the edge weights between $A$ and $B$ cannot be too much bigger than $w_3$. More formally, the claim is as follows.

**Claim D.3.** *Let A, B, C, and D be the set of vertices and $w_1$, $w_2$, and $w_3$ be the edge weights as prescribed in Claim D.2. Furthermore, let $E(A, B)$ be the edges between A and B, and $w(A, B)$ be the weights of $E(A, B)$. Then, there is*

$$w(A, B) \leq \frac{|A|}{|D|} \cdot w_3.$$

*Proof.* As consistent with the proof of Claim D.2, we assume w.log. that $|B| \geq |C|$. Let us construct a tree $\mathcal{T}^{(1)}$ based on $\mathcal{T}^*$ by switching the subtrees induced by $A$ and $D$. Note that in such change of HC tree, the only edges that will have a changed revenue are $E(A, B)$, $E(A, C)$, $E(A, D)$, $E(B, C)$, and edges accounted by $w_3$, which we call $E_3$ (including but not limited to $E(B, D)$ and $E(C, D)$), and edges in. We list these changes as follows.

1. $E(A, B)$: we have that $\mathsf{rev}_G(\mathcal{T}^{(1)}, E_{A,B}) - \mathsf{rev}_G(\mathcal{T}^*, E_{A,B}) \geq |D| \cdot w(A, B)$.

2. $E(A, C)$: we have that $\mathsf{rev}_G(\mathcal{T}^{(1)}, E_{A,C}) - \mathsf{rev}_G(\mathcal{T}^*, E_{A,C}) \geq |D| \cdot w(A, C)$.

3. $E(A, D)$: we have that $\mathsf{rev}_G(\mathcal{T}^{(1)}, E_{A,D}) - \mathsf{rev}_G(\mathcal{T}^*, E_{A,D}) = 0$.

4. $E(B, C)$: we have that $\mathsf{rev}_G(\mathcal{T}^{(1)}, E_{B,C}) - \mathsf{rev}_G(\mathcal{T}^*, E_{B,C}) = 0$.

5. $E_3$: we have that $\mathsf{rev}_G(\mathcal{T}^{(1)}, E_3) - \mathsf{rev}_G(\mathcal{T}^*, E_3) \geq -w_3 \cdot |A|$.

To maintain the optimality of $\mathcal{T}^*$, there should be $\mathsf{rev}_G(\mathcal{T}^{(1)}) - \mathsf{rev}_G(\mathcal{T}^*) \leq 0$. As such, we have

$$
\begin{aligned}
0 &\geq \mathsf{rev}_G(\mathcal{T}^{(1)}) - \mathsf{rev}_G(\mathcal{T}^*) \\
&= \Big(\mathsf{rev}_G(\mathcal{T}^{(1)}, E_{A,B}) - \mathsf{rev}_G(\mathcal{T}^*, E_{A,B})\Big) + \Big(\mathsf{rev}_G(\mathcal{T}^{(1)}, E_{A,C}) - \mathsf{rev}_G(\mathcal{T}^*, E_{A,C})\Big) \\
&\quad + \Big(\mathsf{rev}_G(\mathcal{T}^{(1)}, E_{A,D}) - \mathsf{rev}_G(\mathcal{T}^*, E_{A,D})\Big) + \Big(\mathsf{rev}_G(\mathcal{T}^{(1)}, E_{B,C}) - \mathsf{rev}_G(\mathcal{T}^*, E_{B,C})\Big) \\
&\quad + \mathsf{rev}_G(\mathcal{T}^{(1)}, E_3) - \mathsf{rev}_G(\mathcal{T}^*, E_3) \\
&\geq (w(A, B) + w(A, C)) \cdot |D| - w_3 \cdot |A|.
\end{aligned}
$$

As such, we can move the terms around, and obtain that

$$w(A, B) \leq w(A, B) + w(A, C) \leq \frac{|A|}{|D|} \cdot w_3,$$

as desired. Claim D.3 □

We now use Claim D.3 to prove Claim D.2. We again construct a tree $\mathcal{T}^{(2)}$ based on $\mathcal{T}^*$ by switching the subtrees induced by $A$ *and* $B$ (note that this is different from the proof of Claim D.3). Observe again that only the edges that have different revenue contribution in $\mathcal{T}^{(2)}$ vs. $\mathcal{T}^*$ are $E(A, B)$, $E(A, C)$, $E(A, D)$, $E(B, C)$, and edges accounted by $w_3$. We again list all the changes of revenue induced on these edges.

1. $E(A, B)$: we have that $\mathsf{rev}_G(\mathcal{T}^{(2)}, E_{A,B}) - \mathsf{rev}_G(\mathcal{T}^*, E_{A,B}) = 0$.

2. $E(A, C)$: we have that $\mathsf{rev}_G(\mathcal{T}^{(2)}, E_{A,C}) - \mathsf{rev}_G(\mathcal{T}^*, E_{A,C}) \geq (|B| + |D|) \cdot w(A, C)$.

3. $E(A, D)$: we have that $\mathsf{rev}_G(\mathcal{T}^{(2)}, E_{A,D}) - \mathsf{rev}_G(\mathcal{T}^*, E_{A,D}) \geq |B| \cdot w(A, D)$.

4. $E(B, C)$: we have that $\mathsf{rev}_G(\mathcal{T}^{(2)}, E_{B,C}) - \mathsf{rev}_G(\mathcal{T}^*, E_{B,C}) \geq -(|A| + |D|) \cdot w(B, C)$.

5. $E_3$: we have that $\text{rev}_G(\mathcal{T}^{(2)}, E_3) - \text{rev}_G(\mathcal{T}^*, E_3) \geq -|A| \cdot w_3$.

Therefore, by the optimally of $\mathcal{T}^*$, there should be $\text{rev}_G(\mathcal{T}^{(2)}) - \text{rev}_G(\mathcal{T}^*) \leq 0$. As such, we have

$$
\begin{aligned}
0 &\geq \text{rev}_G(\mathcal{T}^{(2)}) - \text{rev}_G(\mathcal{T}^*) \\
&= \Big( \text{rev}_G(\mathcal{T}^{(2)}, E_{A,B}) - \text{rev}_G(\mathcal{T}^*, E_{A,B}) \Big) + \Big( \text{rev}_G(\mathcal{T}^{(2)}, E_{A,C}) - \text{rev}_G(\mathcal{T}^*, E_{A,C}) \Big) \\
&\quad + \Big( \text{rev}_G(\mathcal{T}^{(2)}, E_{A,D}) - \text{rev}_G(\mathcal{T}^*, E_{A,D}) \Big) + \Big( \text{rev}_G(\mathcal{T}^{(2)}, E_{B,C}) - \text{rev}_G(\mathcal{T}^*, E_{B,C}) \Big) \\
&\quad + \Big( \text{rev}_G(\mathcal{T}^{(2)}, E_3) - \text{rev}_G(\mathcal{T}^*, E_3) \Big) \\
&\geq (|B| + |D|) \cdot w(A, C) + |B| \cdot w(A, D) - (|A| + |D|) \cdot w(B, C) - w_3 \cdot |A| \,.
\end{aligned}
$$

As such, by moving the terms around, we can get that

$$
\begin{aligned}
|B| \cdot (w(A, C) + w(A, D)) &\leq (|B| + |D|) \cdot w(A, C) + |B| \cdot w(A, D) \\
&\leq (|A| + |D|) \cdot w(B, C) + |A| \cdot w_3.
\end{aligned}
$$

We can use the observation that $w_1 = w(A, B) + w(A, C) + w(A, D)$ to obtain that

$$
\begin{aligned}
|B| \cdot w_1 &= (w(A, C) + w(A, B) + w(A, D)) \\
&\leq (|A| + |D|) \cdot w(B, C) + |A| \cdot w_3 + |B| \cdot w(A, B)
\end{aligned}
$$

by adding $|B| \cdot w(A, B)$ on both sides. Now, we can apply Claim D.3 to obtain that

$$
\begin{aligned}
|B| \cdot w_1 &\leq (|A| + |D|) \cdot w(B, C) + |A| \cdot w_3 + |B| \cdot \frac{|A|}{|D|} \cdot w_3 \\
&\leq (|A| + |D|) \cdot w(B, C) + |B| \cdot w_3 + |B| \cdot \frac{|A|}{|D|} \cdot w_3 && \text{(using } |A| \leq |B|\text{)}
\end{aligned}
$$

Note that $w(B, C) = w_2$. As such, the above implies that

$$
|B| \cdot \left( w_1 - (1 + \frac{|A|}{|D|}) w_3 \right) \leq (|A| + |D|) \cdot w_2.
$$

Moving the turns around in the above inequality gives us the desired bound. Claim D.2 $\square$

We now use Claim D.2 to show that the set of edges $(u, v)$ such that

a). has at most $O(\log^2 n)$ non-leaves in $\mathcal{T}^*$; and

b). let $X$ and $Y$ be corresponding super-vertices that contain $u$ and $v$ in a weakly consistent partial tree $\mathcal{I}$; there is $\text{leaves}_{\mathcal{T}^*}[\text{LCA}_{\mathcal{T}^*}(X)] \cap \text{leaves}_{\mathcal{T}^*}[\text{LCA}_{\mathcal{T}^*}(Y)] \neq \emptyset$.

can contribute to at most an $o(1)$ fraction of the optimal cost. This means that our estimation with non-leaves using Lemma D.1 would lead to a good approximation. The formal statement is as follows.

**Lemma D.4.** *Let $\mathcal{I}$ be an arbitrary partial HC tree that is weakly consistent with the optimal HC tree $\mathcal{T}^*$ under the Moseley-Wang objective. Define $E_{\text{low}}(\mathcal{I}) \subseteq V \times V$ as the edges such that for any $(u, v) \in E_{\text{low}}(\mathcal{I})$, there is*

a). $|\text{non-leaves}_{\mathcal{T}^*}[\text{LCA}_{\mathcal{T}^*}(u, v)]| \leq 50000 \cdot \log^2 n$.

b). *Let $X$ and $Y$ be corresponding super-vertices that contain $u$ and $v$ in $\mathcal{I}$; there is*

$$
\text{leaves}_{\mathcal{T}^*}[\text{LCA}_{\mathcal{T}^*}(X)] \cap \text{leaves}_{\mathcal{T}^*}[\text{LCA}_{\mathcal{T}^*}(Y)] \neq \emptyset.
$$

*Then, for sufficiently large $n$, the total contribution of revenue from the edges in $E_{\text{low}}(\mathcal{I})$ is at most $50000 \cdot \frac{\log^4 n}{n}$ fraction of* $\textsf{OPT}^{\text{MW}}$, *i.e.,*

$$\sum_{e=(u,v)\in E_{\text{low}}(\mathcal{I})} w(e) \cdot |\texttt{non-leaves}_{\mathcal{T}^*}[\texttt{LCA}_{\mathcal{T}^*}(u,\,v)]| \leq O(\frac{\log^4 n}{n}) \cdot \textsf{OPT}^{\text{MW}}.$$

*Proof.* For any two super-vertices $X$ and $Y$ in the partial HC tree $\mathcal{I}$, by the *weak contraction property*, if $\texttt{leaves}_{\mathcal{T}^*}[\texttt{LCA}_{\mathcal{T}^*}(X)] \cap \texttt{leaves}_{\mathcal{T}^*}[\texttt{LCA}_{\mathcal{T}^*}(Y)]$ is not empty, the only possible case is to have *inclusion* relationships between the two sets of leaves. Suppose w.log. that $\texttt{leaves}_{\mathcal{T}^*}[\texttt{LCA}_{\mathcal{T}^*}(X)] \supseteq \texttt{leaves}_{\mathcal{T}^*}[\texttt{LCA}_{\mathcal{T}^*}(Y)]$. Let $\tilde{Y}$ be the set of vertices induced by the sibling node of $X$ in $\mathcal{I}$, and it is straightforward to see that $Y \subseteq \tilde{Y}$. We further let $Y'$ be the *larger* immediate child of the tree induced by $\tilde{Y}$.

By the conditions of $i)$. $|\texttt{non-leaves}_{\mathcal{T}^*}[\texttt{LCA}_{\mathcal{T}^*}(u,\,v)]| \leq 50000 \cdot \log^2 n$, $ii)$. $|X| \leq 5000 \log n$, and $iii)$. the inclusion relationship between the leaves, there is

$$\left|\tilde{Y}\right| \geq n - 50000 \cdot (\log^2 n + \log n); \qquad |Y'| \geq \frac{n - 50000 \cdot (\log^2 n + \log n)}{2}.$$

We now use Claim D.2 inductively to bound the weights of $w(E_1)$ for the sets of edges $E_1 \subseteq E_{\text{low}}$ that are split by some internal nodes of $X$. Let $E_2$ be the set of edges that is split in the subtree induced by $Y'$, and let $\tilde{X}$ be vertices that are split by $E_1$ and as the sibling of $\tilde{Y}$. Our induction hypothesis is that

$$w(E_1) < \frac{C \cdot \log n \cdot w(E_2)}{n - 50000 \cdot (\log^2 n + \log n)}$$

for some absolute constant $C$.

To prove this statement, we first look at the base case. By the size bound on $X$, it is straightforward to see that

$$|X| - \left|\tilde{X}\right| \leq 50000 \log n.$$

In the base case, we pick edges $E_1 \subseteq E_{\text{low}}$ such that $E_2$ is split immediately after $E_1$, i.e., $E_1$ are the edges between $\tilde{X}$ and $\tilde{Y}$. Now, we can use Claim D.2 with $A \leftarrow \tilde{X}$, $B \leftarrow Y'$, and $D \leftarrow \emptyset$ ($w_3 = 0$) to argue that

$$\frac{w(E_1)}{|X|} \leq \frac{w(E_1)}{\left|\tilde{X}\right|} < \frac{w(E_2)}{|Y'|}.$$

By the size upper bound of $X$ and the size lower bound of $Y'$, the above inequality implies that

$$\frac{w(E_1)}{50000 \log n} < \frac{2 \cdot w(E_2)}{n - 50000 \cdot (\log^2 n + \log n)} < \frac{C \cdot \log n \cdot w(E_2)}{n - 50000 \cdot (\log^2 n + \log n)}$$

for any $C > 2$, which proves the base case.

For the inductive step, let us suppose the statement holds until some internal node $z$, and we look into $E_1 \subseteq E_{\text{low}}$ that is induced on $\textsf{pa}(z)$. We again use Claim D.2 by letting $A \leftarrow \tilde{X}$, $B \leftarrow Y'$, and $D \leftarrow X \setminus \tilde{X}$. Define $E_3$ as the set of edges that are split between $z$ and $\texttt{LCA}_{\mathcal{T}^*}(\tilde{Y})$. By the induction hypothesis, for every node between $z$ and $\texttt{LCA}_{\mathcal{T}^*}(\tilde{Y})$, the total induced weights is at most

$$\frac{C \cdot \log n \cdot w(E_2)}{n - 50000 \cdot (\log^2 n + \log n)}.$$

Furthermore, since $\frac{|X|}{|\tilde{X}|} \leq |X| \leq 50000 \log n$, and $z$ and $\textsf{pa}\left(\texttt{LCA}_{\mathcal{T}^*}(\tilde{Y})\right)$ are in the same $X$ of $\mathcal{I}$, we can bound the total weights in $E_3$ as follows

$$w(E_3) \leq 50000 \cdot \log n \cdot \frac{C \cdot \log n \cdot w(E_2)}{n - 50000 \cdot (\log^2 n + \log n)}.$$

Furthermore, by the size upper bound of $X$, we have $|A| + |D| \leq 50000 \cdot \log n$ in Claim D.2. Combining the above gives us that for sufficiently large $n$, we have

$$\frac{1}{2} \cdot \frac{w(E_1)}{|X|} \leq \frac{w(E_1) - (50000 \log n + 1) \cdot w(E_3)}{|X|} \leq \frac{w(E_1) - (\frac{|X|}{|\tilde{X}|} + 1) \cdot w(E_3)}{|X|} < \frac{w(E_2)}{|Y'|},$$

where the first inequality follows from the fact that $w(E_3) = O(\frac{\log^2 n}{n}) \cdot w(E_2)$, the second inequality follows from that $|X| / |\tilde{X}| \leq |X| \leq 50000 \log n$, and the last inequality follows from Claim D.2. Therefore, we can again obtain that

$$\frac{w(E_1)}{100000 \log n} < \frac{4 \cdot w(E_2)}{n - 50000 \cdot (\log^2 n + \log n)},$$

which means $w(E_1) < \frac{C \cdot \log n \cdot w(E_2)}{n - 50000 \cdot (\log^2 n + \log n)}$ for a sufficiently large constant $C$ ($C = 400000$ suffices). This concludes our inductive proof for the weight bound of any $E_1 \subseteq E_{\text{low}}$ in any $X$.

Using $w(E_1) \leq \frac{C \cdot \log n}{n - 50000 \cdot (\log^2 n + \log n)} \cdot w(E_2)$ for any $(u, v) \in E_{\text{low}}$, we note that a trivial lower bound of the optimal revenue is

$$\mathsf{OPT}^{\text{MW}} = \Omega(\log n \cdot w(E_2)),$$

since this is the revenue induced on the edge set $E_2$ only. On the other hand, since we need the inclusion relationship between the leaves, and by the fact that the number of non-leaves is at most $\log^2 n$, there are at most $50000(\log^2 n + \log n)$ edges in $E_{\text{low}}$, i.e.,

$$|E_{\text{low}}| \leq 50000(\log^2 n + \log n).$$

Therefore, the total contribution of revenue by $E_{\text{low}}$ is at most

$$\sum_{e=(u,v) \in E_{\text{low}}(\mathcal{I})} w(e) \cdot |\texttt{non-leaves}_{\mathcal{T}^*}[\texttt{LCA}_{\mathcal{T}^*}(u, v)]|$$

$$\leq \sum_{e=(u,v) \in E_{\text{low}}(\mathcal{I})} w(e) \cdot \log^2 n \qquad \text{(by the number of non-leaves)}$$

$$\leq \max\{w(E_1) \mid E_1 \subseteq E_{\text{low}}\} \cdot 50000(\log^2 n + \log n) \cdot \log^2 n \qquad \text{(uniform upper bound)}$$

$$\leq \frac{C \cdot \log n}{n - 50000 \cdot (\log^2 n + \log n)} \cdot w(E_2) \cdot 50000(\log^2 n + \log n) \cdot \log^2 n$$

$$\qquad \text{(by the relationship between } w(E_2) \text{ and } w(E_1) \text{ for } E_1 \subseteq E_{\text{low}})$$

$$\leq O(\frac{\log^4 n}{n} \cdot \mathsf{OPT}^{\text{MW}}), \qquad \text{(by the lower bound of } \mathsf{OPT}^{\text{MW}})$$

as desired. □

**Finalizing the proof of Theorem 3.** We have discussed that the algorithm enjoys $\tilde{O}(n^2)$ running time and $O(n^2)$ query efficiency. For the approximation guarantee, note that each edge $(u, v)$ gains a revenue of $w_{u,v} \cdot |\texttt{non-leaves}_{\mathcal{T}^*}[\texttt{LCA}_{\mathcal{T}^*}(u, v)]|$ in the optimal tree $\mathcal{T}^*$. For edges $(u, v) \notin E_{\text{low}}$, let $E_{\text{high}}^{\text{same}}$ be the set of edges where $u$ and $v$ are in the same super-vertex, and $E_{\text{high}}^{\text{diff}}$ be the set of edges where $u$ and $v$ are in different super-vertices. We now have

- For edges in $E_{\text{high}}^{\text{same}}$, we have $|\texttt{non-leaves}_{\mathcal{T}}[\texttt{LCA}_{\mathcal{T}}(u, v)]| \geq n - 50000 \log n$ that by Lemma D.1. On the other

hand, any vertex pair has at most $n$ non-leaves in the graph. Therefore, we have that

$$
\begin{aligned}
\mathsf{rev}_{G \cap E_{\mathrm{high}}^{\mathrm{same}}}(\mathcal{T}) &= \sum_{e \in E_{\mathrm{high}}^{\mathrm{same}}} w(e) \cdot |\texttt{non-leaves}_{\mathcal{T}}[\mathrm{LCA}_{\mathcal{T}}(u, v)]| \\
&\geq \sum_{e \in E_{\mathrm{high}}^{\mathrm{same}}} w(e) \cdot n - 50000 \log n \\
&\geq \sum_{e \in E_{\mathrm{high}}^{\mathrm{same}}} w(e) \cdot n \cdot (1 - \frac{50000 \log n}{n}) \\
&\geq (1 - \frac{50000 \log n}{n}) \cdot \mathsf{rev}_{G \cap E_{\mathrm{high}}^{\mathrm{same}}}(\mathcal{T}^*).
\end{aligned}
$$

- For edges in $E_{\mathrm{high}}^{\mathrm{diff}}$, we have

$$
\begin{aligned}
|\texttt{non-leaves}_{\mathcal{T}}[\mathrm{LCA}_{\mathcal{T}}(u, v)]| &\geq |\texttt{non-leaves}_{\mathcal{T}^*}[\mathrm{LCA}_{\mathcal{T}^*}(u, v)]| - 50000 \log n \\
&\geq (1 - O(\frac{1}{\log n})) \cdot |\texttt{non-leaves}_{\mathcal{T}^*}[\mathrm{LCA}_{\mathcal{T}^*}(u, v)]|,
\end{aligned}
$$

where the first inequality follows from Lemma D.1, and the second inequality is from the fact that $|\texttt{non-leaves}_{\mathcal{T}^*}[\mathrm{LCA}_{\mathcal{T}^*}(u, v)]| \geq 50000 \cdot \log^2 n$ for every $(u, v) \notin E_{\mathrm{low}}$. As such, we have that

$$
\begin{aligned}
\mathsf{rev}_{G \cap E_{\mathrm{high}}^{\mathrm{diff}}}(\mathcal{T}) &= \sum_{e \in E_{\mathrm{high}}^{\mathrm{diff}}} w(e) \cdot |\texttt{non-leaves}_{\mathcal{T}}[\mathrm{LCA}_{\mathcal{T}}(u, v)]| \\
&\geq \left(1 - O(\frac{1}{\log n})\right) \cdot \sum_{e \in E_{\mathrm{high}}^{\mathrm{diff}}} w(e) \cdot |\texttt{non-leaves}_{\mathcal{T}^*}[\mathrm{LCA}_{\mathcal{T}^*}(u, v)]| \\
&= \left(1 - O(\frac{1}{\log n})\right) \cdot \mathsf{rev}_{G \cap E_{\mathrm{high}}^{\mathrm{diff}}}(\mathcal{T}^*).
\end{aligned}
$$

Therefore, by additionally using Lemma D.4, we have that

$$
\begin{aligned}
\mathsf{rev}_G(\mathcal{T}) &\geq \mathsf{rev}_{G \cap E_{\mathrm{high}}^{\mathrm{same}}}(\mathcal{T}) + \mathsf{rev}_{G \cap E_{\mathrm{high}}^{\mathrm{diff}}}(\mathcal{T}) \\
&\geq (1 - O(\frac{1}{\log n})) \cdot \mathsf{rev}_{G \cap E_{\mathrm{high}}^{\mathrm{same}}}(\mathcal{T}^*) + \mathsf{rev}_{G \cap E_{\mathrm{high}}^{\mathrm{diff}}}(\mathcal{T}^*) \\
&\geq (1 - O(\frac{1}{\log n})) \cdot \left(\mathsf{rev}_{G \cap E_{\mathrm{high}}^{\mathrm{same}}}(\mathcal{T}^*) + \mathsf{rev}_{G \cap E_{\mathrm{high}}^{\mathrm{diff}}}(\mathcal{T}^*)\right) \\
&\geq (1 - O(\frac{1}{\log n})) \cdot \left(1 - O(\frac{\log^4 n}{n})\right) \cdot \mathsf{rev}_G(\mathcal{T}^*) \qquad \text{(using Lemma D.4)} \\
&\geq (1 - o(1)) \cdot \mathsf{rev}_G(\mathcal{T}^*),
\end{aligned}
$$

as desired.

*Remark* 20. We can observe that if we run Algorithm 3 with the *strongly consistent* partial HC tree, we can get a similar (and even stronger) approximation guarantee, albeit with worse efficiency ($\tilde{O}(n^3)$ time). Concretely, note that if we use the strongly consistent partial HC tree, the additive error again only happens on $E^{\mathrm{same}}$, and we do *not* need Lemma D.4 to bound the contributions of the edges in $E_{\mathrm{low}}(\mathcal{I})$ (since $= \emptyset$). In this way, we can further decrease the $o(1)$ term to $O(\frac{\log n}{n})$. However, the sacrifice of the running time is too significant, and we skip the details of this algorithm.

# E. Missing Details of Section 6 (Sublinear Algorithms)

We discuss the analysis of the algorithms and their analysis for the rest of this section.

## E.1. A single-pass semi-streaming HC algorithm for Dasgupta's objective

We present our algorithm for the semi-streaming algorithm in this part. To begin with, we need to define the *model* for streaming hierarchical clustering with the splitting oracle.

**Graph streaming with offline splitting oracle.** We focus on the (dynamic) graph streaming model with the *offline* splitting oracle. In this model, the edges of the graph are inserted and deleted (together with their edge weights), and the algorithm is asked to output an HC tree by the *end* of the stream. Additionally, the algorithm is given an offline splitting oracle $\mathcal{O}$ *before* the graph stream, and the algorithm is allowed to make unlimited computations before the stream starts. The total memory cost is the total number of bits the algorithm maintains at any point, including those dedicated to edge representation and those generated through offline computations.

For polynomial-time efficiency, we assume that the stream itself is of $\text{poly}(n)$ length since otherwise, the stream itself will take super-polynomial time to complete. Under the above model and setting, the formal statement of our result is as follows.

**Theorem 21.** *There exists a single-pass (dynamic) streaming algorithm that given a weighted undirected graph $G = (V, E, w)$ in a $\text{poly}(n)$-length dynamic stream and an offline splitting oracle $\mathcal{O}$, with high probability, in $O(n \cdot \log^3 n)$ (bits) space and polynomial time computes a hierarchical clustering tree $\mathcal{T}$ such that $\text{cost}_G(\mathcal{T}) \leq O(1) \cdot \text{OPT}^{\text{Das}}(G)$, where $\text{OPT}^{\text{Das}}(G)$ is the cost of the optimal hierarchical clustering tree $\mathcal{T}^*$, i.e., $\text{OPT}^{\text{Das}}(G) = \text{cost}_G(\mathcal{T}^*)$.*

*Proof.* The algorithm is to simply first construct a strong partial tree $\mathcal{I}$ with the algorithm of Definition 8 *before* the stream starts, and store edges only inside the same super-vertices during the stream. By the end of the stream, we compute recursive sparsest cuts on the vertices induced on super-vertices of $\mathcal{I}$, and output in the same manner of Algorithm 1. The formal description is as Algorithm 4.

We first prove the time and space efficiency. Essentially, Algorithm 4 computes a strong partial tree $\mathcal{I}$ before the stream starts, and only main edges inside each super-vertex of $O(\log n)$ size. By Theorem 10, the pre-processing part takes polynomial $(O(n^3 \log n))$ time and $O(n \log n)$ space. Furthermore, for each super-vertex, we maintain at most $O(\log^2 n)$ edges; each edge could have at most $\text{poly}(n)$ updates, which means an additive space of $O(\log n)$ bits suffices for each edge. Therefore, we record at most $O(\log n)$ bits for each edge. There are at most $n$ super-vertices, which means the algorithm maintains the information of $O(n \log^2 n)$ edges, which takes $O(n \log^3 n)$ bits. After the stream, we partition the super-vertices with the edges stored, and write down the rest of the HC tree. The time efficiency of the post-processing part is exactly as Lemma 4.1.

For the approximation guarantee, note that we are essentially simulating Algorithm 1 with the same input and output guarantees. As such, we can simply use Lemma 4.2 to argue the approximation guarantee. $\square$

### E.2. Parallel HC algorithms for Moseley-Wang Objective

We now move the PRAM hierarchical clustering algorithm for the Moseley-Wang objective with near-linear $\tilde{O}(n^2)$ work and polylog $(n)$ depth. The formal statement of the algorithm is as follows.

**Theorem 22.** *There exists a PRAM algorithm that given a weighted undirected graph $G = (V, E, w)$ and a splitting oracle $\mathcal{O}$, with high probability, in $O(n^2 \cdot \text{polylog}\, n)$ work and $\text{polylog}\, n$ depth computes a hierarchical clustering tree $\mathcal{T}$ such that $\text{rev}_G(\mathcal{T}) \geq (1 - o(1)) \cdot \text{OPT}^{\text{MW}}(G)$, where $\text{OPT}^{\text{MW}}(G)$ is the revenue of the optimal hierarchical clustering tree $\mathcal{T}^*$, i.e., $\text{OPT}^{\text{MW}}(G) = \text{rev}_G(\mathcal{T}^*)$.*

*Proof.* The algorithm is to run the PRAM weak partial tree algorithm in Corollary 12 to obtain $\mathcal{I}$, and arbitrarily partition the vertices in the super-vertices of $\mathcal{I}$. The formal description of the algorithm is as Algorithm 5.

By Corollary 12, the first step that computes the weak partial tree $\mathcal{I}$ only takes $\tilde{O}(n^2)$ work and $\text{polylog}\, n$ depth. For the second step, since the partition is arbitrary, we can simply perform arbitrary balanced cuts on the vertices for each super-vertex $X$. For an individual $X$, this procedure can be done in $O(\log n \cdot \log \log n)$ work and $O(\log \log n)$ depth. By accounting for all the super-vertices, we blow up the work by at most an $O(n)$ factor and the depth remains the same since we can partition super-vertices in parallel. Therefore, the second step takes $O(n \cdot \log^2 n)$ work and $O(\log \log n)$ depth. Therefore, the entire procedure takes $\tilde{O}(n^2)$ work and $\text{polylog}\, n$ depth.

The output of Theorem 22 follows exactly the same rules of Algorithm 3; therefore, the approximation guarantee follows from Theorem 3. $\square$

*Remark* 23. By the reduction of Proposition 14, the result of Theorem 22 also implies a fully-scalable Massively Parallel Computation (MPC) algorithm that computes the HC tree $\mathcal{T}$ with $(1 - o(1)) \cdot \text{OPT}^{\text{MW}}(G)$ revenue in $\tilde{O}(n^2)$ total memory and $\text{polylog}\, n$ depth. The memory on each machine here is allowed to be $O(n^\alpha)$ for any $\alpha \in (0, 1)$, and we use $\tilde{O}(n^{2-\alpha})$ total machines in the MPC algorithm.

# F. Preliminaries for Partial HC Tree Algorithms

We will discuss the construction of partial HC trees for the rest of this paper (Appendices F to H). In this section, we first introduce the several notions that are essential in the *analysis* of the partial HC tree algorithms. We use these notions extensively in the analysis of both Theorem 10 (Appendix G) and Theorem 11 (Appendix H).

In the high-level overview, we slightly abused the notation to use $V$ to denote the subset of vertices. In our formal description of the algorithms, we will use $\widetilde{V} \subseteq V$ as the set of vertices of the current recursion step, and use $\tilde{n} = \left|\widetilde{V}\right|$ as the size of the set.

## F.1. Composable vertex sets and restricted subtrees

To continue, we first introduce the notions of *composable vertex sets* for a graph and the HC tree restricted to the sets.

**Definition 24** (Composable vertex sets). Let $G = (V, E)$ be a $n$-vertex graph and $\mathcal{T}$ be a hierarchical clustering of $G$. For a subset $\widetilde{V} \subseteq V$, we say $\widetilde{V}$ is a *composable (vertex) set* of $(G, \mathcal{T})$ if $\widetilde{V}$ can be written in a disjoint union of the leaves of *maximal trees*, i.e., a union of vertices $\widetilde{V} = \cup_i \widetilde{V}_i$, such that *all* $\widetilde{V}_i$ satisfies that

$$\texttt{leaves}_\mathcal{T}[\texttt{LCA}_\mathcal{T}(\widetilde{V}_i)] = \widetilde{V}_i.$$

In the special case, we say $\widetilde{V}$ is a *single composable (vertex) set* if there is *only one* such maximal tree, i.e., $\texttt{leaves}_\mathcal{T}[\texttt{LCA}_\mathcal{T}(\widetilde{V})] = \widetilde{V}$.

In other words, if a vertex set $\widetilde{V}$ is composable, it means if we only look at the leaves of $\widetilde{V}$, they still form subtrees of $\mathcal{T}$. An illustration of composable sets can be found in Figure 5.

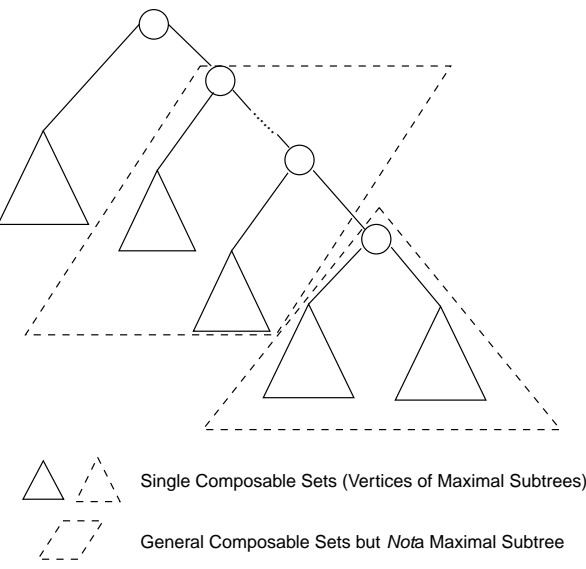

Single Composable Sets (Vertices of Maximal Subtrees)

General Composable Sets but *Not* a Maximal Subtree

*Figure 5.* An illustration of the composable vertex sets and maximal trees as in Definition 24.

We can define the HC tree restricted to composable subsets of vertices as follows.

**Definition 25** (Hierarchical clustering tree restricted to subset). Let $G = (V, E)$ be a $n$-vertex graph and $\mathcal{T}$ be a hierarchical clustering of $G$. For any *composable* subset $S \subseteq V$ such that $|S| \geq 2$, we call $\mathcal{T}(S)$ as $\mathcal{T}$ *restricted to* $S$ if $\mathcal{T}(S)$ is a new binary tree constructed with the following process

1. Extract $\mathcal{T}'$ from $\mathcal{T}$ by taking the induced subtree of the internal node $\text{LCA}_{\mathcal{T}}(S)$.

2. Remove all subtrees whose leaves contain *only* vertices *not* in $S$ from $\mathcal{T}'$.

3. If there exists an internal node $x$ that only has *a single* child, contract $x$ and its child to one node. Repeat until all internal node has two children nodes.

Note that the algorithm in Definition 25 is a *thought process*, and it only serves the purpose of analysis. The third line eventually terminates since there exists at least one subtree with two leaves for any *composable* set $S$ with at least two vertices. An illustration of HC trees restricted to subsets can be shown in Figure 6.

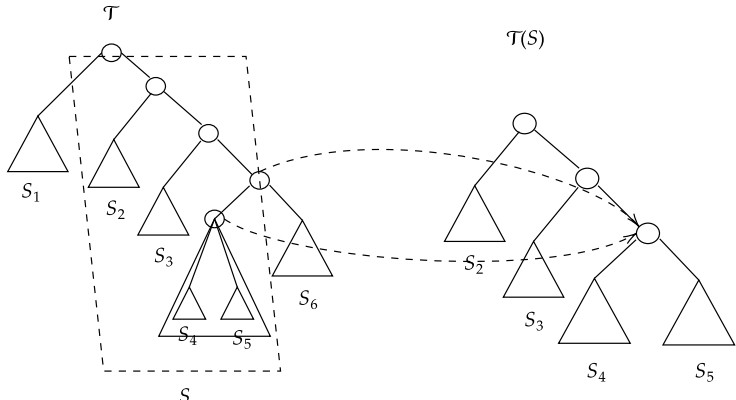

*Figure 6.* An illustration of the HC tree $\mathcal{T}$ to be restricted on a subset of vertices $S$.

We now provide the following observation that the new tree $\mathcal{T}(S)$ preserves the *relative order* of split away vertices of $\mathcal{T}$.

**Observation 2.** *For any triplet of vertices $(u, v, w)$, the orders of split-away for $(u, v, w)$ are the same in $\mathcal{T}(S)$ and $\mathcal{T}$, i.e., $w$ splits away from $(u, v)$ in $\mathcal{T}(S)$ if and only if $w$ splits away from $(u, v)$ in $\mathcal{T}$.*

*Proof.* We first observe that the orders of split away for any triplet $(u, v, w)$ are the same in $\mathcal{T}$ and $\mathcal{T}'$ since $\mathcal{T}'$ is a *subtree* of $\mathcal{T}$. Furthermore, since $S$ is a *conposeable set*, every subtree we remove from $\mathcal{T}'$ is necessarily a *maximal* subtree, i.e., fix the removed vertex set $S$, the leaves of $\text{LCA}_{\mathcal{T}'}(S)$ is $S$ itself.

Let $x$ be the lowest common ancestor of $(u, v)$ in $\mathcal{T}'$, and $x'$ be the lowest common ancestor of $(u, v, w)$ in $\mathcal{T}'$. By our definition, we have $\text{level}_{\mathcal{T}'}(x') > \text{level}_{\mathcal{T}'}(x)$. In $\mathcal{T}(S)$, if none of $x$ and $x'$ is contracted in line 3, then the order of splits away trivially remains the same as in $\mathcal{T}'$.

On the other hand, if at least one of $x$ and $x'$ is contracted, and let the new internal nodes be $y$ and $y'$, we claim that there is still $\text{level}_{\mathcal{T}(S)}(y') > \text{level}_{\mathcal{T}(S)}(y)$. This is simply because every removed tree is a maximal tree, and the only case that $y$ and $y'$ are merged is that the induced vertices of $y'$ become empty, which contradicts the fact that $(u, v)$ is not removed. Therefore, the split order between $(w, u, v)$ is the same as in $\mathcal{T}'$, which in turn is the same in $\mathcal{T}$. $\qquad\square$

### F.2. Small-tree splitting order

We now introduce the following notion of *small-tree split order*, which we frequently use in our analysis.

**Definition 26** (Small-tree split order). Consider any subset $S \subseteq V$ and the hierarchical clustering tree $\mathcal{T}^*(S)$ restricted to $S$, and consider the following process that divides the *leaves* of $S$:

1. Starting from the root $r$, let the split be $S \to (S_l^1, S_r^1)$, and assume w.log. $|S_l^1| \le |S_r^1|$. Let $V_1^{\text{small}}$ be the vertices (leaves) induced by the "smaller subtree" $S_l^1$.

2. Starting from the lowest common ancestor of $S_r^1$, recursively define $V_\ell^{\text{small}}$ as the vertices contained in the "smaller subtree" of level $\ell$ (by splitting from the "larger subtree" of level $\ell - 1$).

For the convenience of notation, we define $V_\ell^{\text{small}} = \emptyset$ when $\ell$ is larger than the depth of the tree. For any integer $N \in [0, n_H]$, we can define the *first $N$ vertices in the small-tree split order* by taking the first $N$ vertices in the order of $V_1^{\text{small}}, V_2^{\text{small}}$, etc.. Inside each set $V_i^{\text{small}}$, we pick vertices in an arbitrarily *fixed* order. We use the notation $V^{\text{small}}(\le N)$ to denote the *first $N$* vertices in the small-tree split order, and we use $V^{\text{small}}(\ge -N)$ to denote the *last $N$* vertices in the small-tree split order.

Intuitively, we can think of the small-tree split order as we always write the "smaller" tree on the left-hand side, and recurse on the "larger" tree for this procedure, then take leaves from the left-most vertices. Note that *every* vertex has to belong to $V_i^{\text{small}}$ for *some $i$*. An illustration can be found in Appendix F.2.

Based on the notion of small-tree split order, we can define the notion of *induced leaves* of the first or last $N$ vertices in the small-tree split order as follows.

**Definition 27** (Induced leaves of the $N$ first split vertices). Let $\mathcal{T}^*(S)$ and $S$ be the hierarchical clustering tree and the set of leaves. For any integers $N \in [0, n_H]$, we define the induced leaves of $V^{\text{small}}(\le N)$ as the union of the $\cup_{i=1}^\ell V_i^{\text{small}}$, where $\ell$ is the *maximum* level such that $V_\ell^{\text{small}}$ contains at least one vertex in $V^{\text{small}}(\le N)$.

An illustration of the induced leaves in Definition 27 can be found in Appendix F.2.

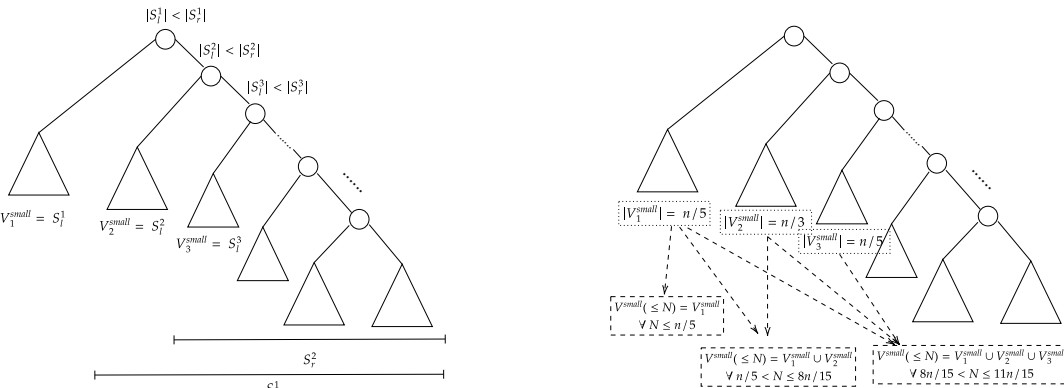

*Figure 7.* An illustration of the notion of small-tree split order Definition 26 and the Induced leaves of the $N$ first split vertices Definition 27.

## G. The Algorithm for the Strongly Consistent Partial HC Tree: Proof of Theorem 10

We now give an algorithm for partial trees that are strongly consistent with the optimal HC tree $\mathcal{T}^*$. We first remind the readers of our main result on the algorithm for strongly consistent partial HC trees as follows.

**Theorem 10.** *There exists an algorithm that given a splitting oracle $\mathcal{O}$ of a weighted undirected graph $G = (V, E, w)$, with high probability, in $O(n^3 \log n)$ time and $O(n^3)$ queries computes a partial hierarchical clustering tree $\mathcal{I}$ that is strongly consistent with the optimal hierarchical clustering tree $\mathcal{T}^*$. Furthermore, the algorithm has the following properties.*

  *i). The runtime of the algorithm is deterministic, and the high probability randomness is over the correctness guarantee.*

  *ii). The algorithm can be implemented in $O(n \log n)$ space.*

We refer the readers to Appendix B for a high-level overview of the algorithm. We directly give the algorithm and the analysis in this section.

**The algorithm.**  We introduce our main algorithm for the construction of strongly consistent partial HC trees. To begin with, we give a 'helper function' `counterpart-tester-strong` as follows. As we have discussed in the high-level overview, this function tests the 'sibling vertices' of a small tree that is 'early enough' in the small-tree splitting order.

---

**Algorithm 6** `counterpart-tester-strong`$^{(\mathcal{O}, V_{\text{input}})}(u, t, \text{threshold}_1, \text{threshold}_2)$: an algorithm to test whether $t$ is among the "counterpart" of $u$

---

**Input:** Vertex set $V_{\text{input}}$; the splitting oracle $\mathcal{O}$; baseline vertex $u$; test vertex $v$; $\text{threshold}_1 \in (0, \tilde{n})$, $\text{threshold}_2 \in (0, \tilde{n})$
**Output:** Whether $v$ is a "counterpart" of $u$
Initialize counters $c_1 \leftarrow 0$, $c_2 \leftarrow 0$  **for** *Every vertex* $t \in V_{\text{input}}$ **do**
│  If $v$ splits away from $(u, t)$, increase $c_1$ by 1  If $t$ splits away from $(u, v)$, increase $c_2$ by 1
**end**
**if** $c_1 \leq \text{threshold}_1$ *and* $c_2 \leq \text{threshold}_2$ **then**
│  Return "$v$ is a counterpart of $u$".
**end**

---

Our main algorithm for the strongly consistent partial HC tree construction is as follows.

---

**Algorithm 7** `strong-partial-tree`$^{\mathcal{O}}(\widetilde{V})$: an algorithm for strongly consistent partial tree

---

**Input:** Vertex set $\widetilde{V}$ of size $\tilde{n}$; the splitting oracle $\mathcal{O}$; parameter $\varepsilon < 1$ a sufficiently small constant
**Output:** A partial tree $\mathcal{T}_{\widetilde{V}}$ that is strongly consistent with $\mathcal{T}^*(\widetilde{V})$

**if** $\left|\widetilde{V}\right| \leq 50000 \log n$ **then**
│  Return a super-vertex
**end**
**for** *Each* $u \in \widetilde{V}$ **do**
│  Initialize $T \leftarrow \emptyset$  Sample a set $S$ of $s = 20 \log n / \varepsilon^2$ vertices from $\widetilde{V}$  **for** $v \in \widetilde{V}$ **do**
│  │  Run `counterpart-tester-strong`$^{(\mathcal{O}, S)}(u, v, (3/5 - \varepsilon) \cdot s, (1/6 - \varepsilon) \cdot s)$  Add $v$ to $T$ if $v$ is a "counterpart"
│  │  of $u$
│  **end**
│  Record the size $|T|$ for this choice of $u$
**end**
Pick $(T^*, \widetilde{V} \setminus T^*)$ such that $T^* \leftarrow T$ is with the largest size (breaking ties arbitrarily)  Recursively call

$$\mathcal{T}_{\widetilde{V} \setminus T^*} \leftarrow \texttt{strong-partial-tree}^{\mathcal{O}}(\widetilde{V} \setminus T^*) \qquad \mathcal{T}_{T^*} \leftarrow \texttt{strong-partial-tree}^{\mathcal{O}}(T^*).$$

Connect the two trees with a common ancestor as the root

---

We first observe that the algorithm takes at most $\tilde{O}(n^3)$ queries to $\mathcal{O}$ and $\tilde{O}(n^3/\varepsilon^2)$ time. The formal statement and analysis are as follows.

**Lemma G.1.** *The algorithm* `strong-partial-tree`$^{\mathcal{O}}(V)$ *takes at most* $O(n^3)$ *queries to* $\mathcal{O}$ *and* $\tilde{O}(n^3 \log n / \varepsilon^2)$ *running time.*

*Proof.* The $O(n^3)$ query upper bound is trivial since there are at most $\binom{n}{3}$ such comparisons, and the algorithm can store the answers for reuse. For the running time, we claim that each recursive call on `strong-partial-tree`$^{\mathcal{O}}(\widetilde{V})$ with $\left|\widetilde{V}\right| = \tilde{n}$ vertices takes at most $O(\tilde{n}^2 \log \tilde{n})$ time. To see this, note that each call of Line 7 takes at most $O(\log n / \varepsilon^2)$ time, and there are at most $O(\tilde{n}^2)$ calls of the `counterpart-tester-strong` function. By Fact A.1, there are at most $O(n)$ such recursion calls induced on internal nodes, which results in at most $O(n^3 \log n / \varepsilon^2)$ running time. □

We now prove the correctness of the algorithm. To this end, we first establish the split-away relationships between the vertex set $\widetilde{V}$ and the sampled set $S$.

**Lemma G.2.** *Consider quantities* $c_1$, $c_2$, $\tilde{c}_1$, *and* $\tilde{c}_2$ *obtained by the following processes of running Algorithm 6:*

- *Let $c_1$ and $c_2$ be the counter returned by* `counterpart-tester-strong`$^{(\mathcal{O},S)}(u, v, (3/5 - \varepsilon) \cdot s, (1/6 - \varepsilon) \cdot s)$ *(i.e., by running line 7).*

- *Let $\tilde{c}_1$ and $\tilde{c}_2$ be the counter obtained by running* `counterpart-tester-strong`$^{(\mathcal{O},\widetilde{V})}(u, v, 3\tilde{n}/5, \tilde{n}/6)$.

*Then, with high probability, the following statements are true.*

- *If $\tilde{c}_1 \geq 3\tilde{n}/5$ and $\tilde{c}_2 \geq \tilde{n}/6\cdot$, then $c_1 \geq (3/5 - \varepsilon) \cdot s$ and $c_2 \geq (1/6 - \varepsilon) \cdot s$.*

- *If $\tilde{c}_1 < (3/5 - 2\varepsilon) \cdot \tilde{n}$ and $\tilde{c}_2 < (1/6 - 2\varepsilon) \cdot \tilde{n}$, then $c_1 < (3/5 - \varepsilon) \cdot s$ and $c_2 < (1/6 - \varepsilon) \cdot s$.*

*Proof.* The lemma is a direct application of Chernoff bound, and we prove the quantities for $c_1$ and $\tilde{c}_1$ only since the relationships between $c_2$ and $\tilde{c}_2$ follow from the same argument. For each fixed pair of $(u, v)$ as the inputs to Algorithm 6, let $\tilde{T} \subset \widetilde{V}$ be the set of vertices in $\widetilde{V}$ reporting "$v$ splits away from $(u,t)$". It is clear that $\left|\tilde{T}\right| = \tilde{c}_1$. Furthermore, define $X_t$ as the indicator random variable for the sampled vertex $t \in \tilde{T} \cap S$ and $X$ as the total number of answers reporting "$v$ splits away from $(u,t)$" in $S$. We have $\mathbb{E}[X] = \sum_{x \in S} \Pr(X_t = 1) = \frac{s \cdot \tilde{c}_1}{\tilde{n}}$.

If $\tilde{c}_1 \geq 3\tilde{n}/5$, we have that $\mathbb{E}[X] \geq \frac{3s}{5}$, and $X$ is summation of independent indicator random variables. Therefore, by Chernoff bound we have

$$\Pr\left(X < (\frac{3}{5} - \varepsilon) \cdot s\right) \leq \Pr\left(X - \mathbb{E}[X] < \varepsilon \cdot s\right) \leq \exp\left(-2 \cdot \frac{\varepsilon^2 s^2}{s}\right) \leq \exp\left(-40 \log n\right) \leq \frac{1}{10} \cdot \frac{1}{n^3},$$

where the third inequality is by the choice that $s = 20 \log n/\varepsilon^2$. On the other hand, if $\tilde{c}_1 < (3/5 - 2\varepsilon) \cdot \tilde{n}$, we have that $\mathbb{E}[X] < (3/5 - 2\varepsilon) \cdot s$. Therefore, by Chernoff bound, we have

$$\Pr\left(X \geq \frac{3s}{5}\right) \leq \Pr\left(X - \mathbb{E}[X] \geq \varepsilon \cdot s\right) \leq \exp\left(-2 \cdot \frac{\varepsilon^2 s^2}{s}\right) \leq \exp\left(-40 \log n\right) \leq \frac{1}{10} \cdot \frac{1}{n^3},$$

where the third inequality is by the choice that $s = 20 \log n/\varepsilon^2$. $\qquad\square$

Using Lemma G.2, we now present the following key lemma for the behavior of the "counterpart-test" of Algorithm 7.

**Lemma G.3.** *For any composable $\widetilde{V}$ such that $\left|\widetilde{V}\right| \geq 50000 \log n$, let $\ell^*$ be the maximal level that $V_{\ell^*}^{\mathrm{small}}$ contains a vertex in $V^{\mathrm{small}}(\leq \tilde{n}/5)$. With high probability, Line 7 in Algorithm 7 satisfies the following properties.*

- *a). For any vertex $u \in \cup_{i=\ell^*+1}^{\infty} V_i^{\mathrm{small}}$, there is*

  - *No vertex $v \in \cup_{i=\ell^*+1}^{\infty} V_i^{\mathrm{small}}$ can be added to $T$ by Line 7 of Algorithm 7.*
  - *No vertex $v \in \cup_{i=1}^{\ell^*-1} V_i^{\mathrm{small}}$ can be added to $T$ by Line 7 of Algorithm 7.*

- *b). For every level $\ell \leq \ell^*$ and vertices $u \in V_\ell^{\mathrm{small}}$, with high probability, there is*

  - *No vertex $v \in V_\ell^{\mathrm{small}}$ can be added to $T$ by Line 7 of Algorithm 7.*
  - *No vertex $v \in V_k^{\mathrm{small}}$ for $k < \ell$ can be added to $T$ by Line 7 of Algorithm 7.*

- *c). There exists a $\tilde{\ell} \leq \ell^*$ such that $\left|\cup_{i=1}^{\tilde{\ell}} V_i^{\mathrm{small}}\right| \geq \frac{\tilde{n}}{25}$, and for any $\ell \leq \tilde{\ell}$ and any vertex $u \in V_\ell^{\mathrm{small}}$, with high probability, all vertices $v \in V_k^{\mathrm{small}}$ for $k > \ell$ are added to $T$ by Line 7 of Algorithm 7.*

*Proof.* By Lemma G.2, we only need to prove the split-away properties on $\widetilde{V}$, and the split-away properties on $S$ follows. Let $\ell^*$ be the maximal level that the $V_{\ell^*}^{\mathrm{small}}$ contains a vertex in $V^{\mathrm{small}}(\leq \tilde{n}/5)$, and suppose the split on this level results in $(S_l^{\ell^*}, S_r^{\ell^*})$. We assume w.log. that $\left|S_l^{\ell^*}\right| \leq \left|S_r^{\ell^*}\right|$ so that $S_l^{\ell^*}$ becomes $V_{\ell^*}^{\mathrm{small}}$. We first observe a structural property.

**Observation 3.** *The size of the vertices in $V_H \setminus V^{\mathrm{small}}(\leq \tilde{n}/5)$ (and equivalently $\cup_{i=\ell^*+1}^{\infty} V_i^{\mathrm{small}}$) satisfies*

$$\frac{2\tilde{n}}{5} < \left|\cup_{i=\ell^*+1}^{\infty} V_i^{\mathrm{small}}\right| \leq \frac{4\tilde{n}}{5}.$$

*Proof.* The upper bound follows from $\left|V^{\text{small}}(\leq \tilde{n}/5)\right| \geq \tilde{n}/5$, as otherwise $V^{\text{small}}(\leq \tilde{n}/5)$ will not have enough vertices. For the lower bound, note that by level $\ell^* - 1$, the size there are still *more than* $\frac{4\tilde{n}}{5}$ vertices remain to be decided. Also, note that $\cup_{\ell^*+1}^{\infty} V_i^{\text{small}}$ collectively forms the 'larger subtree' w.r.t. $V_{\ell^*}^{\text{small}}$. As such, its size has to be *more than* $\frac{1}{2} \cdot \frac{4\tilde{n}}{5} = \frac{2\tilde{n}}{5}$. $\qquad\square$

Note that by the definition of small trees, we also have $\left|V_\ell^{\text{small}}\right| \leq \frac{\tilde{n}}{2}$ for any $\ell$. We now proceed one-by-one to the proofs of Item a)., Item b)., and Item c)..

**Proof of Item a).** Fix any vertex $u \in \cup_{i=\ell^*+1}^{\infty} V_i^{\text{small}}$. For the first statement, note that for every $v \in \cup_{i=\ell^*+1}^{\infty} V_i^{\text{small}}$, all the vertices in $\cup_{i=1}^{\ell^*} V_i^{\text{small}}$ are *splitting away* from $(u, v)$. Define $X_{u,v,t}$ as the indicator random variable answered by $\mathcal{O}$ as "$t$ splits away from $(u, v)$", and define $X_{u,v} = \sum_{t \in \tilde{V}} X_{u,v,t}$ as the total number of vertex split away from $(u, v)$. By Observation 3, the expected number of split away reported by oracle $\mathcal{O}$ is at least

$$\mathbb{E}\left[X_{u,v}\right] \geq \frac{9}{10} \cdot \frac{\tilde{n}}{5} = \frac{9}{50} \cdot \tilde{n}.$$

Note also that $X_{u,v}$ is a summation of $0/1$ independent random variables. As such, we can apply Chernoff bound to get that

$$\Pr\left(X_{u,v} \leq \frac{\tilde{n}}{6}\right) = \Pr\left(X_{u,v} \leq \frac{25}{27} \cdot \mathbb{E}\left[X_{u,v}\right]\right)$$
$$\leq \exp\left(-\frac{(2/27)^2}{3} \cdot \mathbb{E}\left[X_{u,v}\right]\right)$$
$$\leq \frac{1}{10} \cdot \frac{1}{n^3}. \qquad (\frac{9}{50} \cdot \tilde{n} \geq 9000 \log n \text{ using the lower bound on the size of } \widetilde{V})$$

For the second statement, note that for any $t \in \cup_{i=\ell^*}^{\infty} V_i^{\text{small}}$, $v$ actually splits away from $(u, t)$. As such, there is at least $\frac{4\tilde{n}}{5}$ vertices such that "$v$ splits away from $(u, t)$" – this is because the LCA between $(u, v)$ has to be higher than the LCA between $(u, t)$ for any $t \in \cup_{i=\ell^*}^{\infty} V_i^{\text{small}}$. As such, define $Y_{u,v}$ as the number of total answers of "$v$ splits away from $(u, t)$" by $\mathcal{O}$, we again have

$$\mathbb{E}\left[Y_{u,v}\right] \geq \frac{9}{10} \cdot \frac{4\tilde{n}}{5} = \frac{18}{25} \cdot \tilde{n}.$$

As such, we again have

$$\Pr\left(Y_{u,v} \leq \frac{3\tilde{n}}{5}\right) = Pr\left(Y_{u,v} \leq \frac{15}{18} \cdot \mathbb{E}\left[X_{u,v}\right]\right)$$
$$\leq \frac{1}{10} \cdot \frac{1}{n^3}. \qquad (\frac{9}{50} \cdot \tilde{n} \geq 9000 \log n \text{ using the lower bound on the size of } \widetilde{V})$$

By a union bound over at most $n$ vertices, no vertex $v \in \cup_{i=1}^{\ell^*-1} V_i^{\text{small}}$ or $v \in \cup_{i=\ell^*+1}^{\infty} V_i^{\text{small}}$ can pass the test and be recorded as a 'counterpart'. Furthermore, by Lemma G.2, if $X_{u,v} > \tilde{n}/6$ and $Y_{u,v} > 3\tilde{n}/5$, then such $u$ cannot pass the test on the sampled set $S$, which is as desired. This part of the proof can be visualized as in Figure 8(a).

**Proof of Item b)..** We now show that vertices $v \in \cup_{i=1}^{\ell} V_i^{\text{small}}$ cannot pass the "counterpart" test. For vertices in $v \in V_\ell^{\text{small}}$, we claim that there are too many vertices $t$ that split away from $(u, v)$. To see this, let us define $A_{u,v}$ as the total number of answers of "$t$ splits away from $(u, v)$" answered by $\mathcal{O}$. Note that for any vertex $t \in \cup_{i=\ell^*+1}^{\infty} V_i^{\text{small}}$, the answer is always "yes" since $\ell^* \geq \tilde{\ell} \geq \ell$. Therefore, we can lower bound the expectation of $A_{u,v}$ with $\mathbb{E}\left[A_{u,v}\right] \geq \frac{2}{5}\tilde{n} \cdot \frac{9}{10} \geq \frac{9}{25} \cdot \tilde{n}$. Therefore, we can apply Chernoff bound to get

$$\Pr\left(A_{u,v} \leq \frac{\tilde{n}}{6}\right) \leq \exp\left(-\frac{(1/2)^2}{3} \cdot \mathbb{E}\left[A_{u,v}\right]\right) \leq \frac{1}{10} \cdot \frac{1}{n^3}.$$

We now turn to the vertices that are in $\cup_{i=1}^{\ell-1} V_i^{\text{small}}$. Note that by definition, there is $\left|\cup_{i=\ell}^{\infty} V_i^{\text{small}}\right| \geq \left|\cup_{i=\ell^*}^{\infty} V_i^{\text{small}}\right| \geq \frac{4}{5} \cdot \tilde{n}$. For any $t \in \cup_{i=\ell^*}^{\infty} V_i^{\text{small}}$, $v$ splits away from $(u, t)$. As such, define $B_{u,v}$ as the number of answers by $\mathcal{O}$ with "$v$ splits away

from $(u, t)$", there is $\mathbb{E}\left[B_{u,v}\right] \geq \frac{18}{25} \cdot \tilde{n}$. Once again, by applying Chernoff bound, there is

$$\Pr\left(B_{u,v} \leq \frac{3\tilde{n}}{5}\right) \leq \exp\left(-\frac{(1/5)^2}{3} \cdot \mathbb{E}\left[B_{u,v}\right]\right) \leq \frac{1}{10} \cdot \frac{1}{n^3}.$$

Applying a union bound over the cases and all $\tilde{n} \leq n$ vertices and using Lemma G.2 would lead to the desired statements. This part of the proof can be visualized as in Figure 8(b).

**Proof of Item c)..** We fix $\tilde{\ell}$ as the minimum level such that the union of $\cup_{i=1}^{\tilde{\ell}} V_i^{\text{small}}$ has at least $\tilde{n}/25$ vertices. Let $\ell \leq \tilde{\ell}$, and for any vertex $v$ in $V_k^{\text{small}}$ for $k > \ell$, define the following two random variables: $X_{u,v}$ as the number of answers that "$t$ splits away from $(u, v)$", and $Y_{u,v}$ as the number of answers that "$v$ splits away from $(u, t)$". We shall show that both terms are not large and $v$ can always pass the test. (Note that $X$ and $Y$ are overloaded and are *not* related to their meaning in the proof of Line b)..)

Note that by definition, we have $\left|\cup_{i=1}^{\tilde{\ell}-1} V_i^{\text{small}}\right| < \frac{\tilde{n}}{25}$. Since $\ell \leq \tilde{\ell}$, for any vertex $u \in \cup_{i=1}^{\ell} V_i^{\text{small}}$, only vertices $t \in \cup_{i=1}^{\ell-1} V_i^{\text{small}}$ are *actually* splitting away from $(u, v)$ for $v \in \cup_{i=\ell}^{\infty} V_i^{\text{small}}$. As such, we have $\mathbb{E}\left[X_{u,v}\right] \leq (\frac{1}{10} + \frac{1}{25}) \cdot \tilde{n} = \frac{7}{50} \cdot \tilde{n}$ by the correct probability of $\mathcal{O}$. For $X_{u,v}$, we also note that if the answer is at most $(30000 - 100000\varepsilon) \cdot \log n$, then by Lemma G.2, the vertex passes the test with high probability. Therefore, we assume $X_{u,v} \geq \mathbb{E}\left[X_{u,v}\right] \geq (8000 - 100000\varepsilon) \cdot \log n$, and we again apply Chernoff for the tail bound:

$$\Pr\left(X_{u,v} \geq (1/6 - 2\varepsilon) \cdot \tilde{n}\right) \leq \exp\left(-\frac{50 \cdot (2/75 - 2\varepsilon)^2}{7 \cdot 3} \cdot \mathbb{E}\left[Y_{u,v}\right]\right)$$

$$\leq \frac{1}{10} \cdot \frac{1}{n^3}. \qquad \text{(using } \mathbb{E}\left[Y_{u,v}\right] \geq (8000 - 100000\varepsilon) \cdot \log n \text{ and pick } \varepsilon \text{ sufficiently small)}$$

For $Y_{u,v}$, note that only $t \in V_{\tilde{\ell}}^{\text{small}}$ can report "$v$ splits away from $(u, t)$" since the LCA between $(u, t)$ is at least $\tilde{\ell}$ for any other $v$. Therefore, the number of signals we can possibly get is $\tilde{n}/2$ plus the noise induced by the oracle $\mathcal{O}$. As such, we again have $\mathbb{E}\left[Y_{u,v}\right] \leq 11\tilde{n}/20$. Also, note that if $Y_{u,v}$ is less than $(30000 - 100000\varepsilon) \cdot \log n$, the vertex $v$ would always pass the test, which allows us to assume w.log. that $\mathbb{E}\left[Y_{u,v}\right] \geq (30000 - 100000\varepsilon) \cdot \log n$. Therefore, we can again apply Chernoff bound to show

$$\Pr\left(Y_{u,v} \geq (\frac{3}{5} - 2\varepsilon) \cdot \tilde{n}\right) \leq \exp\left(-\frac{20 \cdot (1/20 - 2\varepsilon)^2}{3} \cdot \mathbb{E}\left[Y_{u,v}\right]\right)$$

$$\leq \frac{1}{10} \cdot \frac{1}{n^3}. \qquad \text{(using } \mathbb{E}\left[Z_{u,v}\right] \geq (30000 - 100000\varepsilon) \cdot \log n \text{ and pick } \varepsilon \text{ sufficiently small)}$$

Therefore, we could apply Lemma G.2 to argue that with high probability, the vertex $v$ would pass the test. This part of the proof can be visualized as in Figure 8(c).

Combining the analysis of Item a)., Item b)., and Item c). gives us the desired proof of Lemma G.3. $\qquad \square$

Using Lemma G.3, we can now argue that with high probability, the algorithm always correctly identifies the vertex split from the root as long as the size is at least $50000 \log n$. More formally, we have

**Lemma G.4.** *For any composable $\widetilde{V}$ such that $\left|\widetilde{V}\right| \geq 50000 \log n$, let $\mathcal{T}^*(\widetilde{V})$ be the optimal HC tree restricted to $\widetilde{V}$, and suppose the root split of $\mathcal{T}^*(\widetilde{V})$ is $(S_l^*, S_r^*)$. Then, the output $(T^*, \widetilde{V} \setminus T^*)$ of Line 7 is exactly $(S_l^*, S_r^*)$.*

*Proof.* Assume without the loss the generality that $S_l^* \leq S_r^*$. Note that for any vertex $u \in S_l^*$, it is also among the vertex of $V^{\text{small}}(\leq \tilde{n}/25)$. As such, by Lemma G.3, the entire vertex set of $S_l^*$ is returned. On the other hand, for any vertex $u \in S_r^*$, we claim the induced set $T$ is necessarily smaller than $S_r^*$. To see this, note that by Lemma G.3, the only possible case for Line 7 to *not* return a *subset* of $S_r^*$ is if $u \in V_\ell^{\text{small}}$ such that $\ell > \ell^*$ and $\ell^* = 1$. However, in such a case, the algorithm can at most return the set $S_l^*$, which is necessarily smaller than the size of $S_r^*$. As such, the maximum size of the set is attained by picking the vertex from $S_l^*$, which implies that the return rule of Line 7 gives the partition $(S_l^*, S_r^*)$. $\qquad \square$

In essence, Lemma G.4 is the very natural consequence of Lemma G.3. This can be visualized in Figure 8(d) of Figure 8 – the $T$ with the largest size is always the set of vertices induced by $u_1$, which is exactly what we want.

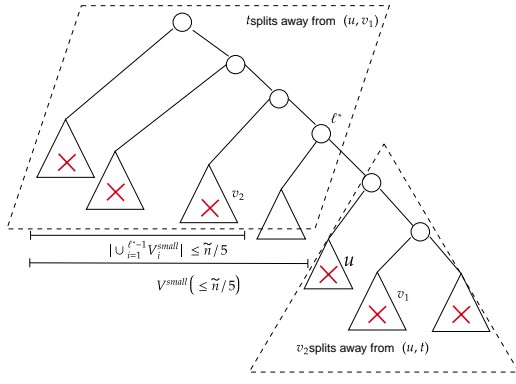

(a) Illustration of Item a). of Lemma G.3 with $u \in \cup_{i=\ell^*+1}^{\infty} V_i^{\text{small}}$.

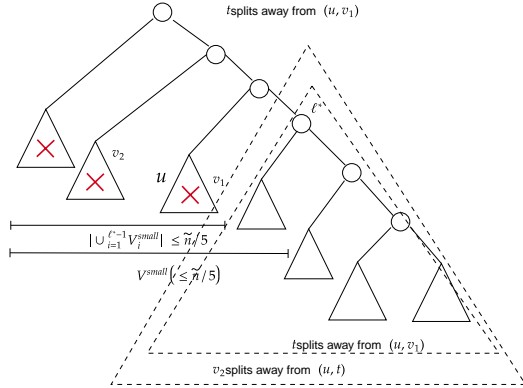

(b) Illustration of Item b). of Lemma G.3 with $u \in V_\ell^{\text{small}}$ for some $\ell \leq \ell^*$.

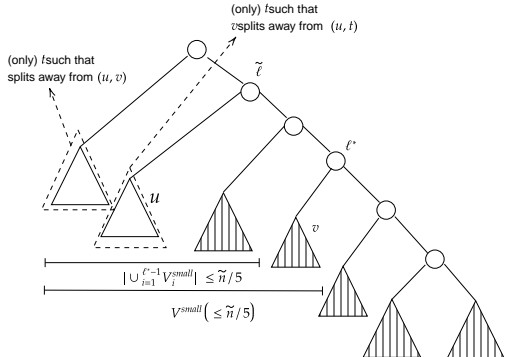

(c) Illustration of Item c). of Lemma G.3 with $u \in V_\ell^{\text{small}}$ for $\ell \leq \tilde{\ell}$ as prescribed in Lemma G.3.

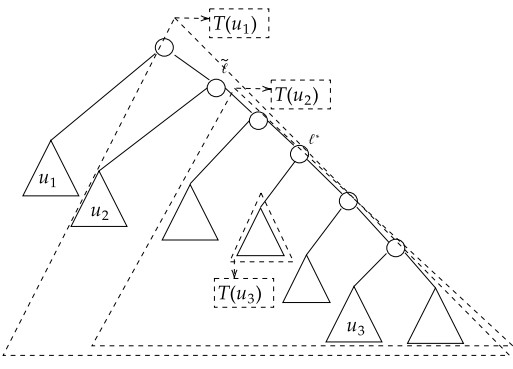

(d) The "counterpart" trees generated by vertices $u_1$, $u_2$, and $u_3$. We can simply take the largest partition to guarantee the root cut.

Figure 8. An illustration of the analysis we used in the proof of Lemma G.3.

Conditioning on the high probability event, any super-vertex has to induce a maximal subtree in $\mathcal{T}^*$, which satisfies the requirement for the tree to be *strong consistent* with $\mathcal{T}^*$. By Lemma G.1, Algorithm 7 deterministically takes at most $O(n^3)$ queries and $O(n^3 \log n / \varepsilon^2) = O(n^3 \log n)$ time by the choice of $\varepsilon = \Theta(1)$.

Finally, we observe that Algorithm 7 only takes $O(n \log n)$ memory to implement. To see this, observe that for any recursive call of Algorithm 7, for any fixed $u$ and $v$, the subroutine `counterpart-tester-strong` can be implemented in $O(n \log n)$ bits of space; furthermore, the space can be re-used for different $u$ and $v$ vertices. As such, we only need to maintain the set $T$ and its size for every $u$, which takes $O(n \log^2 n)$ space. In fact, we can maintain this in $O(n \log n)$ space by keeping the set $T$ with the largest size and re-using the space on the fly. After each recursive call, we can free up the space of $\widetilde{V}$ to store $\widetilde{V} \setminus T^*$ and $T^*$ instead. On a separate $O(n \log n)$-sized space, we can keep writing down the HC tree with

i).    A name of the node that takes $O(\log n)$ bits of memory;

ii).   The set of vertices induced by this node.

In each recursion, we re-use the space of $ii)$ by erasing the set of vertices induced by an internal node $x$ once the children of $x$ are written down. In the end, the sets of vertices are only written on the super-vertices. As such, the entire partial HC tree can be stored in $O(n \log n)$ memory as well.

## H. The Algorithm for the Weakly Consistent Partial HC Tree: Proof of Theorem 11

We present the considerably more involved algorithm for the weakly consistent partial tree construction in *near-linear* time. We first remind the readers of the main theorem of this result.

**Theorem 11.** *There exists an algorithm that given a splitting oracle $\mathcal{O}$ of a weighted undirected graph $G = (V, E, w)$, with high probability, in $O(n^2 \cdot \text{polylog } n)$ time and $O(n^2)$ queries computes a partial hierarchical clustering tree $\mathcal{I}$ that is weakly consistent with the optimal hierarchical clustering tree $\mathcal{T}^*$.*

Our main effort is to show the algorithm that satisfies the guarantee of Theorem 11. As we have discussed in the high-level overview, the algorithm is divided into the "split" and the "merging" parts. We first show an algorithm that given a composable subset of vertices $\widetilde{V} \subseteq V$ such that $\left| \widetilde{V} \right| \geq \Omega(\log n)$, return a set $T$ that induces a *maximal subtree* in $\mathcal{T}(\widetilde{V})$. As such, we can recurse on $T$ and $\widetilde{V} \setminus T$ to gradually reduce the sizes to at most $O(\log n)$. Then, in the second part, we show how to 'glue' the subtrees together to eventually form a partial tree that is consistent with $\mathcal{T}^*$. The overall structure of the algorithm can be shown as Algorithm 8.

---

**Algorithm 8** `weak-partial-tree`$^{\mathcal{O}}(\widetilde{V}, V_{\mathrm{H}})$: an algorithm for partial tree construction

---

**Input:** Input vertex set $\widetilde{V}$; input horizon vertex set $V_{\mathrm{H}}$; a splitting oracle $\mathcal{O}$ as prescribed in Definition 6
**Output:** A partial hierarchical clustering tree for $\widetilde{V}$.
Initialize $\widetilde{V} = V$, $V_{\mathrm{H}} = V$. If $\left| \widetilde{V} \right| < 50000 \log n$, return a super-vertex (defined as in Definition 7). Get the partition of $T$ such that $\mathcal{T}_T$ is a complete subtree, i.e. $(T, \widetilde{V} \setminus T, u^*, V_{\mathrm{H}}) \leftarrow \texttt{vertex-split}^{\mathcal{O}}(\widetilde{V}, V_{\mathrm{H}})$ (Algorithm 12) Recurse on $T$ and $\widetilde{V} \setminus T$, i.e.,

$$\mathcal{T}_T \leftarrow \texttt{weak-partial-tree}^{\mathcal{O}}(T, T); \qquad \mathcal{T}_{\widetilde{V} \setminus T} \leftarrow \texttt{weak-partial-tree}^{\mathcal{O}}(\widetilde{V} \setminus T, V_{\mathrm{H}})$$

for the respective HC trees  Run the merging algorithm $\mathcal{T}_{\widetilde{V}} \leftarrow \texttt{tree-merge}^{\mathcal{O}}(\mathcal{T}_T, \mathcal{T}_{\widetilde{V} \setminus T}, u^*)$ (Algorithm 13).

---

We will introduce and analyze the splitting and the merging algorithms in the subsequent sections.

### H.1. An algorithm to split the vertices

We first discuss our algorithm that produces complete subtrees for a given set of vertices $\widetilde{V}$. For the clarity of presentation, we use $\tilde{n}$ to denote the size of $\widetilde{V}$, i.e., $\tilde{n} = \left| \widetilde{V} \right|$.

### H.1.1. THE ALGORITHM

We now present the actual algorithm. We first give two "tester" subroutines that serves as a more general version of the `counterpart-tester-strong` algorithm we used in Algorithm 7.

---

**Algorithm 9** `counterpart-tester`$^{(\mathcal{O}, V_{\mathrm{H}})}(u, t, \mathrm{threshold}_1, \mathrm{threshold}_2)$: an algorithm to test whether $t$ is among the "counterpart" of $u$

---

**Input:** Vertex set $V_{\mathrm{H}}$; a splitting oracle $\mathcal{O}$; baseline vertex $u$; test vertex $v$; $\mathrm{threshold}_1 \in (0, \tilde{n})$, $\mathrm{threshold}_2 \in (0, \tilde{n})$
**Output:** Whether $v$ is a "counterpart" of $u$
Initialize counters $c_1 \leftarrow 0$, $c_2 \leftarrow 0$ **for** *Every vertex $t \in V_H$* **do**
 | If $v$ splits away from $(u, t)$, increase $c_1$ by 1  If $t$ splits away from $(u, v)$, increase $c_2$ by 1
**end**
**if** $c_1 \leq \mathrm{threshold}_1$ *and* $c_2 \leq \mathrm{threshold}_2$ **then**
 | Return "$v$ is a counterpart of $u$".
**end**

---

Algorithm 9 is very similar to Algorithm 6, and the difference is that instead of testing on $\widetilde{V}$ set, we query on the $V_{\mathrm{H}}$ set instead. This is more than a simple notation change: it allows us to separate the vertices "to be tested" (e.g., vertices in $\widetilde{V}$) vs. the set we "used in the tests" (e.g., vertices in $V_{\mathrm{H}}$).

Next, we introduce the tester that given vertices $u \in V_{\ell_1}^{\mathrm{small}}$ and $t \in V_{\ell_2}^{\mathrm{small}}$, returns whether $\ell_2 < \ell_1$, i.e., whether $t$ is "split earlier" in the small-tree split order of $u$.

---

**Algorithm 10** `predecessor-tester`$^{(\mathcal{O}, V_{\mathrm{H}})}(u, t, \mathrm{threshold})$: an algorithm to test whether $t$ is among the "predecessor" of $u$

---

**Input:** Vertex set $V_{\mathrm{H}}$; a splitting oracle $\mathcal{O}$; baseline vertex $u$; test vertex $t$; $\mathrm{threshold} \in (0, \tilde{n})$
**Output:** Whether $t$ is a "predecessor" of $u$
Initialize counters $c \leftarrow 0$  **for** *Every vertex $s \in V_H$* **do**
 | If $t$ splits away from $(u, s)$, increase $c$ by 1
**end**
**if** $c \geq \mathrm{threshold}$ **then**
 | Return "$t$ is a predecessor of $u$".
**end**

---

Note that unlike Algorithm 9, the vertex $t$ passes the test in Algorithm 10 if the number of "split away" is lower-bounded. With the `predecessor-tester`$^{(\mathcal{O}, V_{\mathrm{H}})}(u, t, \mathrm{threshold})$ algorithm, we can now define the following algorithm that tests whether a root split has happened in the previous iteration.

We now continue to the presentation of our tree-split algorithm. The full algorithm is as Algorithm 12.

---

**Algorithm 11** `test-orphan-predecessor`$^{(\mathcal{O},V_{\mathrm{H}},\widetilde{V})}$: an algorithm to obtain any vertex that is split earlier than the orphaned vertex set in the small-tree split order

---

**Input:** Vertex set $\widetilde{V}$ of size $\tilde{n}$; the horizon set $V_{\mathrm{H}}$ of size $n_H$; a splitting oracle $\mathcal{O}$
**Output:** Vertex $X^*$ that is either empty or contain "predecessor" of orphaned sets
Sample a set $U'$ of $100\log n$ vertices from $\widetilde{V}$ using *fresh randomness* **for** $u' \in U'$ **do**

> **for** $v \in V_H$ **do**
>
>> **if** $n_H \leq 3\tilde{n}$ **then**
>>> | threshold-pred $\leftarrow 3\tilde{n}/2$
>>
>> **end**
>> **else**
>>> | threshold-pred $\leftarrow 2n_H/3$
>>
>> **end**
>> Add $x$ to $X$ if `predecessor-tester`$^{(\mathcal{O},V_{\mathrm{H}})}(u,\,v,\,\text{threshold-pred})$ (Algorithm 10) reports "$v$ is a predecessor of $u$"
>
> **end**
> Record the size $|X|$ for this choice of $u'$

**end**
Return $X^*$ as the largest size of $X$

---

**Algorithm 12** `vertex-split`$^{\mathcal{O}}(\widetilde{V}, V_{\mathrm{H}})$: an algorithm that splits $\widetilde{V}$ to $T$ and $\widetilde{V} \setminus T$

---

**Input:** Vertex set $\widetilde{V}$ of size $\tilde{n}$; the horizon set $V_{\mathrm{H}}$ of size $n_H$; a splitting oracle $\mathcal{O}$
**Output:** Vertex sets $T$, $\widetilde{V} \setminus T$; new horizon set $V_{\mathrm{H}}$
Sample a set $U$ of $500\log n$ vertices from $\widetilde{V}$ **for** $u \in U$ **do**

> Initialize $T \leftarrow \emptyset$ **for** $v \in V_H$ **do**
>> | Run `counterpart-tester`$^{(\mathcal{O},V_{\mathrm{H}})}(u,\,v,\,3n_H/5,\,n_H/6)$ Add $v$ to $T$ if $v$ is a "counterpart" of $u$
>
> **end**
> Record the size $|T|$ for this choice of $u$

**end**
Pick $(T^*, \widetilde{V} \setminus T^*)$ such that $T^* \leftarrow T$ is with the largest size **if** $T^* \cap \widetilde{V} = \emptyset$ **then**

> // Test whether a root split happened in the last iteration $X^* \leftarrow$ `test-orphan-predecessor`$^{(\mathcal{O},V_{\mathrm{H}},\widetilde{V})}$ (using Algorithm 11) **if** $X^* \neq \emptyset$ **then**
>
>> // The case of non-root split Pick an arbitrary $u^* \in X^*$ Initialize $T^* \leftarrow \emptyset$ **for** $v \in V_H$ **do**
>>> | Run `counterpart-tester`$^{(\mathcal{O},V_{\mathrm{H}})}(u^*,\,v,\,3n_H/5,\,n_H/6)$ Add $v$ to $T^*$ if $v$ is a "counterpart" of $u^*$
>>
>> **end**
>> Output with the same rules of the $T^* \cap \widetilde{V} \neq \emptyset$ case
>
> **end**
> **else**
>> // Case for root split Let $V_{\mathrm{H}} \leftarrow \widetilde{V}$, run and output with `vertex-split`$^{\mathcal{O}}(\widetilde{V}, \widetilde{V})$ Enforce a *single level* of recursion call (return "FAIL" if $\widetilde{V} = V_{\mathrm{H}}$ and the algorithm enters the above line again with $\widetilde{V} = V_{\mathrm{H}}$)
>
> **end**

**end**
**else**

> // Keep the current horizon Output $(T^* \cap \widetilde{V}, \widetilde{V} \setminus T^*)$ as the partition and keep the same $V_{\mathrm{H}}$ Output $u^*$ as the vertex $u \in U$ corresponding to the output $T^*$

**end**

---

Note that in Line 12, we sample $500 \log n$ as opposed to $500 \log \tilde{n}$ vertices (this is on purpose as opposed to being a typo). The formal guarantee of the tree split algorithm is as follows.

**Lemma H.1.** *Suppose* $\widetilde{V}$ *is a composable set with size* $\tilde{n}$ *that satisfies the* weak contraction property *as prescribed in Definition 9, and suppose* $\tilde{n} \geq 50000 \log n$. *Furthermore, let* $V_H$ *be a single composable set (e.g.,* $V_H$ *induces a maximal tree in* $\mathcal{T}^*$*) of* $n_H$ *vertices, and suppose*

$$\widetilde{V} = \cup_{i=1}^{\ell} V_i^{\text{small}}$$

*for some* $\ell$ *in the small-tree split order of* $V_H$. *Then, given a splitting oracle* $\mathcal{O}$ *with correct probability at least* $9/10$, *Algorithm 12 with high probability outputs composable sets* $T^* \cap \widetilde{V}$ *and* $\widetilde{V} \setminus T^*$ *and a vertex* $u^*$ *such that*

1. $T^*$ *induces a* single *maximal subtree in* $\mathcal{T}^*(\widetilde{V})$.

2. $\widetilde{V} \setminus T^*$ *satisfies the* weak contraction property *as prescribed in Definition 9, and in particular, the subtrees induced by* $\widetilde{V} \setminus T^*$ *have*

   (a) *one edge connected to the node* $\text{pa}\left(\text{LCA}_{\mathcal{T}^*}(\widetilde{V})\right)$ *(if it exist).*

   (b) *one edge connected to the node that induces* $T^*$ *in* $\mathcal{T}^*$.

3. *The lowest common ancestor between the nodes in* $T$ *is an (immediate) child node of the lowest common ancestor between* $T \cup \{u^*\}$ *in* $\mathcal{T}^*(\widetilde{V})$, *i.e.,*

   $$\text{LCA}_{\mathcal{T}^*(\widetilde{V})}(T \cup \{u^*\}) = \text{pa}\left(\text{LCA}_{\mathcal{T}^*(\widetilde{V})}(T)\right).$$

4. *Size properties: at least one* of the following guarantees hold.

   (a) *The new set* $\widetilde{V} \setminus T^*$ *has at least* $\frac{99}{100}$ *fraction of vertices that are orphaned vertices, i.e., the new* $V^{\text{orphan}}$ *set accounts of at least* $\frac{99}{100}$ *fraction of vertices in* $\widetilde{V} \setminus T^*$;

   (b) *The size of* $T^*$ *satisfies the following properties:*

       i.  *The size of* $T^* \cap \widetilde{V}$ *is at least* $\frac{1}{200} \cdot \tilde{n}$, *i.e.,*

   $$\left| T^* \cap \widetilde{V} \right| \geq \frac{1}{200} \cdot \tilde{n}.$$

       ii.  *If* $V_H = \widetilde{V}$, *the size of* $T^* \cap \widetilde{V}$ *is at most* $(1 - \frac{1}{10000 \log^2 \tilde{n}}) \cdot \tilde{n}$, *i.e.,*

   $$\left| T^* \cap \widetilde{V} \right| \leq (1 - \frac{1}{10000 \log^2 \tilde{n}}) \cdot \tilde{n}.$$

*Furthermore, case Item 4b always happens if* $V_H = \widetilde{V}$, *and the algorithm runs in time* $O(n_H^2 \cdot \log n)$.

We now proceed to the analysis to prove Lemma H.1.

THE ANALYSIS

To begin with, we define the notion of *orphaned vertices* from our split procedure.

**Definition 28** (Orphaned vertices)**.** *Let* $\widetilde{V}$ *be a composable set of vertices, and let* $T = \cup_{i=\ell+1}^{\infty} V_i^{\text{small}}$. *We call* $V_\ell^{\text{small}} \subseteq \widetilde{V}$ *the set of orphaned vertices in* $V_\ell^{\text{small}}$ *with respect to* $T$. *We denote the orphaned set of vertices as* $V_{T,\ell}^{\text{orphan}}$, *and we write* $V^{\text{orphan}}$ *as the simplified notation when the context is clear.*

One can refer to Figure 9 for a visualization of the orphaned vertices (we used $V$ as opposed to $\widetilde{V}$ in the figure). When the context is clear, we ignore the dependence on $V_\ell^{\text{small}}$ and $T$ when talking about orphaned vertices, and simply denote the orphaned vertices as $V^{\text{orphan}}$. In our proof of correctness, we will show "inductively" that the set $\widetilde{V}$ only has a *single* orphaned set $V^{\text{orphan}}$ – a key to guarantee property Item 2 of Lemma H.1.

We now proceed with the relatively straightforward analysis of the running time of a *single* level of recursion.

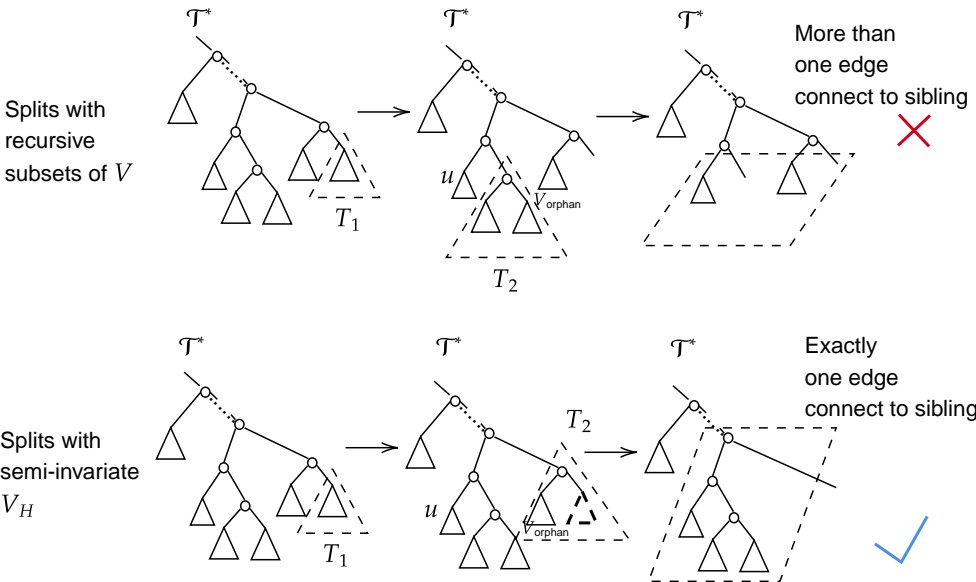

*Figure 9.* An illustration of how horizon sets help maintain weak consistency. With the same $u$, the reason for the different outcomes is that on the top, the small-tree split order *restricted on V* has changed compared to the last iteration. The use of the horizon set helps keep the small-tree split order *invariant* before a root cut happens.

**Lemma H.2.** *The algorithm* `vertex-split`$^{\mathcal{O}}(\widetilde{V}, V_H)$ *runs in time* $O(n_H^2 \cdot \log n)$.

*Proof.* Each run of the `counterpart-tester` takes $O(n_H)$ time. As such, the procedure that finds $u^*$ takes $O(n_H^2 \cdot \log n)$ time since it involves $O(n_H \log n)$ calls of `counterpart-tester`. Similarly, the procedure that finds $X^*$ takes at most $O(n_H \log n)$ calls of `predecessor-tester`, which again result in $O(n_H^2 \cdot \log \tilde{n})$ time. Furthermore, if $X^* \neq \emptyset$, we only make $n_H$ number of `counterpart-tester` calls, which takes $O(n_H^2)$ time. On the other hand, if $X^* = \emptyset$, since we insist on a single level of recursion call, the runtime overhead is at most $O(n_H^2 \cdot \log \tilde{n})$. Adding up the time complexity of the above two procedures gives us the desired bound. $\square$

We proceed with the proof of correctness for the algorithm. To this end, we first observe that by Observation 2 and since $V_H$ is composable, the answers for the splitting oracle $\mathcal{O}$ on $(u, v, w) \in V_H$ is fully preserved in $\mathcal{T}^*(V_H)$. We now give a technical lemma that characterizes the return set $T$ by running `counterpart-tester` on different sets of vertices – this is essentially the same argument we used in Lemma G.3, albeit we switched $\widetilde{V}$ to $V_H$. We provide the lemma and the analysis for the purpose of completeness.

**Lemma H.3** (Cf. Lemma G.3). *For any composable $V_H$ such that $|V_H| \geq 50000 \log n$, let $\ell^*$ be the maximal level that $V_{\ell^*}^{\mathrm{small}}$ contains a vertex in $V^{\mathrm{small}}(\leq n_H/5)$. With high probability, Line 12 in Algorithm 12 satisfies the following properties.*

a). *For any vertex $u \in \cup_{i=\ell^*+1}^{\infty} V_i^{\mathrm{small}}$, there are*

- *No vertex $v \in \cup_{i=\ell^*+1}^{\infty} V_i^{\mathrm{small}}$ can be added to $T$ by Line 12 of Algorithm 12.*
- *No vertex $v \in \cup_{i=1}^{\ell^*-1} V_i^{\mathrm{small}}$ can be added to $T$ by Line 12 of Algorithm 12.*

b). *For every level $\ell \leq \ell^*$ and vertices $u \in V_\ell^{\mathrm{small}}$, with high probability, there are*

- *No vertex $v \in V_\ell^{\mathrm{small}}$ can be added to $T$ by Line 12 of Algorithm 12.*
- *No vertex $v \in V_k^{\mathrm{small}}$ for $k < \ell$ can be added to $T$ by Line 12 of Algorithm 12.*

c). *There exists a $\tilde{\ell}$ such that $\left|\cup_{i=1}^{\tilde{\ell}} V_i^{\mathrm{small}}\right| \geq \frac{n_H}{25}$, and for any $\ell \leq \tilde{\ell}$ and any vertex $u \in V_\ell^{\mathrm{small}}$, with high probability, all vertices $v \in V_k^{\mathrm{small}}$ for $k > \ell$ are added to T by Line 12 of Algorithm 12.*

*Proof.* Let $\ell^*$ be the maximal level that the $V_{\ell^*}^{\mathrm{small}}$ contains a vertex in $V^{\mathrm{small}}(\leq n_H/5)$, and suppose the split on this level results in $(S_l^{\ell^*}, S_r^{\ell^*})$. We again assume w.log. that $\left|S_l^{\ell^*}\right| \leq \left|S_r^{\ell^*}\right|$, which means $S_l^{\ell^*}$ becomes $V_{\ell^*}^{\mathrm{small}}$. The bounds in Observation 3 still hold with parameter $n_H$, i.e., $\frac{2n_H}{5} < \left|\cup_{i=\ell^*+1}^{\infty} V_i^{\mathrm{small}}\right| \leq \frac{4n_H}{5}$. We now proceed to the proofs of Item a)., Item b)., and Item c)., respectively.

**Proof of Item a).** Fix any vertex $u \in \cup_{i=\ell^*+1}^{\infty} V_i^{\mathrm{small}}$. For the first statement, note that for every $v \in \cup_{i=\ell^*+1}^{\infty} V_i^{\mathrm{small}}$, all the vertices in $\cup_{i=1}^{\ell^*} V_i^{\mathrm{small}}$ are *splitting away* from $(u, v)$. Define $X_{u,v,t}$ as the indicator random variable answered by $\mathcal{O}$ as "$t$ splits away from $(u, v)$", and define $X_{u,v} = \sum_{t \in \widetilde{V}} X_{u,v,t}$ as the total number of vertex split away from $(u, v)$. By Observation 3, the expected number of split away reported by oracle $\mathcal{O}$ is at least

$$\mathbb{E}\left[X_{u,v}\right] \geq \frac{9}{10} \cdot \frac{n_H}{5} = \frac{9}{50} \cdot n_H.$$

Note also that $X_{u,v}$ is a summation of $0/1$ independent random variables. As such, we can apply Chernoff bound to get that

$$\Pr\left(X_{u,v} \leq \frac{n_H}{6}\right) = \Pr\left(X_{u,v} \leq \frac{25}{27} \cdot \mathbb{E}\left[X_{u,v}\right]\right)$$
$$\leq \exp\left(-\frac{(2/27)^2}{3} \cdot \mathbb{E}\left[X_{u,v}\right]\right)$$
$$\leq \frac{1}{10} \cdot \frac{1}{n^3}. \qquad (\frac{9}{50} \cdot n_H \geq 9000 \log n \text{ using the lower bound on the size of } V_H)$$

For the second statement, note that for any $t \in \cup_{i=\ell^*}^{\infty} V_i^{\mathrm{small}}$, $v$ actually splits away from $(u, t)$. As such, there is at least $\frac{4n_H}{5}$ vertices such that "$v$ splits away from $(u, t)$" – this is because the LCA between $(u, v)$ has to be higher than the LCA between $(u, t)$ for any $t \in \cup_{i=\ell^*}^{\infty} V_i^{\mathrm{small}}$. As such, define $Y_{u,v}$ as the number of total answers of "$v$ splits away from $(u, t)$" by $\mathcal{O}$, we again have

$$\mathbb{E}\left[Y_{u,v}\right] \geq \frac{9}{10} \cdot \frac{4n_H}{5} = \frac{18}{25} \cdot n_H.$$

As such, we again have

$$\Pr\left(Y_{u,v} \leq \frac{3n_H}{5}\right) = Pr\left(Y_{u,v} \leq \frac{15}{18} \cdot \mathbb{E}\left[X_{u,v}\right]\right)$$
$$\leq \frac{1}{10} \cdot \frac{1}{n^3}. \qquad (\frac{9}{50} \cdot n_H \geq 9000 \log n \text{ using the lower bound on the size of } V_H)$$

By a union bound over at most $n$ vertices, no vertex $v \in \cup_{i=1}^{\ell^*-1} V_i^{\mathrm{small}}$ or $v \in \cup_{i=\ell^*+1}^{\infty} V_i^{\mathrm{small}}$ can pass the test and be recorded as a 'counterpart'.

**Proof of Item b)..** We now show that vertices $v \in \cup_{i=1}^{\ell} V_i^{\mathrm{small}}$ cannot pass the "counterpart" test. For vertices in $v \in V_\ell^{\mathrm{small}}$, we claim that there are too many vertices $t$ that split away from $(u, v)$. To see this, let us define $A_{u,v}$ as the total number of answers of "$t$ splits away from $(u, v)$" answered by $\mathcal{O}$. Note that for any vertex $t \in \cup_{i=\ell^*+1}^{\infty} V_i^{\mathrm{small}}$, the answer is always "yes" since $\ell^* \geq \tilde{\ell} \geq \ell$. Therefore, we can lower bound the expectation of $A_{u,v}$ with $\mathbb{E}\left[A_{u,v}\right] \geq \frac{2}{5} n_H \cdot \frac{9}{10} \geq \frac{9}{25} \cdot n_H$. Therefore, we can apply Chernoff bound to get

$$\Pr\left(A_{u,v} \leq \frac{n_H}{6}\right) \leq \exp\left(-\frac{(1/2)^2}{3} \cdot \mathbb{E}\left[A_{u,v}\right]\right) \leq \frac{1}{10} \cdot \frac{1}{n^3}.$$

We now turn to the vertices that are in $\cup_{i=1}^{\ell-1} V_i^{\mathrm{small}}$. Note that by definition, there is $\left|\cup_{i=\ell}^{\infty} V_i^{\mathrm{small}}\right| \geq \left|\cup_{i=\ell^*}^{\infty} V_i^{\mathrm{small}}\right| \geq \frac{4}{5} \cdot n_H$. For any $t \in \cup_{i=\ell^*}^{\infty} V_i^{\mathrm{small}}$, $v$ splits away from $(u, t)$. As such, define $B_{u,v}$ as the number of answers by $\mathcal{O}$ with "$v$ splits away from $(u, t)$", there is $\mathbb{E}\left[B_{u,v}\right] \geq \frac{18}{25} \cdot n_H$. Once again, by applying Chernoff bound, there is

$$\Pr\left(B_{u,v} \leq \frac{3n_H}{5}\right) \leq \exp\left(-\frac{(1/5)^2}{3} \cdot \mathbb{E}\left[B_{u,v}\right]\right) \leq \frac{1}{10} \cdot \frac{1}{n^3}.$$

Applying a union bound over the cases and all $n_H \leq n$ vertices gives us the desired statements.

**Proof of Item c)..** We fix $\tilde{\ell}$ as the minimum level such that the union of $\cup_{i=1}^{\tilde{\ell}} V_i^{\text{small}}$ has at least $n_H/25$ vertices. Let $\ell \leq \tilde{\ell}$, and for any vertex $v$ in $V_k^{\text{small}}$ for $k > \ell$, define the following two random variables: $X_{u,v}$ as the number of answers that "$t$ splits away from $(u,v)$", and $Y_{u,v}$ as the number of answers that "$v$ splits away from $(u,t)$". We shall show that both terms are not large and $v$ can always pass the test. (Note that $X$ and $Y$ are overloaded and are *not* related to their meaning in the proof of Line a)..)

Note that by definition, we have $\left| \cup_{i=1}^{\tilde{\ell}-1} V_i^{\text{small}} \right| < \frac{n_H}{25}$. Since $\ell \leq \tilde{\ell}$, for any vertex $u \in \cup_{i=1}^{\ell} V_i^{\text{small}}$, only vertices $t \in \cup_{i=1}^{\ell-1} V_i^{\text{small}}$ are *actually* splitting away from $(u,v)$ for $v \in \cup_{i=\ell}^{\infty} V_i^{\text{small}}$. As such, we have $\mathbb{E}[X_{u,v}] \leq (\frac{1}{10} + \frac{1}{25}) \cdot n_H = \frac{7}{50} \cdot n_H$ by the correct probability of $\mathcal{O}$. For $X_{u,v}$, we also note that if the answer is at most $3000 \log n$, then the vertex trivially passes the test. Therefore, we assume $X_{u,v} \geq \mathbb{E}[X_{u,v}] \geq 3000 \log n$, and we again apply Chernoff for the tail bound:

$$\Pr\left( X_{u,v} \geq \frac{n_H}{6} \right) \leq \exp\left( -\frac{(4/21)^2}{3} \cdot \mathbb{E}[Y_{u,v}] \right)$$
$$\leq \frac{1}{10} \cdot \frac{1}{n^3}. \qquad \text{(using the condition } \mathbb{E}[Y_{u,v}] \geq 3000 \log n)$$

For $Y_{u,v}$, note that only $t \in V_{\tilde{\ell}}^{\text{small}}$ can report "$v$ splits away from $(u,t)$" since the LCA between $(u,t)$ is at least $\tilde{\ell}$ for any other $v$. Therefore, the number of signals we can possibly get is $n_H/2$ plus the noise induced by the oracle $\mathcal{O}$. As such, we again have $\mathbb{E}[Y_{u,v}] \leq 11n_H/20$. Also, note that if $Y_{u,v}$ is less than $30000 \log n$, the vertex $v$ would always pass the test, which allows us to assume w.log. that $\mathbb{E}[Y_{u,v}] \geq 30000 \log n$. Therefore, we can again apply Chernoff bound to show

$$\Pr\left( Y_{u,v} \geq \frac{3n_H}{5} \right) \leq \exp\left( -\frac{(1/11)^2}{3} \cdot \mathbb{E}[Y_{u,v}] \right)$$
$$\leq \frac{1}{10} \cdot \frac{1}{n^3}. \qquad \text{(using the condition } \mathbb{E}[Z_{u,v}] \geq 30000 \log n)$$

$$\text{Lemma H.3} \quad \square$$

A direct corollary of Lemma H.3 is that the set $T$ induced by any $u \in \cup_{i=1}^{\tilde{\ell}} V_i^{\text{small}}$ is larger than the $T$ induced by $u \in \cup_{i=\tilde{\ell}+1}^{\infty} V_i^{\text{small}}$, and the "higher level" vertices in $\cup_{i=1}^{\tilde{\ell}} V_i^{\text{small}}$ induces larger sets. More formally, we can summarize this observation as follows.

**Lemma H.4.** *Conditioning on the high-probability event of Lemma H.3, the following statements are true:*

- *Let $T_1$ be the vertex set induced by $u_1 \in \cup_{i=1}^{\tilde{\ell}} V_i^{\text{small}}$ from Line 12 in Algorithm 12, and let $T_2$ be the vertex set induced by $u_2 \in \cup_{i=\tilde{\ell}+1}^{\infty} V_i^{\text{small}}$ from Line 12 in Algorithm 12. We have $|T_1| \geq |T_2|$.*

- *Let $\ell_1 \leq \ell_2 \leq \tilde{\ell}$. Let $T_1$ be the vertex set induced by $u_1 \in V_{\ell_1}^{\text{small}}$ from Line 12 in Algorithm 12, and let $T_2$ be the vertex set induced by $u_2 \in V_{\ell_2}^{\text{small}}$ from Line 12 in Algorithm 12. We have $|T_1| \geq |T_2|$.*

*Proof.* We prove the second bullet first since the conclusion can be used to prove the first bullet. Note that conditioning on the high-probability event of Lemma H.3, if $\ell_1 \leq \ell_2$, we have $T_2 \subseteq T_1$ by Item c).. Therefore, we have $|T_1| \geq |T_2|$.

For the first bullet, note that conditioning on the high-probability event of Lemma H.3, the set $T_2$ can either be $V_{\tilde{\ell}}^{\text{small}}$ (by Item a).) or $\cup_{i=\tilde{\ell}+1}^{\infty} V_i^{\text{small}}$ (by Item b).). In either case, the size of such a set is at most $\cup_{i=\tilde{\ell}+1}^{\infty} V_i^{\text{small}}$, which is the set $T_1$ generated by $u \in V_{\tilde{\ell}}^{\text{small}}$. Furthermore, by the result in the second bullet, if $u \in V_i^{\text{small}}$ for some $i \leq \tilde{\ell}$, the induced set $T_1$ can only be larger. This proves the first bullet. $\square$

We now show that conditioning on Lemma H.3 (resp. Lemma H.4), if the size of the orphaned set is relatively small, we will not need the subroutine in Line 12, and all the guarantees in Lemma H.1 will be satisfied.

**Lemma H.5.** *Let $\widetilde{V}$ and $V_H$ be as prescribed by Lemma H.1, and suppose the size of the orphaned set is at most $\frac{99}{100} \cdot \tilde{n}$, i.e., $\left| V^{\text{orphan}} \right| \leq \frac{99}{100} \cdot \tilde{n}$. Then, with high probability, we have $T^* \cap \widetilde{V} \neq \emptyset$, and the resulting $T^*$ and $\widetilde{V} \setminus T^*$ satisfy the properties of Lemma H.1.*

*Proof.* We discuss the cases based on whether $V_{\mathrm{H}} = \widetilde{V}$ and whether there exists a $V_\ell^{\mathrm{small}}$ such that $\left|V_\ell^{\mathrm{small}}\right| \geq \frac{99}{100} \cdot \tilde{n}$.

1. **If $V_{\mathrm{H}} = \widetilde{V}$.** In this case, we show that with high probability, the guarantees in Item 1, Item 2, and Item 3 of Lemma H.1 always hold, and the Item 4b case of Lemma H.1 is going to happen. To see this, note that with high probability, we will sample a vertex that is among the first $\frac{\tilde{n}}{25}$ vertices in the small-tree split order: the probability for us to not sample a vertex from $V^{\mathrm{small}}(\leq \frac{\tilde{n}}{25})$ is at most

$$(\frac{24}{25})^{500 \log n} \leq \frac{1}{10} \cdot \frac{1}{n^2}.$$

As such, we can condition on a vertex $v'$ among $V^{\mathrm{small}}(\leq \frac{\tilde{n}}{25})$ is sampled. By Lemma H.3 and Lemma H.4, the counterpart set induced by $v' \in V^{\mathrm{small}}(\leq \frac{\tilde{n}}{25})$ is among $\cup_{i=1}^{\tilde{\ell}} V_i^{\mathrm{small}}$, which necessarily induces a larger set than any other $u \in \cup_{i=\tilde{\ell}+1}^{\infty} V_i^{\mathrm{small}}$. Therefore, the induced set $T^*$ must be from the vertex $v'$.

We now use this to verify the desired properties. The proofs of Item 1, Item 2, and Item 3 are straightforward as follows.

- For Item 1, note that by Lemma H.3, the induced set $T^*$ is always $\cup_{i=\ell}^{\infty} V_i^{\mathrm{small}}$, which forms a maximal subtree in $\mathcal{T}^*(V_{\mathrm{H}})$. Similarly, $T \cap \widetilde{V}$ induces a maximal subtree in $\mathcal{T}^*(\widetilde{V})$.
- For Item 2, note that as long as $T^*$ includes the orphaned set $V^{\mathrm{orphan}}$, there will be only one edge connecting to the node induces $T^*$ in $\mathcal{T}^*$. In this case, there is no orphaned set, and Item 2 holds trivially.
- Item 3 directly follows from Lemma H.3 since $T^*$ is induced by $u^*$.

For the size upper and lower bounds (Item 4), we verify that the guarantees for case 4b always holds. Note that conditioning on the high-probability event of Lemma H.3, the size is at least $\frac{2n_H}{5} = \frac{2\tilde{n}}{5}$ (Observation 3). Therefore, the set $T^* \cap \widetilde{V} = T^*$ has size at least $\frac{2\tilde{n}}{5} \geq \frac{1}{200} \cdot \tilde{n}$, which proves the lower bound (Item 4(b)i). For the upper bound (Item 4(b)ii), we note that for the size of $T^* \cap \widetilde{V}$ to be more than $(1 - \frac{1}{10000 \log^2 \tilde{n}}) \cdot \tilde{n}$, a *necessary* condition is to sample a vertex $u \in V^{\mathrm{small}}(\leq \tilde{n}/10000 \log^2 \tilde{n})$. Since we sample $500 \log n$ vertices, define $X$ as the random variable for the number of vertices sampled from $V^{\mathrm{small}}(\leq \tilde{n}/10000 \log^2 \tilde{n})$, we have

$$\mathbb{E}[X] \leq \frac{1}{20 \log n}.$$

Since $X$ is a summation of independent random variables supported on $[0, 1]$, we can apply Chernoff bound to show that

$$\Pr(X \geq 1) = \Pr(X \geq 50 \log n \cdot \mathbb{E}[X])$$
$$\leq \exp\left(-\frac{2500 \log^2 n \cdot \frac{1}{20 \log n}}{2 + 20 \log n}\right)$$
$$\leq \frac{1}{10} \cdot \frac{1}{n^2}.$$

Therefore, we can apply a union bound to show that with high probability, the size of $T^* \cap \widetilde{V}$ will *not* be larger than $(1 - \frac{1}{10000 \log^2 \tilde{n}}) \cdot \tilde{n}$, as desired.

2. **If $V_{\mathrm{H}} \neq \widetilde{V}$.** We need to handle this case with more care. We first show that at least one vertex that is in $\widetilde{V} \setminus V^{\mathrm{orphan}}$ can be sampled with high probability. To see this, note that by the size bound on $\left|V^{\mathrm{orphan}}\right|$, the probability for a vertex in $\widetilde{V} \setminus V^{\mathrm{orphan}}$ to *not* be sampled is at most $99/100$. Therefore, the probability for no vertices in $\widetilde{V} \setminus V^{\mathrm{orphan}}$ to be sampled is at most

$$(99/100)^{500 \log n} \leq \frac{1}{10} \cdot \frac{1}{n^2}.$$

We condition on the high-probability event that at least one vertex from $\widetilde{V} \setminus V^{\mathrm{orphan}}$ is sampled for the rest of the proof. We now discuss two sub-cases.

a). **If there exists a $V_\ell^{\text{small}}$ such that $\left|V_\ell^{\text{small}}\right| \geq \frac{99}{100} \cdot \tilde{n}$ and no vertex from $\cup_{i=1}^{\ell-1} V_i^{\text{small}}$ is sampled.** In this case, we show that with high probability, Item 1, Item 2, and Item 3 of Lemma H.1 always hold, and Item 4a in Lemma H.1 is going to happen. Note that in this case, there exist vertices of $V_\ell^{\text{small}}$ that are among $V^{\text{small}}(\leq \tilde{n}/100)$, and it is of the size at least $\tilde{n}/100$. As such, the probability for us to sample at least one vertex from $V_\ell^{\text{small}}$ is at least

$$1 - (\frac{1}{100})^{500 \log n} \geq 1 - \frac{1}{10} \cdot \frac{1}{n^5}.$$

Let $v \in V_\ell^{\text{small}}$ be the sampled vertex. Note that since $V_{\text{H}} \neq \widetilde{V}$, the vertex we sample from $V_\ell^{\text{small}}$ is among the vertices of $\cup_{i=1}^{\tilde{\ell}} V_i^{\text{small}}$ in Lemma H.3. Therefore, by the same argument of the $V_{\text{H}} = \widetilde{V}$ case, if we pick $T^*$ with the largest size, the entire set of $\cup_{i=\ell+1}^{\infty} V_i^{\text{small}}$ is going to be included in $T^*$. Therefore, the properties of Item 1, Item 2, and Item 3 follow from the same argument of the $V_{\text{H}} = \widetilde{V}$ case.

Furthermore, by the condition that no vertex from $\cup_{i=1}^{\ell-1} V_i^{\text{small}}$ is sampled, we cannot have larger such $T^*$ sets (see Lemma H.4). As such, the set we will pick is necessarily the set $T$ corresponds to $v \in V_\ell^{\text{small}}$, and $V_\ell^{\text{small}}$ becomes the new $V^{\text{orphan}}$ set of the next iteration.

Finally, note that we have $\left|V_\ell^{\text{small}}\right| \geq \frac{99}{100} \cdot \tilde{n}$. And after we remove $T^*$ from $\widetilde{V}$, we have $(\widetilde{V} \leftarrow \widetilde{V} \setminus T^*)$, which means $(\tilde{n} \leftarrow \tilde{n} - C)$ for some $C > 0$. As such, for the next iteration, we must have $\left|V^{\text{orphan}}\right| \geq \frac{99}{100} \cdot \tilde{n}$.

b). **If there exists a $V_\ell^{\text{small}}$ such that $\left|V_\ell^{\text{small}}\right| \geq \frac{99}{100} \cdot \tilde{n}$ and a vertex from $\cup_{i=1}^{\ell-1} V_i^{\text{small}}$ is sampled.** In this case, we show that Item 1, Item 2, and Item 3 of Lemma H.1 always hold, and Item 4b of Lemma H.1 is going to happen with high probability. Note that since a vertex $v' \in \cup_{i=1}^{\ell-1} V_i^{\text{small}}$ is sampled, and since $v'$ is among $\cup_{i=1}^{\tilde{\ell}} V_i^{\text{small}}$ in Lemma H.3, the *entire set* of $V_\ell^{\text{small}}$ is going to be included in $T^*$. The properties as prescribed by Item 1, Item 2, and Item 3 follow from the argument in the $\widetilde{V} = V_{\text{H}}$ case, and the size lower bound becomes

$$\left|T^* \cap \widetilde{V}\right| \geq \frac{99}{100} \cdot \tilde{n} \geq \frac{1}{200} \cdot \tilde{n},$$

as desired. Finally, note that we do not need to guarantee the size upper bound (Item 4(b)ii) since we will not be able to meet the $V_{\text{H}} = \widetilde{V}$ condition.

c). **If $\left|V_\ell^{\text{small}}\right| < \frac{99}{100} \cdot \tilde{n}$ for all $\ell$ among $\widetilde{V}$.** In this case, we show that Item 1, Item 2, and Item 3 of Lemma H.1 always hold, and Item 4b case of Lemma H.1 will happen. We first show the size lower bound of Item 4b: note that with high probability, we can sample one vertex that is among the first $1/200$ vertices to be split in $\widetilde{V}$ in the small-tree split order: the probability for us to *not* sample any vertex among the first $n/200$ vertices in the small-tree split order is at most

$$\left(\frac{199}{200}\right)^{500 \log n} \leq \frac{1}{3} \cdot \frac{1}{n^2}.$$

Therefore, we condition on the event that a vertex $v'$ in $V^{\text{small}}(\leq \tilde{n}/200)$ is sampled. Since we have the condition that $\left|V_\ell^{\text{small}}\right| < \frac{99}{100} \cdot \tilde{n}$ for all $\ell$ among $\widetilde{V}$, a vertex among $V^{\text{small}}(\leq \tilde{n}/200)$ can induce at most $\frac{99\tilde{n}}{100} + \frac{\tilde{n}}{200} = \frac{199}{200} \cdot \tilde{n}$ vertices. As such, let $\ell'$ be the level in the small-tree split order of $v'$, we have

$$\left|\cup_{i=\ell'+1}^{\infty} V_i^{\text{small}}\right| \geq \frac{1}{200} \cdot \tilde{n}.$$

Furthermore, as in the case analysis of $V_{\text{H}} \neq \widetilde{V}$, which means $v'$ is among the vertices of $\cup_{i=1}^{\tilde{\ell}} V_i^{\text{small}}$ in Lemma H.3. Therefore, the properties of Item 1, Item 2, and Item 3 follow from the same argument as in the $V_{\text{H}} = \widetilde{V}$ case.

For the size bounds of $T^* \cap \widetilde{V}$, by Lemma H.4, if we pick the largest $T^*$ by the subroutine of line Line 12, at least the entire set of $\cup_{i=\ell'+1}^{\infty} V_i^{\text{small}}$ is going to be included, which means the size of $T^* \cap \widetilde{V}$ is of size at least $\frac{1}{200} \cdot \tilde{n}$. This gives us the size lower bound.

Finally, we again note that we do *not* need to guarantee the size upper bound (Item 4(b)ii) since we will not be able to meet the $V_{\text{H}} = \widetilde{V}$ condition.

$\square$

We now handle the case when $V^{\text{orphan}}$ becomes large. We first note that if we happen to sample a vertex $u \in \widetilde{V} \setminus V^{\text{orphan}}$, we can still guarantee $T^* \cap \widetilde{V} \neq \emptyset$ and obtain the properties as prescribed by Lemma H.1.

**Lemma H.6.** *Let $\widetilde{V}$ and $V_H$ be as prescribed by Lemma H.1, and suppose the size of the orphaned set is more than $\frac{99}{100} \cdot \tilde{n}$, i.e., $\left|V^{\mathrm{orphan}}\right| > \frac{99}{100} \cdot \tilde{n}$. Furthermore, suppose $U \cap (\widetilde{V} \setminus V^{\mathrm{orphan}}) \neq \emptyset$. Then, with high probability, we have $T^* \cap \widetilde{V} \neq \emptyset$, and the resulting $T^*$ and $\widetilde{V} \setminus T^*$ satisfy the properties of Lemma H.1.*

*Proof.* In the lemma statement, we have already conditioned on a vertex sampled from $\widetilde{V} \setminus V^{\mathrm{orphan}}$. Furthermore, we can again show that the probability for us to sample a vertex among the first $n_H/25$ in the small-tree split order is at least

$$1 - \left(\frac{24}{25}\right)^{500 \log n} \geq 1 - \frac{1}{10} \cdot \frac{1}{n^2}.$$

The events of $U \cap (\widetilde{V} \setminus V^{\mathrm{orphan}}) \neq \emptyset$ and $U \cap V^{\mathrm{small}}(\leq n_H/25) \neq \emptyset$ are *not* independent. Nevertheless, we can still apply a *union bound* and argue that both events happen with high probability.

Conditioning on the high-probability events as above, we can argue by Lemma H.3 and Lemma H.4 that the entire set of $V^{\mathrm{orphan}}$ is going to be included by the subroutine as defined in line Line 12 of Algorithm 12, which gives us the size lower bound. We do *not* need to guarantee the size upper bound since we cannot meet the condition of $V_H = \widetilde{V}$.

Finally, for properties of Item 1, Item 2, and Item 3, we can repeat our proofs in Lemma H.5. We provide the analysis again for the purpose of self-contained proof.

- For Item 1, note that by Lemma H.3, the induced set $T^*$ is always $\cup_{i=\ell}^{\infty} V_i^{\mathrm{small}}$. Therefore, $T \cap \widetilde{V}$ induces a maximal subtree in $\mathcal{T}^*(\widetilde{V})$.

- For Item 2, note that as long as $T^*$ includes the orphaned set $V^{\mathrm{orphan}}$, there will be only one edge connecting to the node induces $T^*$ in $\mathcal{T}^*$. This is exactly what we proved in the lemma.

- Item 3 directly follows from Lemma H.3 since $T^*$ is induced by $u^*$.

This concludes the proof of Lemma H.6. $\qquad\square$

By Lemma H.6, the only case of concern now is when $T^* \cap \widetilde{V} = \emptyset$, i.e., $\left|V^{\mathrm{orphan}}\right| > \frac{99}{100} \cdot \tilde{n}$ and $U$ does *not* contain samples from $\widetilde{V} \setminus V^{\mathrm{orphan}}$. We now show that our procedure in Line 12 can effectively distinguish between the cases of $\widetilde{V} \setminus V^{\mathrm{orphan}} = \emptyset$ (root cut) and $\widetilde{V} \setminus V^{\mathrm{orphan}}$ being small.

**Lemma H.7.** *Let $\widetilde{V}$ and $V_H$ be as prescribed by Lemma H.1, and suppose the size of the orphaned set is more than $\frac{99}{100} \cdot \tilde{n}$, i.e., $\left|V^{\mathrm{orphan}}\right| > \frac{99}{100} \cdot \tilde{n}$. Furthermore, suppose $T^* \cap \widetilde{V} = \emptyset$ in Line 12 of Algorithm 12. Then, the following statements are true.*

*(i).* *If $\widetilde{V} \setminus V^{\mathrm{orphan}} \neq \emptyset$, with high probability, we have $X^* \neq \emptyset$ and $X^* \subseteq (\widetilde{V} \setminus V^{\mathrm{orphan}})$, i.e., $X^*$ only contains vertices in $\widetilde{V}$ but not in $V^{\mathrm{orphan}}$.*

*(ii).* *If $\widetilde{V} \setminus V^{\mathrm{orphan}} = \emptyset$, with high probability, we have $X^* = \emptyset$.*

*Proof.* Consider the small-tree splitting order of $V_H$, and let $u \in V_\ell^{\mathrm{small}}$ for some $\ell \leq \tilde{\ell}$, where $V_{\tilde{\ell}}^{\mathrm{small}} = V^{\mathrm{orphan}}$. We prove that with high probability, $i$). no vertices $v \in \widetilde{V} \cap (\cup_{i=\ell}^{\infty} V_\ell^{\mathrm{small}})$ can be added to $X$ by Line 11; and $ii$). all vertices in $v \in \widetilde{V} \cap (\cup_{i=1}^{\ell-1} V_\ell^{\mathrm{small}})$ are added to $X$ by Line 11. (Note that this is why we name the subroutine as a "predecessor" test.)

We first observe that by our definition, there is $V^{\mathrm{orphan}} \subseteq \widetilde{V} \cap (\cup_{i=\ell}^{\infty} V_\ell^{\mathrm{small}})$. Too see $i$), note that any $v \in \widetilde{V} \cap (\cup_{i=\ell}^{\infty} V_\ell^{\mathrm{small}})$ can only split away from $(u, t)$ for $t \in V_\ell^{\mathrm{small}}$: this is true since for every $t \notin V_\ell^{\mathrm{small}}$, the lowest common ancestor between $(u, t)$ induces a subtree in $V_H$ that contains $v$. Moreover, since $V_\ell^{\mathrm{small}} \subseteq \widetilde{V}$, we have $\left|V_\ell^{\mathrm{small}}\right| \leq \tilde{n}$. Define $X_v$ as the number of answers "$v$ splits away from $(u, t)$" for $t \in V_H$ from $\mathcal{O}$. By our choice of the parameter *threshold-pred*, we assume w.log. $X_v \geq \tilde{n}$ since otherwise $v$ will not join $X$ anyway. If $n_H \leq 3\tilde{n}$, we have $\mathbb{E}[X_v] \leq \tilde{n} + \frac{n_H}{10} = \frac{13}{10} \cdot \tilde{n}$ in expectation. Since

$X_v$ is a summation of independent random variables supported on $\{0, 1\}$, we can apply Chernoff bound to obtain that

$$
\begin{aligned}
\Pr\left(X_v \geq \frac{3}{2} \cdot \tilde{n}\right) &= \Pr\left(X_v \geq \frac{15}{13} \cdot \mathbb{E}\left[X_v\right]\right) \\
&\leq \exp\left(-\frac{(2/13)^2}{3} \cdot \mathbb{E}\left[X_v\right]\right) \\
&\leq \frac{1}{10} \cdot \frac{1}{n^3}. \qquad \text{(using the condition on the lower bound of } X_v)
\end{aligned}
$$

On the other hand, if $n_H > 3\tilde{n}$, we have $\mathbb{E}\left[X_v\right] \leq \frac{n_H}{3} + \frac{n_H}{10} = \frac{13}{30} \cdot n_H$. We again assume w.log. that $X_v \geq \tilde{n}$, and we can apply Chernoff bound to obtain that

$$
\begin{aligned}
\Pr\left(X_v \geq \frac{2}{3} \cdot n_H\right) &= \Pr\left(X_v \geq \frac{20}{13} \cdot \mathbb{E}\left[X_v\right]\right) \\
&\leq \exp\left(-\frac{(8/13)^2}{3} \cdot \mathbb{E}\left[X_v\right]\right) \\
&\leq \frac{1}{10} \cdot \frac{1}{n^3}. \qquad \text{(using the condition on the lower bound of } X_v)
\end{aligned}
$$

Therefore, we can apply a union bound and argue that with high probability, no vertices $v \in \widetilde{V} \cap (\cup_{i=\ell}^{\infty} V_\ell^{\text{small}})$ can be added to $X$ by Line 11, as desired by $i)$.

We now proceed to show $ii)$. all vertices in $v \in \widetilde{V} \cap (\cup_{i=1}^{\ell-1} V_\ell^{\text{small}})$ are added to $X$ by Line 11. Note that $v \in \widetilde{V} \cap (\cup_{i=1}^{\ell-1} V_\ell^{\text{small}})$ implies $v \in \widetilde{V} \setminus V^{\text{orphan}}$. Therefore, $v$ splits from $(u, t)$ for every $t$ in the orphaned set *and* for every $t$ as the sibling of $V^{\text{orphan}}$, i.e., $t \in \cup_{i=\ell+1}^{\infty} V_i^{\text{small}}$. Define $Y_v$ as the number of answers "$v$ splits away from $(u, t)$" for $t \in V_H$ from $\mathcal{O}$. By our choice of the parameter *threshold-pred*, if $n_H \leq 3\tilde{n}$, since we have $\left|V^{\text{orphan}}\right| \geq \frac{99}{100}\tilde{n}$ and $\left|\cup_{i=\ell+1}^{\infty} V_i^{\text{small}}\right| \geq \left|V^{\text{orphan}}\right|$, we have $\mathbb{E}\left[Y_v\right] \geq \frac{9}{10} \cdot \frac{199}{100} \cdot \tilde{n}$ in expectation. Since $Y_v$ is a summation of independent random variables supported on $\{0, 1\}$, we can apply Chernoff bound to obtain that

$$
\begin{aligned}
\Pr\left(Y_v \leq \frac{3}{2} \cdot \tilde{n}\right) &= \Pr\left(Y_v \leq \frac{17}{15} \cdot \mathbb{E}\left[Y_v\right]\right) \\
&\leq \exp\left(-\frac{(2/15)^2}{3} \cdot \mathbb{E}\left[Y_v\right]\right) \\
&\leq \frac{1}{10} \cdot \frac{1}{n^3}. \qquad \text{(using } \mathbb{E}\left[Y_v\right] \geq \frac{17}{10} \cdot \tilde{n})
\end{aligned}
$$

On the other hand, if $n_H > 3\tilde{n}$, we have at least $n_H - \frac{1}{100}\tilde{n} \geq \frac{299}{300} n_H$ vertices $t$ such that $v$ splits away from $(u, t)$. Therefore, we have $\mathbb{E}\left[Y_v\right] \geq \frac{9}{10} \cdot \frac{299}{300} \cdot n_H \geq \frac{4}{5} \cdot n_H$ in expectation. Therefore, we can again apply Chernoff bound to obtain that

$$
\begin{aligned}
\Pr\left(Y_v \leq \frac{2}{3} \cdot n_H\right) &= \Pr\left(Y_v \geq \frac{5}{6} \cdot \mathbb{E}\left[Y_v\right]\right) \\
&\leq \exp\left(-\frac{(1/6)^2}{3} \cdot \mathbb{E}\left[Y_v\right]\right) \\
&\leq \frac{1}{10} \cdot \frac{1}{n^3}. \qquad \text{(using } \mathbb{E}\left[Y_v\right] \geq \frac{4}{5} \cdot n_H \geq \frac{4}{5} \cdot \tilde{n})
\end{aligned}
$$

Therefore, we can apply a union bound to obtain the desired statement on $ii)$.

By our statements in $i)$ and $ii)$ as above, we can already conclude that $X^* \subseteq (\widetilde{V} \setminus V^{\text{orphan}})$. Therefore, Item (ii). of Lemma H.7 follows straightforwardly since if $\widetilde{V} \setminus V^{\text{orphan}} = \emptyset$, any of its subset can only be empty as well. For Item (i)., what remains to show is that with high probability, there is $X^* \neq \emptyset$. Note that if $u \in V^{\text{orphan}}$ and $\widetilde{V} \setminus V^{\text{orphan}} \neq \emptyset$, then by our statements in $i)$ and $ii)$ above, the set $X^*$ will *not* be empty. Since we assume $\left|V^{\text{orphan}}\right| \geq \frac{99}{100} \cdot \left|\widetilde{V}\right|$, the probability for

$U'$ to *not* have any vertex $u \in V^{\text{orphan}}$ is at most

$$(\frac{1}{100})^{100 \log n} \leq \frac{1}{10} \cdot \frac{1}{n^3},$$

as desired. Thus, with high probability, $X^*$ is not empty, which proves Item (i). and concludes the proof of Lemma H.7. □

*Finalizing the proof of Lemma H.1.* By Lemma H.2, the algorithm runs in $n_H^2 \log n$ time. For the set of vertices $\widetilde{V}$, we either have $|V^{\text{orphan}}| < \frac{99}{100} \cdot \tilde{n}$ or $|V^{\text{orphan}}| \geq \frac{99}{100} \cdot \tilde{n}$. In the former case, we apply Lemma H.5, and all guarantees in Lemma H.1 are satisfied. Otherwise, if $|V^{\text{orphan}}| \geq \frac{99}{100} \cdot \tilde{n}$ *and* $U \cap (\widetilde{V} \setminus V^{\text{orphan}}) \neq \emptyset$, by Lemma H.6, we can still satisfy the properties as prescribed by Lemma H.1.

The only remaining case is if $|V^{\text{orphan}}| \geq \frac{99}{100} \cdot \tilde{n}$ *and* $U \cap (\widetilde{V} \setminus V^{\text{orphan}}) = \emptyset$. In such a case, the algorithm will enter Line 12. If $\widetilde{V} \setminus V^{\text{orphan}} = \emptyset$, then by Item (ii). of Lemma H.7, the algorithm uses $V_H = \widetilde{V}$ for a single level of recursion call, and the properties of Lemma H.1 are satisfied by the guarantees of Lemma H.5 (since now $V^{\text{orphan}} = \emptyset$). Otherwise, if $\widetilde{V} \setminus V^{\text{orphan}} \neq \emptyset$, note that by Item (i). of Lemma H.7, any arbitrary vertex $u \in U^*$ belongs to $V_\ell^{\text{small}}$ for some $\ell < \tilde{\ell}$, such that $V_{\tilde{\ell}}^{\text{small}} = V^{\text{orphan}}$. Since $|V^{\text{orphan}}| \geq \frac{99}{100} \tilde{n}$ and $n_H \geq \tilde{n}$, the vertex $u$ is among $V^{\text{small}}(\leq n_H/100)$. As such, by Lemma H.3, with high probability, the vertex set $T^*$ contains all vertices in $V^{\text{orphan}}$, and the size is sufficiently large to guarantee Item 4(b)i of Lemma H.1. We do not need to guarantee Item 4(b)ii since $V^{\text{orphan}} \neq \emptyset$, and we will not meet the $\widetilde{V} = V_H$ condition. The guarantees of Item 1 and Item 2 are similarly satisfied since we remove a set $\cup_i V_i^{\text{small}}$ that contains $V^{\text{orphan}}$. Finally, by a similar argument as we used in Lemma H.5 and Lemma H.6, Item 3 is satisfied as desired. □

## H.2. An algorithm to merge two subtrees

We now move to the algorithm that merges two partial trees. Note that this task is *not* trivial: we use the thought process of small-tree split order in the proof of Lemma H.1, but the *algorithm* `vertex-split`$^{\mathcal{O}}(\widetilde{V})$ does *not* immediately tell us which node did we "extract" the set $T$. As such, it still takes considerable work to merge the two trees on the "right" internal node.

Exactly here is why we need the split algorithm `vertex-split`$^{\mathcal{O}}(\widetilde{V})$ to return the vertex $u^*$. Note that our goal is essentially to find the lowest common ancestor $x$ between $u^*$ and $T$, and "stitch" the tree $\mathcal{T}_T$ to the node. As such, a natural strategy is to ask whether in $\mathcal{T}^*$ (resp. $\mathcal{T}^*(\widetilde{V})$), whether a vertex $v \in \widetilde{V}$ *splits away* from $(u^*, x)$ for $x \in T$. If $\mathcal{O}$ is to answer the queries correctly, all vertices that are "outside" $\text{LCA}_{\mathcal{T}^*(\widetilde{V})}(T \cup \{u^*\})$ would answer *yes*, and all vertices that are among the leaves of $\text{LCA}_{\mathcal{T}^*(\widetilde{V})}(T \cup \{u^*\})$ would answer *no*. We then use the fact that $T$ is always large enough to beat the noise from $\mathcal{O}$.

The formal description of the algorithm is as Algorithm 13.

---

**Algorithm 13** `tree-merge`$^{\mathcal{O}}(\mathcal{T}_T, \mathcal{T}_{\widetilde{V} \setminus T}, u^*)$: an algorithm to merge partial trees $\mathcal{T}_T$ and $\mathcal{T}_{\widetilde{V} \setminus T}$.

---

**Input:** Vertex set $\widetilde{V}$ of size $\tilde{n}$; a splitting oracle $\mathcal{O}$; Partial trees $\mathcal{T}_T$ and $\mathcal{T}_{\widetilde{V} \setminus T}$ constructed by `vertex-split`$^{\mathcal{O}}(\widetilde{V}, V_H)$;
   vertex $u^*$ by `vertex-split`$^{\mathcal{O}}(\widetilde{V}, V_H)$

**Output:** A partial tree on $\mathcal{T}_{\widetilde{V}}$

Initialize $S' \leftarrow \emptyset$ **for** $s \in \widetilde{V} \setminus T$ **do**

>  Initialize a counter $c_s \leftarrow 0$ **for** *Every vertex* $t \in T$ **do**
>  >  If $s$ splits away from $(u^*, t)$, increase $c_s$ by 1
>
>  **end**
>  **if** $c_s \leq \frac{1}{2} \cdot |T|$ **then**
>  >  Add $s$ to $S'$
>
>  **end**

**end**

Take the lowest common ancestor $x = \text{LCA}_{\mathcal{T}_{\widetilde{V} \setminus T}}(S')$ If $S'$ does *not* induces a *maximal tree* in $\mathcal{T}_{\widetilde{V} \setminus T}$, i.e., $\text{leaves}_{\mathcal{T}_{\widetilde{V} \setminus T}}[S'] \neq S'$, abort the algorithm and report "fail" If the algorithm does not fail, split node $x$ into two nodes: the left node induces the subtree of $x$, and the right node induces the subtree $\mathcal{T}_T$.

---

We now present the guarantees of the tree-merging algorithm.

**Lemma H.8.** *Given any composable vertex set $\widetilde{V} \subseteq V$ such that $\left|\widetilde{V}\right| = \tilde{n} \geq 50000 \log n$, a splitting oracle $\mathcal{O}$ with correct probability $9/10$, and suppose $\mathcal{T}_{\widetilde{V} \setminus T}$, $\mathcal{T}_T$, and $u^*$ are obtained by Algorithm 12. Furthermore, assume that*

*i). The high probability events of Lemma H.1 happens;*

*ii). The partial tree $\mathcal{T}_{\widetilde{V} \setminus T}$ is (weakly) consistent with $\mathcal{T}^*(\widetilde{V} \setminus T)$, $\mathcal{T}_T$ is (weakly) consistent with $\mathcal{T}^*(T)$.*

*Then, with high probability, Algorithm 13 runs in time $O(\tilde{n}^2)$, and outputs a partial tree $\mathcal{T}_{\widetilde{V}}$ that is weakly consistent with $\mathcal{T}^*(\widetilde{V})$.*

*Proof.* We first remind the readers of the definition of a partial tree $\mathcal{I}$ *weakly consistent* with another tree $\mathcal{T}$. For $\mathcal{I}$ to be consistent with $\mathcal{T}$, there should be

a). each super-vertex of $\mathcal{I}$ corresponding to a connected subtree in $\mathcal{T}$ with out-degree at most 2, and each of the edge connects to either a parent or a sibling node; and

b). for leaves $(x, y)$ in $\mathcal{I}$, let $X$ and $Y$ be the corresponding leaves in $\mathcal{T}$, the subtrees induced by $\text{LCA}_{\mathcal{T}}(x, y)$ and $\text{LCA}_{\mathcal{T}}(X \cup Y)$ contain exactly the same set of leaves.

We now show that with high probability, the algorithm $\texttt{tree-merge}^{\mathcal{O}}(\mathcal{T}_T, \mathcal{T}_{\widetilde{V} \setminus T}, u^*)$ outputs a partial tree that satisfies $a)$ and $b)$ w.r.t. $\mathcal{T}^*(\widetilde{V})$. For $a)$, we note that the super-vertices in $\mathcal{T}^*(\widetilde{V})$ and $(\mathcal{T}^*(\widetilde{V} \setminus T), \mathcal{T}^*(T))$ are exactly the same. Furthermore, by the high probability event of Lemma H.1, both $\mathcal{T}^*(\widetilde{V})$ and $(\mathcal{T}^*(\widetilde{V} \setminus T), \mathcal{T}^*(T))$ satisfy the *weakly consistent* property. As such, the guarantee of $a)$ follows.

The main work here is to prove the guarantee prescribed by $b)$. To this end, we first observe that for any set of vertices $X$ and $Y$, if $\text{LCA}_{\mathcal{T}^*(\widetilde{V})}(X \cup Y)$ only contain vertices in $T$ (resp. $\widetilde{V} \setminus T$), then the assumptions of $\mathcal{T}_{\widetilde{V} \setminus T}$ being consistent with $\mathcal{T}^*(\widetilde{V} \setminus T)$ and $\mathcal{T}_T$ being consistent with $\mathcal{T}^*(T)$ is sufficient to prove the leaves induced by $\text{LCA}_{\mathcal{T}_{\widetilde{V}}}(X \cup Y)$ is the same as the leaves of $\text{LCA}_{\mathcal{T}^*(\widetilde{V})}(X \cup Y)$. This is evident by using the procedure that constructs $\mathcal{T}^*(\widetilde{V})$ as in Definition 25.

The final missing piece is the vertex sets $X$ and $Y$ that induce vertices in both $T$ and $\widetilde{V} \setminus T$, which is the place where we need to show that the merging algorithm finds the "correct" node to merge. Let $S = V^{\text{orphan}}$ be the orphaned vertices by removing $T$ from $\widetilde{V}$. Since we condition on the high probability event of Lemma H.1, there must be an internal vertex $z$, such that $z = \text{LCA}_{\mathcal{T}^*(\widetilde{V})}(T \cup \{u^*\})$, and nodes $r_T$ and $r_S$ such that $i).\ z = \text{pa}(r_T)$ and $z = \text{pa}(r_S)$ in $\mathcal{T}^*(\widetilde{V})$ and $ii).$ the induced leaves of $r_T$ is $T$ and the induced leaves of $r_S$ is $S$ such that $u^* \in S$. Furthermore, we have $|T| \geq 50000 \log n \cdot \frac{1}{200} \geq 200 \cdot \log n$ by Lemma H.1. We now claim that by running $\texttt{tree-merge}^{\mathcal{O}}(\mathcal{T}_T, \mathcal{T}_{\widetilde{V} \setminus T}, u^*)$, the set $S'$ we recover is exactly the leaves of $S$ (the $V^{\text{orphan}}$ set of vertices). The detailed analysis is as follows.

- For each $s \in S$, observe that in $\mathcal{T}^*$ (and $\mathcal{T}^*(\widetilde{V})$), $s$ does *not* split away from $(u^*, t)$ for $t \in T$. Therefore, define $C_s$ as the random variable that records "$s$ split away from $(u^*, t)$ for $t \in T$" from $\mathcal{O}$, we have in expectation $\mathbb{E}[C_s] \leq \frac{1}{10} \cdot |T|$. If $C_s \leq 20 \cdot \log n$, it trivially fails the test. Otherwise, if $C_s \geq 20 \cdot \log n$, we can apply Chernoff bound to get that

$$
\begin{aligned}
\Pr\left(C_s \geq \frac{1}{2} \cdot |T|\right) &= \Pr(C_s \geq 5 \cdot \mathbb{E}[C_s]) \\
&\leq \exp\left(-\frac{5^2}{3} \cdot \mathbb{E}[C_s]\right) \\
&\leq \frac{1}{5} \cdot \frac{1}{n^3}. \qquad\qquad (\text{using } C_s \geq 20 \cdot \log n)
\end{aligned}
$$

- For each $d \notin S$, observe that in $\mathcal{T}^*$ (and $\mathcal{T}(\widetilde{V})$), $d$ *does* split away from $(u^*, t)$ for $t \in T$. Therefore, define $C_d$ as the random variable that records "$d$ split away from $(u^*, t)$ for $t \in T$" from $\mathcal{O}$, we have in expectation $\mathbb{E}[C_d] \geq \frac{9}{10} \cdot |T|$.

Therefore, we can apply Chernoff bound to get that

$$\Pr\left(C_d \le \frac{1}{2} \cdot |T|\right) = \Pr\left(C_d \le \frac{5}{9} \cdot \mathbb{E}\left[C_d\right]\right)$$

$$\le \exp\left(-\frac{(4/9)^2}{3} \cdot \mathbb{E}\left[C_d\right]\right)$$

$$\le \frac{1}{5} \cdot \frac{1}{n^3}. \qquad\qquad \text{(using } C_d \ge 200 \cdot \log n\text{)}$$

We can then apply a union bound over at most $\tilde{n} \le n$ vertices in $\widetilde{V} \setminus T$ to get the desired statement.

Observe that any internal node that induces leaves in both $T$ and $\widetilde{V} \setminus T$ has to at least include the whole set of $S$ and $T$. For any leaves $x$ and $y$ in $T_{\widetilde{V}}$, let the induced set of vertices in the leaves of $\mathcal{T}^*(\widetilde{V})$ be $Z = S \cup T \cup P$. By the above argument, $S \cup T$ should be returned in the set of vertices. Furthermore, by the consistency between $\mathcal{T}_{\widetilde{V}\setminus T}$ and $\mathcal{T}^*(\widetilde{V} \setminus T)$, the set $P$ should also be returned. This concludes the proof. $\qquad\square$

The merging algorithm in Lemma H.8 could be visualized as in Figure 10.

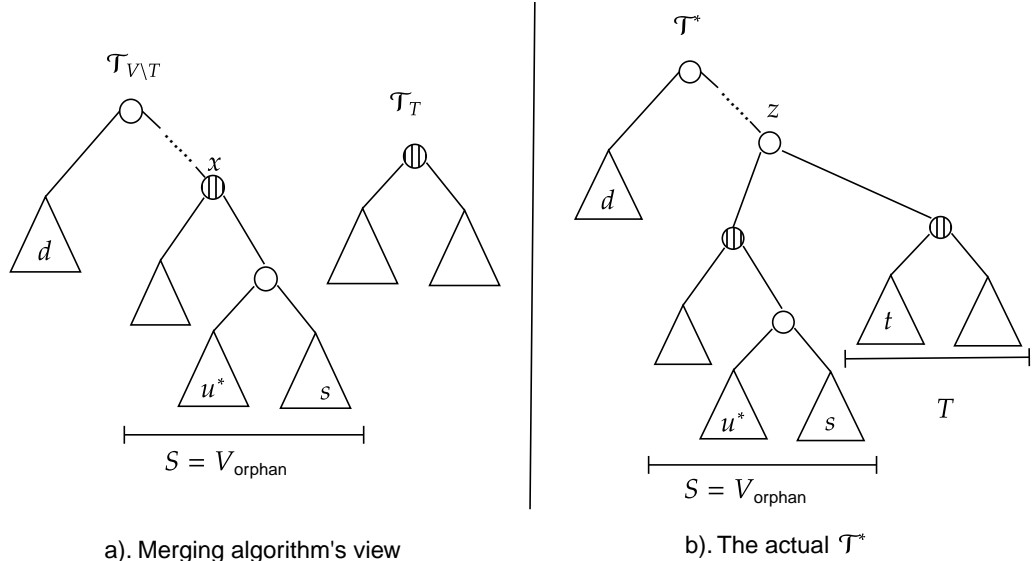

a). Merging algorithm's view

b). The actual $\mathcal{T}^*$

*Figure 10.* An illustration of the algorithm that merges $\mathcal{T}^*(V \setminus T)$ and $\mathcal{T}^*(T)$ The shaded internal node is the "actual" node to split $S$ and $T$ in $\mathcal{T}^*$. If the oracle is correct, $s$ does *not* split away from $(u^*, t)$, but $d$ *does* split away from $(u^*, t)$. The size of $T$ is large enough to overcome the adversarial noise.

### H.3. Finalizing the proof of Theorem 11

We now move to prove Theorem 11 for our partial tree construction. We first bound the number of possible internal nodes as $n$ by Fact A.1. Therefore, we can apply a union bound on *all* splits for Lemma H.1, and argue that with high probability, the events of Lemma H.1 hold for every partition. Furthermore, conditioning on Lemma H.1 always holds across the splits, we can again apply a union bound to show that Lemma H.8 holds across all internal nodes in the partial tree construction with high probability. We condition on the high probability events of Lemma H.1 and Lemma H.8 across the internal nodes for the rest of the proof.

**Proof of efficiency.** We now show that the depth of recursive calls on Lemma H.1 is $O(\log^3 n)$, i.e., the longest sequence of recursive calls induced by any fixed $\widetilde{V}$ is at most $O(\log^3 n)$ in Algorithm 8. To see this, consider any set of vertices $\widetilde{V}$

with size $\tilde{n}$, and we look into *three* levels of splits on $\widetilde{V}$. Suppose the vertices sets are $\widetilde{V} \to (\widetilde{V}_1, \widetilde{V}_2, \widetilde{V}_3, \widetilde{V}_4, \widetilde{V}_5, \widetilde{V}_6, \widetilde{V}_7, \widetilde{V}_8)$. We claim that there is

$$\max_i \left( \{ \left| \widetilde{V}_i \right| \}_{i=1}^8 \right) \leq (1 - \frac{1}{10000 \log^2 \tilde{n}}) \cdot \tilde{n}.$$

We prove the above statement by using Lemma H.1. Conditioning on the high probability event of Lemma H.1, if the first split of $\widetilde{V}$ falls into the condition of $\widetilde{V} = V_H$, then by Item 4(b)i and Item 4(b)ii of Lemma H.1, the balanceness of sizes already follows. Otherwise, we can have the following cases

- If we enter the case of Item 4b, note that we have $|T| \geq \frac{\tilde{n}}{200}$. As such, in the first split $\widetilde{V} \to (\widetilde{V} \setminus T, T)$, we already have $\left| \widetilde{V} \setminus T \right| \leq \frac{199}{200} \cdot \tilde{n} \leq (1 - \frac{1}{10000 \log^2 \tilde{n}}) \cdot \tilde{n}$. The size of $T$ might be large; however, it must have $\widetilde{V} = V_H$ for the first split on $T$ (by Algorithm 8). Therefore, in the next iteration, the maximum size is at most $(1 - \frac{1}{10000 \log^2 \tilde{n}}) \cdot \tilde{n}$ as well.

- If we enter the case of Item 4a, note that the set $T$ of the current iteration is of size at most $\frac{\tilde{n}}{100} \leq (1 - \frac{1}{10000 \log^2 \tilde{n}}) \cdot \tilde{n}$. Furthermore, in the next iteration, the $V^{\text{orphan}}$ set accounts for at least $\frac{99}{100}$ fraction of vertices. Hence, we can apply Lemmas H.6 and H.7 and argue that the split will be in the case of Item 4b with $T_2$ as the new set $T^*$. Now, we have $\widetilde{V} \setminus (T \cup T_2)$ with size at most $\frac{\tilde{n}}{100} \leq (1 - \frac{1}{10000 \log^2 \tilde{n}}) \cdot \tilde{n}$. The new set $T_2 \cap (\widetilde{V} \setminus T)$ might be large, but since it forms a *single* maximal subtree in $\mathcal{T}(T_2 \cap (\widetilde{V} \setminus T))$, there is $V_H = \widetilde{V}$, and in the *third* iteration, the maximum size is going be to at most $(1 - \frac{1}{10000 \log^2 \tilde{n}}) \cdot \tilde{n}$, as desired.

Since the size reduces by a $(1 - \frac{1}{10000 \log^2 \tilde{n}})$ factor for every three level of splits, after $60000 \log^3 \tilde{n}$ recursive calls, we have

$$\text{remaining size} \leq \tilde{n} \cdot (1 - \frac{1}{10000 \log^2 \tilde{n}})^{20000 \log^3 \tilde{n}}$$
$$\leq \tilde{n} \cdot \exp\left(-2 \log \tilde{n}\right) \leq O(1),$$

to which point the remaining vertices will be collapsed to a super-vertex by our algorithm. Therefore, since $\tilde{n} \leq n$, the longest sequence of dependent calls is at most $O(\log^3 n)$.

Finally, to complete the proof of efficiency, note that by Lemma H.1, the runtime for each call of Algorithm 12 is $O(n_H^2 \cdot \log n)$. The tree has depth at most $\log^3 n$; and at each level, the total number of runtime is at most $O(n^2 \cdot \log n)$ since we have $\sum n_H \leq n$. Similarly, each call of the merging algorithm will happen only after the split algorithm, which causes an overall $O(n^2)$ runtime overhead on any level. Therefore, the total runtime is bounded by $O(n^2 \cdot \log^4 n) = O(n^2 \cdot \text{polylog } n)$, as desired.

**Proof of correctness.**  We inductively prove the correctness of Theorem 11. On the level of the leaves in Algorithm 8, by Lemma H.1, if the leave contains more than one vertex, it must be a composable set with out-degree at most 2 in $\mathcal{T}^*$, and exactly one of them connecting to a parent node, and the other connecting to the sibling of the orphaned vertex. As such, when we merge two components $X$, $Y$ in which at least one of them is a super-vertex, we can guarantee the out-degree is still at most 2, and the LCA of $X$ and $Y$ induces the same vertices as on $\mathcal{T}^*(\text{LCA}_{\mathcal{T}^*}(X \cup Y))$, which implies the partial tree is weakly consistent with $\mathcal{T}^*(X \cup Y)$. On the other hand, when merging two components who are both *not* super-vertices, we can use Lemma H.8, and the assumptions of weak consistency come from the guarantees on previous partitions. Therefore, the weak consistency inductively applies to every level of the merging process, which gives the desired correctness guarantee.

# I. Discussions about Additional Settings for Our Algorithms

We discuss our algorithms in additional settings, which include general success probability (other than $9/10$) and splitting oracle for *approximately optimal* HC trees.

## I.1. General success probabilities

We use a success probability of $9/10$ in our algorithms for technical convenience. Here, we discuss algorithms with more general success probabilities. We remark that due to *adversarial* incorrect answers, our algorithm cannot work with $\frac{1}{2} + \varepsilon$

success probability for arbitrary $\varepsilon$. In fact, it is unclear whether *any* algorithm would work with $\frac{1}{2} + \varepsilon$ success probability and *adversarial* incorrect answers. Concretely, suppose that in the optimal HC tree, the first cut is balanced with size $(n/2, n/2)$. This appears to be a quite easy example. Now, let us fix a vertex $u$ and determine whether a vertex $v$ is on the same side of $u$. If $v$ is on the same side of $u$, there are $n/2$ vertices $w \in V$ such that $w$ splits away from $(u, v)$; conversely, if $v$ is on the opposite side, there is no such $w$ vertex. However, due to *adversarial* incorrect answers, we can report $(n/2 - n/3 = n/6)$ such $w$ vertices in the former case, and $n/3$ such w vertices in the latter case. As such, in the above example, the correct probability for the oracle must be at least $3/4$ to get anything meaningful. Finally, we remark our algorithm would work for any success probability $\frac{1}{2} + C$ for sufficiently large $C = \Omega(1)$: all the analysis will go through with changes in the constants. Furthermore, if we deal with *random* incorrect answers instead, we will be able to work with $\frac{1}{2} + \varepsilon$ success probability for any $\varepsilon = \Omega(1/n)$.

Finally, we give a remark on the discrepancy between success probabilities between learning-augmented HC and other learning-augmented graph algorithms. In some graph algorithm, e.g., in (Braverman et al., 2024; Cohen-Addad et al., 2024; Dong et al., 2025), the learning-augmented oracle could work with $1/2 + C$ probability for any $C = \Omega(1)$. The reason our algorithm should work with a sufficiently high success probability is due to the hierarchical structure. For instance, in the paper that studied learning-augmented max-cut (e.g. (Dong et al., 2025)), the errors in the algorithm are "one shot". However, in the HC problem, if the construction of the partial tree is wrong at any level, the error will propagate to all subsequent nodes, and it is not clear how to control the error if this happens. Therefore, a constant success probability sufficiently larger than $1/2$ is necessary.

### I.2. Splitting oracle with approximately optimal HC trees

A natural extension of our algorithms is to explore HC algorithms with splitting oracles from an *approximately* optimal HC tree. In other words, for a triplet of vertices $(u, v, w)$, the oracle $\mathcal{O}$ answers the "splitting away" query based on an HC tree $\mathcal{T}$ that achieves $\alpha$-approximation of the optimal tree $\mathcal{T}^*$. We remark that our algorithms based on the strongly consistent partial HC trees (i.e., the algorithms of Theorems 1, 2 and 21) could work with approximation HC trees. In particular, if the splitting oracle is constructed from an $\alpha$-approximation HC tree $\mathcal{T}$, our algorithm will produce HC trees with an extra $O(\alpha)$ factor on the approximation guarantees.

On the other hand, however, it is not immediately clear whether our algorithms based on weakly consistent partial HC trees could work for oracles from approximate HC trees. The main difficulty here is that to analyze the revenue decrement induced by the weakly consistent partial HC tree, we need to prove Lemma D.4 that characterizes the revenue structure of the optimal tree. We proved the statement by showing that if the statement is not true, we can increase the revenue, which forms a contradiction with the optimal HC tree (see Claim D.2 for details). We cannot easily argue that the same structural statement with an approximately optimal HC tree. This could be an interesting problem to resolve for future work.

## J. Splitting Oracle and Learning Theory

In this section, we offer formal learning theory analysis for learning a splitting oracle for the learning-augmented hierarchical clustering problem. In particular, we utilize the PAC learning framework to show that a high-quality predictor can be efficiently learned, provided that the input instances are drawn from a specific distribution. We remark that similar results have been shown in other settings by (Izzo et al., 2021; Chen et al., 2022c; Ergun et al., 2022; Grigorescu et al., 2022). Thus although the results of this section are by now standard techniques, they still provide an end-to-end framework for designing learning-augmented algorithms.

First, we suppose that there exists an underlying distribution $\mathcal{D}$, from which our input is drawn. In particular, $\mathcal{D}$ generates independent instances for hierarchical clustering, corresponding to the setting where similar instances of hierarchical clustering are being solved. Note that this is exactly the setting where we would like to apply learning-augmented algorithms. If there is instead generalization failure or distribution-shift, then inherently machine learning models will perform poorly.

Then our goal is to efficiently learn a predictor $f$ from a family $\mathcal{F}$ of possible functions, where the input to each predictor $f$ is a weighted undirected graph $G = (V, E, w)$ and three specific nodes, and the output is a feature vector. We remark that each input instance $G$ can be encoded as a vector in $\mathbb{R}^{n^2+n}$, by first considering the weighted $n \times n$ adjacency matrix of the graph. We can then flatten the matrix into a vector of dimension $n^2$ and then append a 3-sparse binary vector of length $n$, corresponding to the three vertices in the input. We also assume that the output of $f$ has at most $n$ dimension, indicating a binary vector for which vertex should be split from the other two vertices.

We define a loss function $L : f \times G \to \mathbb{R}$, which intuitively defines how accurate a predictor $f$ performs on each input instance $G$. For example, $f$ can represent the splitting oracle on $G$ and $L$ can denote the number of inaccurate responses compared to the best hierarchical clustering on $G$.

Now our goal is to learn the function $f \in \mathcal{F}$, which minimizes the following objective:

$$\underset{G \sim \mathcal{D}, (x,y,z) \in V^3}{\mathbb{E}} [L(f_G(x, y, z))]. \tag{3}$$

Let $f^*$ be an optimal function in $\mathcal{F}$,, so that $f^* = \operatorname{argmin} \underset{G \sim \mathcal{D}, (x,y,z) \in V^3}{\mathbb{E}} [L(f_G(x, y, z))]$ is a minimizer of the above objective. Assuming that for each graph instance $G$, triplet $(x, y, z) \in V$, and each $f \in \mathcal{F}$, we can efficiently compute $f_G(x, y, z)$ as well as $L(f_G(x, y, z))$, say in polynomial time $T(n)$, then we have:

**Proposition 29.** *There exists an algorithm that uses* $\operatorname{poly}\left(T(n), \frac{1}{\varepsilon}\right)$ *samples and returns a function* $\hat{f}$, *such that with probability at least* $\frac{9}{10}$,

$$\underset{G \sim \mathcal{D}, (x,y,z) \in V^3}{\mathbb{E}} \left[L(\hat{f}_G(x, y, z))\right] \leq \min_f \underset{G \sim \mathcal{D}, (x,y,z) \in V^3}{\mathbb{E}} [L(f_G(x, y, z))] + \varepsilon.$$

In particular, Proposition 29 is a PAC-style result that bounds the number of samples necessary to achieve a good probability of learning an approximately-optimal function $\hat{f}$. The algorithm corresponding to Proposition 29 is straightforward; it is simply the empirical minimizer after a sufficient number of samples are drawn. To prove correctness, we first require the following definition of pseudo-dimension for a function class, which is a generalization of VC dimension to real-valued functions.

**Definition 30** (Pseudo-dimension, e.g., Definition 9 in (Lucic et al., 2017)). *Let* $\mathcal{X}$ *denote a ground set, and let* $\mathcal{F}$ *be a collection of functions mapping elements from* $\mathcal{X}$ *to the interval* $[0, 1]$. *Consider a fixed set* $S = \{x_1, \ldots, x_n\} \subset \mathcal{X}$, *a set of real numbers* $R = \{r_1, \ldots, r_n\}$, *where each* $r_i \in [0, 1]$, *and a function* $f \in \mathcal{F}$. *The subset* $S_f = \{x_i \in S \mid f(x_i) \geq r_i\}$ *is referred to as the induced subset of* $S$ *determined by the function* $f$ *and the real values* $R$. *We say that the set* $S$ *with associated values* $R$ *is shattered by* $\mathcal{F}$ *if the number of distinct induced subsets is* $|\{S_f \mid f \in \mathcal{F}\}| = 2^n$. *Then the pseudo-dimension of* $\mathcal{F}$ *is defined as the size of the largest subset of* $\mathcal{X}$ *that can be shattered by* $\mathcal{F}$ *(or it is infinite if no such maximum exists).*

Using pseudo-dimension, we can now present an accuracy-sample complexity trade-off for empirical risk minimization with and the number of necessary samples. First, we define $\mathcal{H}$ be the class of functions in $\mathcal{F}$ composed with $L$, i.e., $\mathcal{H} := \{L \circ f : f \in \mathcal{F}\}$. Moreover, by normalization, we can assume the range of $L$ is contained within $[0, 1]$. Then we have the following generalization bounds:

**Theorem 31.** *(Anthony & Bartlett, 1999) Let* $\mathcal{D}$ *be a distribution over problem instances in* $\mathcal{G}$, *and let* $\mathcal{H}$ *be a class of functions* $h : \mathcal{G} \to [0, 1]$ *with pseudo-dimension* $d_{\mathcal{G}}$. *Consider* $t$ *i.i.d. samples* $G_1, G_2, \ldots, G_t$ *drawn from* $\mathcal{D}$. *There exists a universal constant* $c_0$ *such that for any* $\varepsilon > 0$, *if* $t \geq c_0 \cdot \frac{d_{\mathcal{H}}}{\varepsilon^2}$, *then for all* $h \in \mathcal{H}$, *we have the following with probability at least* $\frac{9}{10}$:

$$\left| \frac{1}{t} \sum_{i=1}^{t} h(G_i) - \underset{G \sim \mathcal{D}}{\mathbb{E}} [h](G) \right| \leq \varepsilon.$$

We have the following immediate corollary by applying the triangle inequality.

**Corollary 32.** *Let* $G_1, \ldots, G_t$ *be a set of independent samples from* $\mathcal{D}$ *and let* $\hat{h} \in \mathcal{H}$ *be a function that minimizes* $\frac{1}{t} \sum_{i=1}^{t} h(G_i)$. *If the number of samples* $t$ *is chosen as in Theorem 31, then with probability at least* $\frac{9}{10}$,

$$\underset{G \sim \mathcal{D}}{\mathbb{E}} \left[\hat{h}(G)\right] \leq \min \underset{G \sim \mathcal{D}}{\mathbb{E}} [h^*(G)] + 2\varepsilon.$$

Thus, the main question is to analyze the pseudo-dimension of our function class $\mathcal{H}$. To that end, we first relate the pseudo-dimension to the VC dimension of a related class of threshold functions.

**Lemma J.1** (Pseudo-dimension to VC dimension, Lemma 10 in (Lucic et al., 2017)). *For any* $h \in \mathcal{H}$, *let* $B_h$ *denote the indicator function of the threshold function, i.e.,* $B_h(x, y) = sgn(h(x) - y)$. *Then the pseudo-dimension of* $\mathcal{H}$ *equals the VC-dimension of the subgraph class* $B_{\mathcal{H}} = \{B_h \mid h \in \mathcal{H}\}$.

What remains is to bound the VC dimension of the function to compute in the class, which follows from the following standard result.

**Lemma J.2** (Theorem 8.14 in (Anthony & Bartlett, 1999)). *Let $\tau : \mathbb{R}^a \times \mathbb{R}^b \to \{0,1\}$, defining the class*

$$\mathcal{T} = \{x \mapsto \tau(\theta, x) : \theta \in \mathbb{R}^a\}.$$

*Assume that any function $\tau$ can be computed by an algorithm that takes as input the pair $(\theta, x) \in \mathbb{R}^a \times \mathbb{R}^b$ and produces the value $\tau(\theta, x)$ after performing no more than $t$ of the following operations:*

- *arithmetic operations $+, -, \times, /$ on real numbers,*

- *comparisons involving $>, \geq, <, \leq, =$, and outputting the result of such comparisons,*

- *outputting $0$ or $1$.*

*Then, the VC dimension of $\mathcal{T}$ is bounded by $O(a^2 t^2 + t^2 a \log a)$.*

We can now apply these results to prove Proposition 29 by instantiating Lemma J.2 with the computational complexity of evaluating any function in the class $\mathcal{H}$.

*Proof of Proposition 29.* From Lemma J.1, we know that the pseudo-dimension of $\mathcal{H}$ is equivalent to the VC dimension of the threshold functions defined by $\mathcal{H}$. Next, from Lemma J.2, we observe that the VC dimension of the relevant class of threshold functions is polynomial in the computational complexity of evaluating a function from the class. In other words, Lemma J.2 implies that the VC dimension of $B_{\mathcal{H}}$ (as defined in Lemma J.1) is polynomial in the number of arithmetic operations required to compute the threshold function corresponding to some $h \in \mathcal{H}$. According to our definition, this quantity is polynomial in $T(n)$. Thus, the pseudo-dimension of $\mathcal{H}$ is also polynomial in $T(n)$, and the desired result follows. □

Note that we can initialize Proposition 29 with various oracles, in terms of the input and output predictions. Indeed, if each function in the family of oracles we are considering can be computed efficiently, then Proposition 29 guarantees that a polynomial number of samples is sufficient to learn a nearly optimal oracle.

