# OpenReview forum: "Learning-Augmented Hierarchical Clustering"
_ICML.cc/2025/Conference — ICML 2025 poster_

### Official Review · Reviewer_Q4HP · 2025-03-11

**Overall Recommendation:** 4

**Summary:**

Hierarchical clustering, wherein vertices (representing items in a dataset) are grouped into clusters of increasing refinement following a tree structure, is a well-motivated procedure of interest to practitioners and gives rise to interesting theoretical problems. In particular, several of the most prominent objectives in hierarchical clustering admit impossibility results. In this work, the authors show that having even only marginally informative predictions allows one to overcome these impossibility results. They consider a “splitting oracle,” which given an input triple of three vertices outputs a single vertex. With probability $p$, this output vertex is the first vertex that would be separated from the other two in some (near-)optimal hierarchical clustering tree. With probability $1-p$, the oracle outputs an arbitrary vertex. For small constant values of $p$, e.g. $p= ½ + \varepsilon$ for $\varepsilon > 0$ fixed, this oracle gives only a small amount of additional information to the algorithm. Yet the authors show that this is enough to drastically reduce the hardness of the problem.

**Claims And Evidence:**

Claim: Learning-augmented algorithms can efficiently approximate optimal HC trees for prominent HC objectives, breaking impossibility barriers.

Evidence:
- In Thm 1, they show that for $p=9/10$, there exists a polynomial-time algorithm that makes $O(n^3)$ queries and produces a constant approximation to the Dasgupta (Das) optimal HC tree.
- In Thm 2 they improve the running time, showing that an $\tilde{O}(n^3)$ time algorithm that makes $O(n^3)$ queries produces an $O(\sqrt{\log\log(n)}$ approximation to the Das optimal HC tree.
- In Thm 3 they show an $\tilde{O}(n^2)$ time algorithm that makes $O(n^2)$ queries produces a constant approximation to the Moseley-Wang (MW) optimal HC tree.

**Essential References Not Discussed:**

I do not know of any essential references that are not discussed.

**Experimental Designs Or Analyses:**

Not applicable; this is a theoretical work and does not include experiments.

**Methods And Evaluation Criteria:**

Not applicable; this is a theoretical work and does not include experiments.

**Other Comments Or Suggestions:**

The assumption of large success probability p=9/10 and highlights from the Discussion of general success probabilities (Appendix I.1) should be emphasized earlier in the text. When I first read Thms. 1-3, I assumed based on the exposition that they held for any arbitrarily small yet constant probability of success (e.g. ½ + \varepsilon). After reading Appendix I.1, it is now my understanding that for arbitrary (adversarial) incorrect responses, a significantly large probability of success is necessary (see Questions below), while for randomly-chosen incorrect responses probability $½ + \varepsilon$ does suffice. Is this understanding correct?

In the exposition, $\delta$ is used to denote success probability, while later $p$ is used. I would suggest picking one of the two and using it consistently, as there's already several symbols floating around that describe success probabilities (e.g. $\varepsilon$ and $C$). If I am confused and actually $\delta$ and $p$ have different meanings, please correct me.

**Other Strengths And Weaknesses:**

Strengths:
- This work clearly establishes that predictions fundamentally decrease the hardness of well-motivated hierarchical clustering problems.
- Several of the algorithms proposed (esp. Algo. 2 and 3) seem like they could be implemented in practice. It would be interesting to compare the performance of these theoretically-grounded algorithms with heuristics used for HC in practice.
- The partial HC trees introduced could be objects of further study. They seem well-motivated for practitioners, who may be content to learn a hierarchical clustering that encompasses only the majority of vertices, and leaves some reasonably-sized sub-population ambiguous.

I find no major weaknesses in the work, but I did not check the proofs included in the supplementary material. I do think the assumption of large probability of success (p=9/10) was surprising to me, as in other works (e.g. max-cut and max IS with predictions) the emphasis is on overcoming barriers using arbitrarily small probability of success. I would suggest that an abridged discussion of this assumption be presented closer to the statements of the main theorems, as currently it could be overlooked in the main body of the paper.

**Questions For Authors:**

I already feel the paper is compelling. These questions are mostly for my own understanding.

1. Is there an intuitive reason why the results established in this work only apply to the Mosely-Wang (MW) and Dasgupta (Das) objectives, as opposed to the Cohen-Adad  (CA) objective? In particular, it is striking that the impossibility results for oblivious algorithms for both MW and Das stem from the Small Set Expansion hypothesis, whereas impossibility results for the CAobjective stem from UCG. Is the fact that your results focus on MW and Das implicitly connected to this divide?

2. In Appendix I.1, I am confused about some of the remarks about general success probabilities. The penultimate line states “...we remark our algorithm would work for any success probability ½ + C for sufficiently large $C=\Omega(1)$.” I am confused by the meaning of $C=\Omega(1)$ when discussing success probabilities, as this does not seem to imply any nontrivial lower bound on C. Do the authors have a more specific lower bound on ½ + C in mind? The main results of the paper imply that ½ + C = 9/10 suffices, while the counter-example in Appendix I.1 implies that ½ + C must be at least ¾.

**Relation To Broader Scientific Literature:**

Learning-augmented algorithms/algorithms with predictions have been an area of expanding recent study. Several works in this area have focused on showing that even marginally-informative predictions (i.e. those whose probability of giving a correct response is arbitrarily close to ½) can break hardness barriers for classical well-studied problems, such as max-cut, independent set.

Cohen-Addad, Vincent, et al. "Max-Cut with $\epsilon $-Accurate Predictions." arXiv preprint arXiv:2402.18263 (2024).

Braverman, Vladimir, et al. "Learning-augmented maximum independent set." arXiv preprint arXiv:2407.11364 (2024).

This work similarly studies how predictions can break impossibility results for hierarchical clustering.

**Theoretical Claims:**

I did not check any of the proofs in the supplementary material.

---

> ### Author Rebuttal · Authors · 2025-04-01
>
> Thank you for your careful review, positive evaluation, and helpful questions. Our responses and clarifications are as follows.
>
> > Relation To Broader Scientific Literature
>
> Thanks for pointing out the additional papers related to our work. We will add them and some discussions about the connections.
>
> >  I do think the assumption of large probability of success (p=9/10) was surprising to me, as in other works (e.g. max-cut and max IS with predictions) the emphasis is on overcoming barriers using arbitrarily small probability of success. I would suggest that an abridged discussion of this assumption be presented closer to the statements of the main theorems, as currently it could be overlooked in the main body of the paper.
>
> > After reading Appendix I.1, it is now my understanding that for arbitrary (adversarial) incorrect responses, a significantly large probability of success is necessary (see Questions below), while for randomly-chosen incorrect responses, probability $½+\varepsilon$ does suffice. Is this understanding correct?
>
> > (Q2) In Appendix I.1, I am confused about some of the remarks about general success probabilities. The penultimate line states “...we remark our algorithm would work for any success probability ½ + C for sufficiently large C.
>
> We answer these concerns and questions together here since all of them are related to the success probability of the learning-augmented oracle. We first want to say that the reviewer’s understanding of our algorithm is correct: in the setting of adversarial noise, the success probability should be at least $1/2+C\geq 3/4$, and $9/10$ suffices. On the other hand, if the errors of the oracle are random, we should be able to ensure correctness with $1/2+\varepsilon$ for any constant $\varepsilon>0$ (we forgot to mention “constant” in the current Appendix I.1 and will add that in the final version).
>
> The reason our algorithm should work with a sufficiently high success probability is exactly due to the hierarchical structure of the problem. In both max-cut and MIS papers (which we are sufficiently familiar with), the errors in the algorithm are ‘one-shot’. However, in the HC problem, if the construction of the partial tree is wrong at any level, the error will propagate to all subsequent nodes, and it is not clear how one could control the error if this happens. Furthermore, the oracles in the max-cut and MIS papers are about ‘memberships’, while the oracle in our setting is about ‘relationships’. The latter type of oracle does not give *any* trivial solution or even a ‘partial solution to be fixed’ (like the case in the MIS paper). Therefore, a constant success probability much larger than $1/2$ is necessary.
>
> > In the exposition, $\delta$ is used to denote success probability, while later $p$ is used.
>
> Thanks for the catch! We will unify this and use $p$ as the success probability.
>
> > Is there an intuitive reason why the results established in this work only apply to the Mosely-Wang (MW) and Dasgupta (Das) objectives, as opposed to the Cohen-Adad (CA) objective? In particular, it is striking that the impossibility results for oblivious algorithms for both MW and Das stem from the Small Set Expansion hypothesis, whereas impossibility results for the CAobjective stem from UCG. Is the fact that your results focus on MW and Das implicitly connected to this divide?
>
> We do not believe our results are connected to the division between the UGC vs. small set expansion. The reason is that our constructions for the weakly and strongly consistent partial trees are objective-oblivious, and we can also construct these partial trees w.r.t. the Cohen-Addad objective. The reason for us not getting results for the Cohen-Addad objective is due to the limited bandwidth, we are not sure how the error would propagate in the CA objective. It will be an interesting direction to explore as a future step.

---

> > ### Comment · Reviewer_Q4HP · 2025-04-02
> >
> > Thank you for answering my questions--I will maintain my score. I would like to see this paper at ICML.

---

### Official Review · Reviewer_vBZ1 · 2025-03-13

**Overall Recommendation:** 4

**Summary:**

The paper studies learning-augmented algorithms for hierarchical clustering (HC). In this problem, a set of data points is given along with a similarity measure, which induces a weighted undirected graph $G=(V,E,w)$. The goal is to partition the points/vertices into a binary tree that captures the hierarchical relationships within the data. The paper considers two objective functions: the Dasgupta cost minimization objective (Dasgupta, 2016) and the Moseley-Wang revenue maximization objective (Moseley & Wang, 2017). For Dasgupta’s objective, prior work (Charikar & Chatziafratis, 2017a; Roy & Pokutta, 2017) showed that there is no polynomial-time algorithm that could achieve any constant-factor approximation under the Small Set Expansion hypothesis. The best-known algorithm achieves an $O(\sqrt{\log n})$-approximation (Charikar & Chatziafratis, 2017b; Cohen-Addad et al., 2019). For the Moseley-Wang objective, (Chatziafratis et al., 2020b) showed that achieving a $(1 − C)$-approximation is impossible under the Small Set Expansion hypothesis for some fixed constant $C \in (0, 1)$.

Motivated by the practical aspect of the problem, the paper considers HC with learning-augmented oracles. Specifically, it introduces a splitting oracle that, when queried for a triplet of vertices, outputs the vertex that is first separated away from the other two with respect to an optimal HC tree. The oracle returns the correct answer with probability $p$ and returns arbitrarily with the remaining probability. Using such a predictor, for Dasgupta’s objective, the paper achieves a constant-factor approximation in polynomial time (albeit with a large exponent), and a more practical $O(\sqrt{\log \log n})$-approximation that improves upon the state-of-the-art. For the Moseley-Wang objective, the paper achieves any constant factor approximation in polynomial time, which overcomes prior impossibility barriers. The key technical contribution of the paper is the introduction of partial HC trees, which capture structural properties of optimal HC trees. The paper shows that these partial HC trees can be efficiently constructed given access to the splitting oracle, independent of the specific objective function.

The paper further explores sublinear algorithms for HC with splitting oracles (in the streaming and PRAM model), and obtains results that outperform their non-learning counterparts.

**Claims And Evidence:**

This paper is purely theoretical. All claims are formally stated as theorems or lemmas and are supported by rigorous proofs.

**Essential References Not Discussed:**

None.

**Experimental Designs Or Analyses:**

NA

**Methods And Evaluation Criteria:**

This paper is purely theoretical, and the proposed methods are analyzed through rigorous mathematical proofs. The evaluation is based on formal theoretical criteria, which are appropriate for the problem at hand.

**Other Comments Or Suggestions:**

Typos:

- Page 1, Line 49, right column: $1+C$ -> $1-C$
- Page 3, Line 149, right column: deonte -> denote
- Page 4, Line 168, left column: I think $level_\mathcal{T}(\cdot)$ is not formally defined.
- Page 4, Definition 5: Should $\mathcal{T}$ be $\mathcal{T}^*$?
- Page 5, Definition 8: ‘the same set of vertex’ -> ‘the same set of vertices’; also applies to Definition 9.
- Page 7, Line 356, left column: I think a period is missing.

**Other Strengths And Weaknesses:**

Strengths:

- Hierarchical clustering is a fundamental and extensively studied problem. The paper is well-motivated to study this problem in the learning-augmented setting with recent advances in machine learning. The paper is a good addition to the literature on learning-augmented algorithms.

- The technical contributions are solid. The theoretical results outperform prior state-of-the-art and even impossibility barriers.

- The paper is extremely well-written. The technical sections are clearly structured and easy to follow.

Weaknesses:

- the error probability of the splitting oracle cannot be too large, i.e., the success probability of the oracle is assumed to be at least 1/2+C, for sufficiently large $C=\Omega(1)$.

- Since the paper is motivated by the recent advances in machine learning, I think some empirical results for the proposed algorithms would strengthen the evaluation.

**Questions For Authors:**

Q1: I am curious about the practical aspects of the splitting oracle in the streaming setting. While the paper assumes that the oracle is given offline and queried as a black box, is there any possibility that the oracle can be implemented using small space?

Q2: Do you see any chance for your algorithms to be made sublinear time?

Q3: In line 2813, you mentioned that for *random* incorrect answers, you will be able to work with $1/2+\epsilon$ success probability for any $\epsilon>0$. Could you give more details on that? Specifically, how does the resulting approximation ratio depend on  $\epsilon$?

**Relation To Broader Scientific Literature:**

The paper studies learning-augmented algorithms for hierarchical clustering. Hierarchical clustering is a well-studied problem. There are two main objective functions considered in prior work: the Dasgupta cost minimization objective (Dasgupta, 2016) and the Moseley-Wang revenue maximization objective (Moseley & Wang, 2017). The paper provides improved algorithms with respect to these two objectives in the learning-augmented framework, showing the power of machine learning oracles. Therefore, the paper is a good addition to the literature on learning-augmented algorithms.

**Theoretical Claims:**

I have checked all the content in the main text (i.e., the first 8 pages). However, I did not thoroughly verify all the proofs in the appendix. I reviewed the technical overview and sketched the proofs, and they appear to be correct.

---

> ### Author Rebuttal · Authors · 2025-04-01
>
> Thank you for your insightful questions and positive evaluation. Our responses to the questions are as follows.
>
> > The error probability of the splitting oracle cannot be too large
>
> We agree with the reviewer that this is a limitation of our algorithm, and due to the combinatorial nature of many subroutines, it’s unclear exactly what minimal error can be tolerated on this front. However, we hope that modern ML models can achieve decent accuracy across the entirety of the similarity graph, which may consist of a number of dissimilar items.
>
> We agree that additional experiments could add value to the paper. We have conducted preliminary experiments on social network-like graphs and plan to expand the discussion accordingly.
>
> > Typos
>
> Thanks for pointing these out. We will make changes accordingly.
>
> > Implementing the oracle in small space
>
> This is a great question. We are not exactly sure how the oracle could be implemented in small space. However, since the input of the oracle is only triplets of vertices, and the output is essentially an ordering, the model size could be very small (say, for a 3-layer neural network, we could implement with 3xNx1 with one N-node hidden layer). Of course, to ensure good performances, the actual model might need to be bigger. We could also use cloud-based services, and the aim is to reduce the memory cost on local machines. For instance, we could query the triplets to LLMs and implement the streaming algorithm on our local machine. Exploring whether these ideas work is an interesting direction to pursue.
>
> > Potential sublinear time algorithms
>
> We believe it is possible to revise our weakly consistent partial tree algorithm to achieve *sublinear* (i.e., $o(n^2)$) time. The `counterpart testing’ subroutine, as we could see in the construction of the strongly consistent partial tree, could be made $O(n \log{n})$ time. Due to the balanced partition, this part should take $\tilde{O}(n)$ time across the algorithm. However, the ‘root test’, which is crucial to the correctness of our algorithm, still requires $O(n^2)$ time. Making this part sublinear time appears to require more work, and is an interesting direction to explore as future research.
>
> > Performance of the algorithm with random noise
>
> Due to the persistent noise and the combinatorial nature of our algorithm, it is difficult to get something in the form of query-quality trade-offs. If the noises are random, for the adversarial example we mentioned in appendix I.1, as long as $\varepsilon=\Omega(1/n)$, we should be able to obtain enough signals to distinguish the cases for the two sides. We also realize that the way it is currently written is not precise, and we meant to say that if the noise is random, we could guarantee correctness for any *constant* $\varepsilon$ (as opposed to some fixed $C\geq 1/4$). We will make changes about this in future versions.

---

### Official Review · Reviewer_xyMS · 2025-03-13

**Overall Recommendation:** 2

**Summary:**

This paper explores hierarchical clustering in a learning-augmented framework. Unlike other clustering methods such as $k$-means or $k$-medians, hierarchical clustering constructs a clustering tree $\mathcal{T}$ to represent similarity across all item pairs, and does not have a predetermined target number of clusters. The quality of $\mathcal{T}$ can be assessed using various metrics; this work focuses on the Dasgupta and Moseley-Wang objectives.

The study assumes access to a splitting oracle that, given any triplet of items $(u,v,w)$, predicts which one should split away first in the optimal clustering tree $\mathcal{T}^*$. The oracle’s predictions are assumed to be correct with probability $9/10$ and arbitrary otherwise. The paper investigates how leveraging these predictions enables the design of efficient polynomial-time approximation algorithms.

For the Dasgupta objective, the best-known polynomial-time algorithm achieves an $O(\sqrt{\log n})$-approximation, and no polynomial-time algorithm can guarantee a constant approximation. The authors overcome these limitations by utilizing the splitting oracle. They show that with $O(n^3)$ oracle queries, it is possible to achieve with high probability the following
- A constant-factor approximation in $O(n^{50002})$ time, instead of exponential time in the setting without predictions. While this result is not practical because of the extremely high polynomial degree, it is interesting from a theoretical point of view as it shows that using an oracle allows breaking classical impossibility results.
- An $O(\sqrt{\log \log n})$-approximation in $O(n^4)$ time.

For the Moseley-Wang objective, the authors propose a $(1-o(1))$-approximation algorithm that requires $O(n^2)$ oracle queries and runs in $O(n^2 \text{poly} \log n)$ time. This surpasses the impossibility result of Chatziafratis et al. (2020) for the setting without predictions.

## Update after rebuttal
My primary concern with this paper is in the assumption that the oracle is correct with a fixed, known probability of 0.9. This assumption weakens the results, as it is unclear how the algorithm's performance scales with varying oracle accuracy. Moreover, the assumption that the success probability is known contradicts the core motivation behind learning-augmented algorithms, which is to design algorithms having a good performance without any knowledge of the oracle accuracy.

In their rebuttal, the authors state that their algorithms still work when the oracle is correct with probability $1/2 + \epsilon$, and gives uniformly random predictions with the remaining probability. They refer to this as "the standard model," but this is not correct. The standard model in algorithms with $\epsilon$-accurate predictions assumes that the oracle is correct with probability $\epsilon$ and adversarial otherwise, which is the model studied in the paper, and which requires $p=0.9$.

The authors also claim that the algorithm does not require precise knowledge of $p$, but only that $p \geq 1/2 + \Omega(1)$ for some sufficiently large $\Omega(1)$. However, without specifying the constant hidden in the $\Omega(1)$, this claim is trivial and uninformative, as the algorithm already works for $p \geq 0.9$. Thus it does not address my concern.

**Claims And Evidence:**

Yes

**Essential References Not Discussed:**

The paper gives a very good overview of state-of-the-art results of hierarchical clustering without predictions, which allows a good understanding of the proposed results and their relevance.

The paper deviates from the standard setting of learning-augmented algorithms, which assumes no guarantees on the quality of predictions. Instead, it considers a specific scenario where predictions are independently accurate with a certain probability, which is referred to in the literature as "algorithms with $\epsilon$-accurate predictions". While some prior works on similar settings (for other problems) are cited, there is no discussion on the model. Maybe this should be made more explicit in the introduction.

A very interesting aspect of the paper is that the algorithm actively decides when to query predictions, rather than receiving them as a fixed input as the standard setting in learning-augmented algorithms, and an objective is to have a small number of queries. The authors should emphasize this feature more and consider citing relevant works that explore similar settings. Such works include for example
- "Parsimonious Learning-Augmented Caching", Im et al. 2022
- "Algorithms for Caching and MTS with reduced number of predictions", Sadek et al. 2024
- "Learning-Augmented Priority Queues", Benomar et al. 2024

**Experimental Designs Or Analyses:**

NA

**Methods And Evaluation Criteria:**

Yes

**Other Comments Or Suggestions:**

The accuracy of the oracle is denoted $1-\delta$ in Lign 72 (right column), while it is denoted $p$ in the rest of the paper

**Other Strengths And Weaknesses:**

**Strengths**.
- The setting is very interesting, and the oracle model makes sense
- The authors consider different objective functions and improve upon standard results for all of them
- The problem is technically challenging, yet the paper provides many strong results.

Weaknesses:
- Assuming an oracle with accuracy $p = 0.9$ is quite restrictive. The authors mention in the appendix that their results can be generalized for $p = 1 + \epsilon$ with $\epsilon$ above some unknown constant—though this is trivial, as the results already hold for $p = 0.9$. However, I find their arguments unconvincing. Understanding the dependency on $p$ is crucial for quantifying the impact of oracle accuracy. Additionally, for $p = 0.5$, the algorithm should be able to perform as good as standard hierarchical clustering methods without predictions (e.g., for the Dasgupta objective). This suggests the possibility of designing an algorithm that interpolates between the performance of the best algorithm without predictions, when $p$ is close to 1/2, and the optimal algorithm when the oracle is perfectly accurate. Moreover, the choice of $p=0.9$ is extremely arbitrary if the precise value $0.9$ is used nowhere in the proofs.
- The paper is difficult to read: Some sections/paragraphs are extremely wordy (Section B.1 for example), notations are used before being introduced, some paragraphs are very poorly written (for eg. L 403-412 (left column)), some sentences are very informal (L 813: 'this is a big “suppose” as we will see later'), ...

**Questions For Authors:**

1/ In Section 5, the complexity of Algorithm 3 seems quadratic and not near linear: the algorithm uses the construction of Theorem 11, which requires O(n^2) queries and $\Omega(n^2)$ time, then another sublinear operation. The overall complexity is near-quadratic, and not near-linear as the section title suggests. Can the authors clarify this?

2/ Can the authors give a reference or proof for the concentration bound of Proposition 13?
The bound they state is known to be true for Bernoulli random variables, i.e. taking values in the set {0,1}, and not for any random variables bounded in $[0,1]$ a.s.

**Relation To Broader Scientific Literature:**

The paper studies the problem of hierarchical clustering augmented with a splitting oracle.
To my knowledge, this is the first paper studying this problem. Therefore, there are no prior works proving results in the same setting.

The authors show that using the oracle allows significant improvement upon previous algorithms and results for the problem of hierarchical clustering, and even break impossibility results. The improvements are cited in my summary of the paper.

**Theoretical Claims:**

All the proofs are in the appendices. I skimmed through some proofs to grasp the main ideas and overall logic but did not examine any in detail.

---

> ### Author Rebuttal · Authors · 2025-04-01
>
> Thank you for your detailed review and insightful questions. Our responses and clarification are as below.
>
> > The paper deviates from the standard setting of learning-augmented algorithms, which assumes no guarantees on the quality of predictions.
>
> Although there is a body of literature that assumes no guarantees on the quality of predictions, we believe the setting with performance guarantees for the predictions is also quite standard, especially for graph-related problems. To that end, we have expanded the discussion on the “algorithms with $\epsilon$-accurate predictions”, including “Learning-Augmented Streaming Algorithms for Approximating Max-Cut” (DVP [ITCS’25]), “Learning-Augmented Approximation Algorithms for Maximum Cut and Related Problems” (COGLP [NeurIPS’24]), and “Learning-augmented Maximum Independent Set” (DBSW [APPROX’24]).
>
> We want to emphasize that since we study an *offline* problem (as opposed to an online one), we could always guarantee *robustness* studied in the online learning-augmented algorithm literature by running a separate algorithm without prediction. In the end, we could evaluate the costs of the two HC trees (with and without predictions) and output the one with the lower cost.
>
> > Discussions about “algorithm actively decides when to query predictions, rather than receiving them as a fixed input”
>
> We agree that our algorithm requires adaptivity, and we can add more discussion about this in future versions. Thanks for providing the related literature.
>
> > Assuming an oracle with p=0.9 accuracy is quite restrictive.
>
> First, we want to clarify that our discussion did not mention the error probability of $1+\varepsilon$ but rather $1/2+\varepsilon$ and $1/2+\Omega(1)$. Due to the offline nature of our problem, indeed, we could recover the best result for hierarchical clustering when the prediction is bad by simply running a separate algorithm without using any prediction. We also discussed a simple example (in I.1) that requires the correct probability to be at least 3/4. This is due to the combinatorial subroutines used in our algorithm
>
> > Accessibility.
>
> We thank the reviewer for the comments on readability. We have strived to make the paper readable by inserting multiple figures and explaining things in detail as much as possible, e.g., by providing intuition for both our algorithm and our analysis in the technical overview that is Appendix B. Although elements of our results have multiple moving parts and are complex to describe, we completely agree that accessibility is an important part of the paper. Thus, we will continue to make editorial passes over the document to improve presentation.
>
> > The accuracy of the oracle is denoted $1-\delta$ vs. $p$
>
> We thank the reviewer for pointing this out, and we have made changes accordingly.
>
> > In Section 5, the complexity of Algorithm 3 seems quadratic and not near linear.
>
> We view the input as the similarity graph with $\Omega(n^2)$ edges on $n$ vertices, so that the input size is $\Omega(n^2)$ and thus our algorithm that has runtime roughly quadratic in the number of vertices actually has runtime near-linear in the input size. We have clarified this point.
>
> > The bound in Proposition 13.
>
> We only deal with *discrete* random variables in this paper, and for all the applications of Proposition 13, we only used supports on $\{0,1\}$. For discrete random variables with supports as a (finite) set of value between $[0,1]$, the bound should be true also. We agree that Chernoff bounds on *continuous* random variables might be different, and we will make Proposition 13 more precise in later versions.

---

> > ### Comment · Reviewer_xyMS · 2025-04-03
> >
> > I thank the authors for their response.
> >
> > **$\epsilon$-accurate predictions.** The setting of $\epsilon$-accurate predictions is indeed quite standard, and I completely agree that studying this model makes sense. My point, however, was that it does not align with the classical learning-augmented framework. A brief discussion highlighting this distinction would be beneficial.
> >
> > **Robustness.** I agree that the algorithm can be easily adapted to ensure robustness and had no concerns on this point.
> >
> > **Probability $p=0.9$ of accuracy.** The $1+\epsilon$ in my review was a typo; thanks for noticing.
> > I remain unconvinced by the authors’ response regarding the choice of $p=0.9$. This value seems both very high and arbitrary. Why was $p=0.9$ chosen specifically over, say, $p=0.8$?
> >
> > The example in Appendix I.1 demonstrates that the proposed algorithm requires a success probability of at least 3/4, but this does not imply that no improvements are possible for $p = 1/2 + \epsilon$ for arbitrary $\epsilon > 0$. Rather, it suggests that better algorithms are needed to potentially perform well under weaker guarantees on $p$.
> >
> > Fixing the value of $p$ obscures important limitations of the algorithm. If the analysis only holds for $p\geq 1/2 + C$ for some known $C>0$, then it is crucial to see how the behaviour degrades as $p$ approaches the critical value $1/2 + C$, as this would help understanding the limitations of the algorithm and also help improve it in future work.
> >
> > Finally, since $p$ is fixed, the algorithm's design implicitly assumes prior knowledge of its exact value.. This is a limitation from the point of view of designing algorithms with $\epsilon$-accurate predictions. Prior work in this line of research, including those cited in the paper and the authors’ rebuttal, assumes that the accuracy of the ML model is either unknown or known only within a certain range rather than being precisely determined. Fixing $p$ in the paper also hides this important aspect.
> >
> > The assumption $p = 0.9$ is the major limitation of the paper in my opinion, as outlined above and in my review. I remain unconvinced by the authors' rebuttal regarding the justification for fixing the value of $p$. For these reasons, I maintain my score.

---

> > > ### Author Response · Authors · 2025-04-06
> > >
> > > Thank you for the continued correspondence in this discussion.
> > >
> > > > I remain unconvinced by the authors’ response regarding the choice of $p=0.9$. This value seems both very high and arbitrary. Why was $p=0.9$ chosen specifically over, say, $p=0.8$?
> > >
> > > We remark that our algorithm can actually handle success probability $\frac{1}{2}+\varepsilon$ for $\varepsilon=\Omega(1)$ in the standard model. However, in our more nuanced model, we can only tolerate a sufficiently large error probability. Specifically:
> > >
> > > - The main challenge is dealing with a sufficiently large success probability along with the presence of *adversarial* inputs when the answer is incorrect. In our model, when the oracle errs with probability $1-p$, we assume the answer is the worst-case scenario, potentially even adaptive, designed to maximally obstruct the algorithm's success. We remark that other works studying adversarial input upon prediction failures often require additional assumptions to deal with these cases, e.g., literature in learning-augmented $k$-means clustering generally that adversarial failures must actually be distributed in a "uniform" way. We do not make such an assumption here.
> > >
> > > - Mathematically, our analysis requires on combinatorial properties that relate the ratio of functions of $p$ and $1-p$, so we cannot achieve $p$ arbitrarily close to $0.5$.
> > >
> > > - If the incorrect answers are random (i.e, with probability $1-p$, toss a coin and answer a vertex  $x\in (u,v,w)$ that splits away from the other two), then our algorithm **works with any $\frac{1}{2}+\Omega(1)$ success probability**. This discussion is currently in Appendix I, and we will expand the discussion in future versions.
> > >
> > > - In particular, many of the previous paper that studied $\frac{1}{2}+\varepsilon$ success probability has more structured error that is consistent with the ‘random error’ model in our case.  For instance, in the paper “Learning-Augmented Approximation Algorithms for Maximum Cut and Related Problems” (COGLP [NeurIPS’24]) and “Learning-augmented Maximum Independent Set” (DBSW [APPROX’24]), when the oracle is wrong, the answer simply gets flipped, as opposed to looking into the algorithm’s state and give an adversarial answer. In fact, this property is crucial for correctness in the algorithm of “Learning-augmented Maximum Independent Set” (DBSW [APPROX’24]). **In these settings, our algorithm can indeed work with $1/2+\varepsilon$ probability for any $\varepsilon=\Omega(1)$.**
> > >
> > > - While it might be possible to develop improved algorithms specifically targeting the counterexample in Appendix I, the adversarial noise makes it unclear how such an algorithm would function effectively. For the application of deciding “whether vertex $v$ is in the same partition of $u$”, the sheer *noise* could overwhelm the *signal* in our example. As such, any algorithm with further improvements with *adversarial noise* would need to follow some other frameworks.
> > >
> > > - Finally, we remark that instead of knowing the exact value of $p$, we only need to know that $p$ has a lower bound, i.e., $p\geq\frac{1}{2}+\Omega(1)$ for some sufficiently large $\Omega(1)$.
> > >
> > > We thank the reviewer for their valuable insights, and we will incorporate a more detailed discussion about the counterexample and the difference between random and adversarial noise in future versions. We hope our response has addressed your questions about the failure probability settings.

---

### Official Review · Reviewer_HUGU · 2025-03-13

**Overall Recommendation:** 4

**Summary:**

Brief Summary of the Paper:

The paper introduces and studies learning-augmented algorithms for hierarchical clustering (HC) where the type of advice given comes in the form of a splitting oracle. This continues a long line of research on algorithms with ML predictions (or augmented with ML advice) and extends ideas to the context of hierarchical clustering.

The problem itself is Hierarchical Clustering where the output of the algo is a tree whose leaves are the given datapoints, and the input is a collections of points with pairwise similarities. The formal framework studied here is under the two objectives for Dasgupta's cost and Moseley-Wang reward objective. These two are complementary objectives (they add up to sth constant based on the input graph) and they try to encode the fact that similar points should be split as late in the hierarchy as possible.

The main contributions are 5 algorithms that leverage this abovementioned ML splitting oracle to achieve improved approximation guarantees for Dasgupta’s and Moseley-Wang’s objectives. Importantly, all the algorithms construct partial HC trees (strongly or weakly consistent with the optimal tree) using the oracle, then refine them using techniques like sparsest cut. Sparsest Cut was one of the first methods to be used for obtaining good approximation algorithms for Dasgupta's cost and here the authors show how to exploit it for the framework of advice.

A meta-comment here, is that the authors, through this ML oracle splitting aadvice, they are able to go beyond hardness results that hold for the HC objectives in the standard worst-case input (i.e. without the advice).

**Claims And Evidence:**

Yes.

**Essential References Not Discussed:**

I believe that it's better to point out why adversarial noise is important: constructing the optimal tree is easy if the oracle always gives the correct triplet. (check Aho et al. "Inferring a tree from lowest common ancestors with an application to the optimization of relational expressions.", 1981). Studying models where triplets (or even quartets info is given, has been studied in other works listed below, and it would be good to provide some comparison with your setting) Other related works not cited:


1. Chatziafratis, Vaggos, Rad Niazadeh, and Moses Charikar. "Hierarchical clustering with structural constraints." International conference on machine learning. PMLR, 2018.

2. Ghoshdastidar, Debarghya, Michaël Perrot, and Ulrike von Luxburg. "Foundations of comparison-based hierarchical clustering." Advances in neural information processing systems 32 (2019).

3. Alon, Noga, Yossi Azar, and Danny Vainstein. "Hierarchical clustering: A 0.585 revenue approximation." Conference on Learning Theory. PMLR, 2020.

4. Vaggos Chatziafratis, Mohammad Mahdian, Sara Ahmadian. "Maximizing Agreements for Ranking, Clustering and Hierarchical Clustering via MAX-CUT.", AISTATS 2021

5. Jiang, Tao, Paul Kearney, and Ming Li. "A polynomial time approximation scheme for inferring evolutionary trees from quartet topologies and its application." SIAM Journal on Computing 30.6 (2001): 1942-1961.

6. Snir, Sagi, and Raphael Yuster. "A linear time approximation scheme for maximum quartet consistency on sparse sampled inputs." SIAM Journal on Discrete Mathematics 25.4 (2011): 1722-1736.

7. Alon, Noga, Sagi Snir, and Raphael Yuster. "On the compatibility of quartet trees." SIAM Journal on Discrete Mathematics 28.3 (2014): 1493-1507.

Including these would better capture the background for introducing your problem.

**Experimental Designs Or Analyses:**

N/A

**Methods And Evaluation Criteria:**

The paper assumes we have access to a splitting oracle; all algorithms are based on this assumption. Appendix J shows that it is possible to learn such an oracle efficiently. Overall, this setting is reasonable for a theory paper, and it is interesting from a practical perspective as well.

**Other Comments Or Suggestions:**

-It seems that (1/2 + ε) should be (1/2 - ε) in the context of discussing success probabilities (Appendix I.1, lines 2803-2806). PLease check.


-It would be better to include some overview or intuition of the proofs in the main text!

**Other Strengths And Weaknesses:**

Strengths:

-This work is the first time that combines Learning-Augmented algorithms with HC, which is creative. The paper is well-written.

-The reviewer thinks this is a very natural setting, and the authors do a good job motivating and explaining overall their approach.

-The algorithms and proofs are very interesting and constitute solid contributions.

-All results are interesting in their own right but also when comparing with some expected hardness of the problems studied and how the authors bypass it.

Weakness:

-I couldn't find serious weaknesses (other than some minor literature omissions). Overall, this is a nice theoretical contribution that could benefit from a discussion about the lower bounds (samples, runtime etc.) and/or some experimental section, though I don't think this reduces the value of the contribution as is.

**Questions For Authors:**

For constructing weakly consistent trees: if the input has m edges, what would be the runtime of your algo? why is ~O(n^2) is considered near-linear? Do you assume that all edges are present or sth like this? I think you should clarify that.

**Relation To Broader Scientific Literature:**

The paper is studying HC which is at the heart of many different applications. So as such, it presents interesting results to many other papers from the litearture.

**Theoretical Claims:**

This is a proof-heavy paper. I checked the main body and proofs from Appendix B (Technical Overview), and overall I believe it is solid.

---

> ### Author Rebuttal · Authors · 2025-04-01
>
> Thank you for your careful review and positive evaluation. Our responses are as follows.
>
> > Additional References
>
> We agree with the reviewer that discussing the related papers and in particular the role of adversarial noise will help demonstrate the challenges with constructing a near-optimal tree. We thank the reviewer for pointing out the additional references; we have added these discussions to the new version.
>
> > O(n^2) time vs. ‘near-linear’.
>
> Due to the construction of weak partial trees, the run time of the algorithm is $\tilde{O}(n^2)$ even if the input graph has $m=o(n^2)$ edges. However, since the similarity graph on $n$ vertices has $\Omega(n^2)$ edges, our algorithm is still near-linear time. We have clarified this point in the updated version.
>
> Furthermore, we believe it is possible to revise our weakly consistent partial tree algorithm to input sparsity runtime, i.e., $o(n^2)$ runtime when the similarity graph is sparse. This could be an interesting direction to pursue as a future step.
>
> > Other Comments or Suggestions.
>
> We have addressed these points – thanks for the suggestions!

---

### Decision · Program_Chairs · 2025-05-01

**Decision:**

Accept (poster)

**Comment:**

The submission studies learning-augmented algorithms for hierarchical clustering. Most reviewers are very positive about the submission, and I think that the weakness pointed out by the reviewers can be easily addressed. Hence, I recommend to accept the paper.